# Targeting PIKfyve-driven lipid metabolism in pancreatic cancer

Caleb Cheng[1,2,3], Jing Hu[1,4,5], Rahul Mannan[1,4], Tongchen He[1,6], Rupam Bhattacharyya[1,4], Brian Magnuson[1,4], Jasmine P. Wisniewski[1], Sydney Peters[1], Saadia A. Karim[7], David J. MacLean[7], Hüseyin Karabürk[8], Li Zhang[9], Nicholas J. Rossiter[3], Yang Zheng[1,4], Lanbo Xiao[1,4,10], Chungen Li[11], Dominik Awad[9], Somnath Mahapatra[1,4], Yi Bao[1,4], Yuping Zhang[1,4], Xuhong Cao[1,4,12], Zhen Wang[11], Rohit Mehra[1,4], Pietro Morlacchi[13], Vaibhav Sahai[10,14], Marina Pasca di Magliano[10,15], Yatrik M. Shah[9,10,16], Lois S. Weisman[8,17], Jennifer P. Morton[7,18], Ke Ding[11], Yuanyuan Qiao[1,4,10 ✉], Costas A. Lyssiotis[9,10,16 ✉] & Arul M. Chinnaiyan[1,4,10,12,19 ✉]

Pancreatic ductal adenocarcinoma (PDAC) subsists in a nutrient-deregulated microenvironment, making it particularly susceptible to treatments that interfere with cancer metabolism[1,2]. For example, PDAC uses, and is dependent on, high levels of autophagy and other lysosomal processes[3–5]. Although targeting these pathways has shown potential in preclinical studies, progress has been hampered by the difficulty in identifying and characterizing favourable targets for drug development[6]. Here, we characterize PIKfyve, a lipid kinase that is integral to lysosomal functioning[7], as a targetable vulnerability in PDAC. Using a genetically engineered mouse model, we established that PIKfyve is essential to PDAC progression. Furthermore, through comprehensive metabolic analyses, we found that PIKfyve inhibition forces PDAC to upregulate a distinct transcriptional and metabolic program favouring de novo lipid synthesis. In PDAC, the KRAS–MAPK signalling pathway is a primary driver of de novo lipid synthesis. Accordingly, simultaneously targeting PIKfyve and KRAS–MAPK resulted in the elimination of the tumour burden in numerous preclinical human and mouse models. Taken together, these studies indicate that disrupting lipid metabolism through PIKfyve inhibition induces synthetic lethality in conjunction with KRAS–MAPK-directed therapies for PDAC.

PDAC is one of the deadliest cancers, having a five-year survival rate of just 13%[8]. This is mediated in large part by a lack of effective therapeutic options. The PDAC tumour microenvironment is central to this resistance and features a lot of stromal fibroblasts and extracellular matrix deposition that cause PDAC to experience elevated interstitial pressure, low vascularity and disrupted nutrient availability[9]. To circumvent this poor nutrient access, PDAC cells have become excellent scavengers, using intracellular and extracellular recycling pathways, sourcing non-classical nutrients from their environment through the expression of high-avidity nutrient transporters, bulk engulfment and crosstalk with other pro-tumour cell types[3,5,9–13].

Lysosome-dependent pathways have multiple roles in PDAC. For example, these pathways have been shown to maintain the availability of biosynthetic intermediates[3,5,10], regulate iron homeostasis[14–17], degrade MHC-1, increasing immune evasion[18], and adapt to the inhibition of

Kirsten rat sarcoma virus (KRAS) or the downstream mitogen-activated protein kinase (MAPK) pathway[19,20]. Studies exploring this have provided support for targeting autophagy and lysosome-dependent pathways to disrupt PDAC metabolism as a therapeutic strategy, and have resulted in many clinical trials using autophagy and the lysosomal inhibitor hydroxychloroquine (HCQ) with chemotherapy or MAPK inhibitors in PDAC (NCT01273805, NCT01978184, NCT01506973, NCT04911816, NCT04524702, NCT01494155, NCT03344172, NCT04386057 and NCT04132505)[21–25].

Despite the considerable interest in, and potential of, targeting autophagy and lysosomal processes in PDAC, preclinical and clinical studies have been hampered by the lack of effective therapeutics targeting specific effectors of these processes[26]. The lipid kinase PIKfyve is the only cellular source of phosphatidylinositol 3,5-bisphosphate (PtdIns(3,5)P₂) and phosphatidylinositol 5-phosphate (PtdIns5P), which

[1]Michigan Center for Translational Pathology, University of Michigan, Ann Arbor, MI, USA. [2]Medical Scientist Training Program, University of Michigan, Ann Arbor, MI, USA. [3]Cellular and Molecular Biology Program, University of Michigan, Ann Arbor, MI, USA. [4]Department of Pathology, University of Michigan, Ann Arbor, MI, USA. [5]Department of Pathology, Qilu Hospital, Cheeloo College of Medicine, Shandong University, Jinan, People's Republic of China. [6]Department of Urology, Xiangya Hospital, Central South University, Changsha, People's Republic of China. [7]CRUK Scotland Institute, Glasgow, UK. [8]Department of Cell and Developmental Biology, University of Michigan, Ann Arbor, MI, USA. [9]Department of Molecular and Integrative Physiology, University of Michigan, Ann Arbor, MI, USA. [10]Rogel Cancer Center, University of Michigan, Ann Arbor, MI, USA. [11]State Key Laboratory of Chemical Biology, Shanghai Institute of Organic Chemistry, Chinese Academy of Sciences, Beijing, People's Republic of China. [12]Howard Hughes Medical Institute, University of Michigan, Ann Arbor, MI, USA. [13]Agilent Technologies, Lexington, MA, USA. [14]Department of Internal Medicine, Division of Hematology and Oncology, University of Michigan, Ann Arbor, MI, USA. [15]Department of Surgery, University of Michigan, Ann Arbor, MI, USA. [16]Department of Internal Medicine, Division of Gastroenterology, University of Michigan, Ann Arbor, MI, USA. [17]Life Sciences Institute, University of Michigan, Ann Arbor, MI, USA. [18]School of Cancer Sciences, University of Glasgow, Glasgow, UK. [19]Department of Urology, University of Michigan, Ann Arbor, MI, USA. ✉e-mail: qiaoy@med.umich.edu; clyssiot@umich.edu; arul@med.umich.edu

are signalling lipids crucial for lysosome activity and autophagy[7,27]. Previous work has shown that inhibition of PIKfyve decreased the cellular levels of PtdIns(3,5)P$_2$ and PtdIns5P[28] and disrupted autophagy flux and lysosome function, leading to increased immune activity and tumour suppression in multiple cancer models[29–32]. Importantly, two PIKfyve inhibitors, apilimod and ESK981, have passed phase 1 clinical trials (NCT02594384 and NCT00875264)[33,34], highlighting the rapid translational potential of targeting PIKfyve to disrupt autophagy and lysosomal processes in cancers.

To this end, we sought to evaluate PIKfyve as a therapeutic target in PDAC and found that genetic knockout or pharmacological inhibition of PIKfyve markedly reduced PDAC tumour development and growth. Using a multi-omics approach to characterize the metabolic impact of PIKfyve inhibition on PDAC cells, we found that PIKfyve inhibition created a synthetic dependency on de novo lipid synthesis. We also found that de novo lipid synthesis was driven by the KRAS–MAPK pathway and, accordingly, therapeutic inhibition of both PIKfyve and KRAS–MAPK signalling resulted in sustained tumour regression or elimination in multiple mouse models of PDAC, including the *Pft1a-Cre;LSL-Kras*$^{G12D/+}$; *LSL-Trp53*$^{R172H/+}$ (KPC) autochthonous model. Taken together, our findings establish that PIKfyve is a targetable metabolic vulnerability in PDAC and demonstrate that dual inhibition of PIKfyve and KRAS–MAPK, to disrupt coordinated lipid homeostasis, is a promising and rapidly translatable therapeutic strategy to treat PDAC.

## *Pikfyve* is dispensable in healthy pancreas

To study the role of *Pikfyve* in pancreatic cancer development, we first evaluated *Pikfyve* expression in the autochthonous PDAC genetically engineered mouse model KPC. Using BaseScope, which is an RNA in situ hybridization (RNA-ISH) technique that has a short probe specifically targeting *Pikfyve* exon 6, we discovered that *Pikfyve* expression was greatly and consistently higher in pancreatic intra-epithelial neoplasia (PanIN) and PDAC tissue than in the surrounding healthy tissue in situ (Fig. 1a,b). Corroborating this, using RNA-ISH on a panel of human PDAC samples, we found that *PIKFYVE* was overexpressed in PDAC cells compared with matched, surrounding healthy pancreatic cells (Fig. 1c and Extended Data Fig. 1a). We investigated the rationales for the increased *PIKFYVE* expression in PDAC and found that, although some characteristics of PDAC, such as hypermethylation of *IKZF1* (Extended Data Fig. 1b,c) and hyperproliferation (Extended Data Fig. 1c), did not consistently affect *PIKFYVE* levels, increased cell density, serum starvation and lipid deprivation correlated with higher *PIKFYVE* expression in both healthy pancreas and PDAC cells (Extended Data Fig. 1e,f). These data indicate that PanIN and PDAC may have an elevated utilization of PIKfyve-driven processes relative to healthy pancreatic tissue to adapt to the nutrient-disrupted microenvironment.

To assess whether PDAC cells have an elevated dependence on PIKfyve, we first evaluated the essentiality of PIKfyve in normal pancreatic development. To this end, we generated conditional pancreatic *Pikfyve*-knockout mice by using the *Ptf1a* promoter-driven Cre recombinase (*Ptf1a-Cre;Pikfyve*$^{f/f}$) (Extended Data Fig. 1g). After confirming the loss of PIKfyve protein in pancreatic tissue (Extended Data Fig. 1h,i), we assessed the physiological impact of *Pikfyve* loss on pancreatic development. *Pikfyve* loss did not affect pancreatic weight, morphology or function in terms of insulin production (Extended Data Fig. 1j,k), indicating that *Pikfyve* is not essential for normal pancreatic tissue development or function.

## PDAC tumorigenesis requires *Pikfyve*

We then sought to evaluate the effect of *Pikfyve* loss on PDAC development by crossing *Pikfyve*$^{+/+}$, *Pikfyve*$^{f/+}$ and *Pikfyve*$^{f/f}$ with the KC model (*Ptf1a-Cre;LSL-Kras*$^{G12D/+}$) to assess pancreatic tumorigenesis (Fig. 1d and Extended Data Fig. 2a,b). In monitoring these cohorts of mice,

we found that *Pikfyve* loss substantially extended the survival of mice with the KC genotype (Fig. 1e). To determine whether this was correlated with a difference in pancreatic disease burden, we evaluated the pancreata from a separate cohort of mice and found that, compared with pancreata from their KC *Pikfyve*$^{+/+}$ littermates, pancreata from KC *Pikfyve*$^{f/+}$ and KC *Pikfyve*$^{f/f}$ mice weighed less and were closer in weight to the pancreata of wild-type mice at 27 weeks of age (Extended Data Fig. 2c,d). Furthermore, pancreata from KC mice with *Pikfyve* loss retained a higher degree of normal histological structures, based on haematoxylin and eosin (H&E) or immunohistochemistry (IHC) staining for cytokeratin 19 (CK19) (Extended Data Fig. 2e,f). Consistent results were recapitulated at a later age of 40 weeks, both on macroscopic and microscopic evaluations (Extended Data Fig. 2g,h).

We next evaluated the role of *Pikfyve* in the KPC model to assess the effect of PIKfyve on tumour progression (Extended Data Fig. 2i–k). We analysed pancreata from 15 mice in the KPC *Pikfyve*$^{+/+}$ cohort and 16 mice in the KPC *Pikfyve*$^{f/f}$ cohort and found that the pancreata of the KPC *Pikfyve*$^{f/f}$ mice weighed significantly less than those of KPC *Pikfyve*$^{+/+}$ mice, relative to their total body weight (Fig. 1f). To determine whether this effect was correlated with a decrease in disease onset or development, we performed histopathological analysis on these pancreata and observed that the pancreata of KPC *Pikfyve*$^{f/f}$ mice displayed a significantly lesser degree of disease onset and progression than the pancreata of KPC *Pikfyve*$^{+/+}$ mice (Fig. 1g,h) at similar ages (Extended Data Fig. 2l). Taken together, these data indicate that *Pikfyve* loss suppresses pancreatic cancer onset and progression in the KC and KPC models, respectively, without affecting normal pancreatic tissue. Collectively, these data indicate that PDAC has an elevated requirement for PIKfyve-driven processes.

## PIKfyve inhibition blunts PDAC growth

Given that genetic perturbation of *Pikfyve* attenuated PDAC development, we sought to evaluate whether pharmacological PIKfyve inhibition would elicit similar effects. We first used a cellular thermal shift assay (CETSA) to confirm that apilimod and ESK981, which are PIKfyve inhibitors that have cleared phase 1 clinical trials[33,34], bind to mouse PIKfyve protein (Fig. 2a). Importantly, PIKfyve inhibition decreased the levels of PtdIns(3,5)P$_2$ and PtdIns5P in a time-dependent manner for time points as short as 30 minutes (Fig. 2b,c). Consistently, PIKfyve perturbation also increased the amount of PtdIns3P, without noticeably affecting the levels of PtdIns4P or PtdIns(4,5)P$_2$ (Extended Data Fig. 3a–c). To evaluate the impact of PIKfyve inhibition on PDAC development, we prophylactically treated a cohort of six-week-old KPC mice with ESK981 for four weeks (Fig. 2d). At the end point of ten weeks, we found that the weights of KPC pancreata treated with ESK981 were reduced to levels approaching those of wild-type pancreata (Fig. 2e). Furthermore, ESK981-treated pancreata retained a higher degree of histopathologically unremarkable pancreatic tissue and had relatively reduced PanIN and PDAC burden (Fig. 2f and Extended Data Fig. 3d,e).

To determine the impact of PIKfyve inhibition on PDAC tumour growth, we used in vivo allograft and xenograft models to test the efficacy of ESK981 (Extended Data Fig. 3f). ESK981 therapy decreased the tumour burden of mice that had KPC 7940B (7940B)-derived orthotopic tumours without affecting the host's body weight (Fig. 2g–i and Extended Data Fig. 3g). Similarly, in an orthotopic UM19 (*KRAS*$^{G12D}$) primary cell-derived xenograft (pCDX), ESK981 treatment also greatly decreased the tumour burden (Fig. 2j,k and Extended Data Fig. 3h). ESK981 also blunted tumour growth and end-point tumour burden in subcutaneous allografts of KPC 1344 cells (Extended Data Fig. 3i,j), as well as in xenograft models of MIA PaCa-2 and BxPC-3 cells (Extended Data Fig. 3k–o). In both the MIA PaCa-2 and BxPC-3 CDX models, ESK981 treatment reduced the proliferation of these tumours, according to Ki-67 staining (Extended Data Fig. 3p). Furthermore, ESK981 treatment induced substantial apoptosis in MIA PaCa-2 models, as well as in UM2

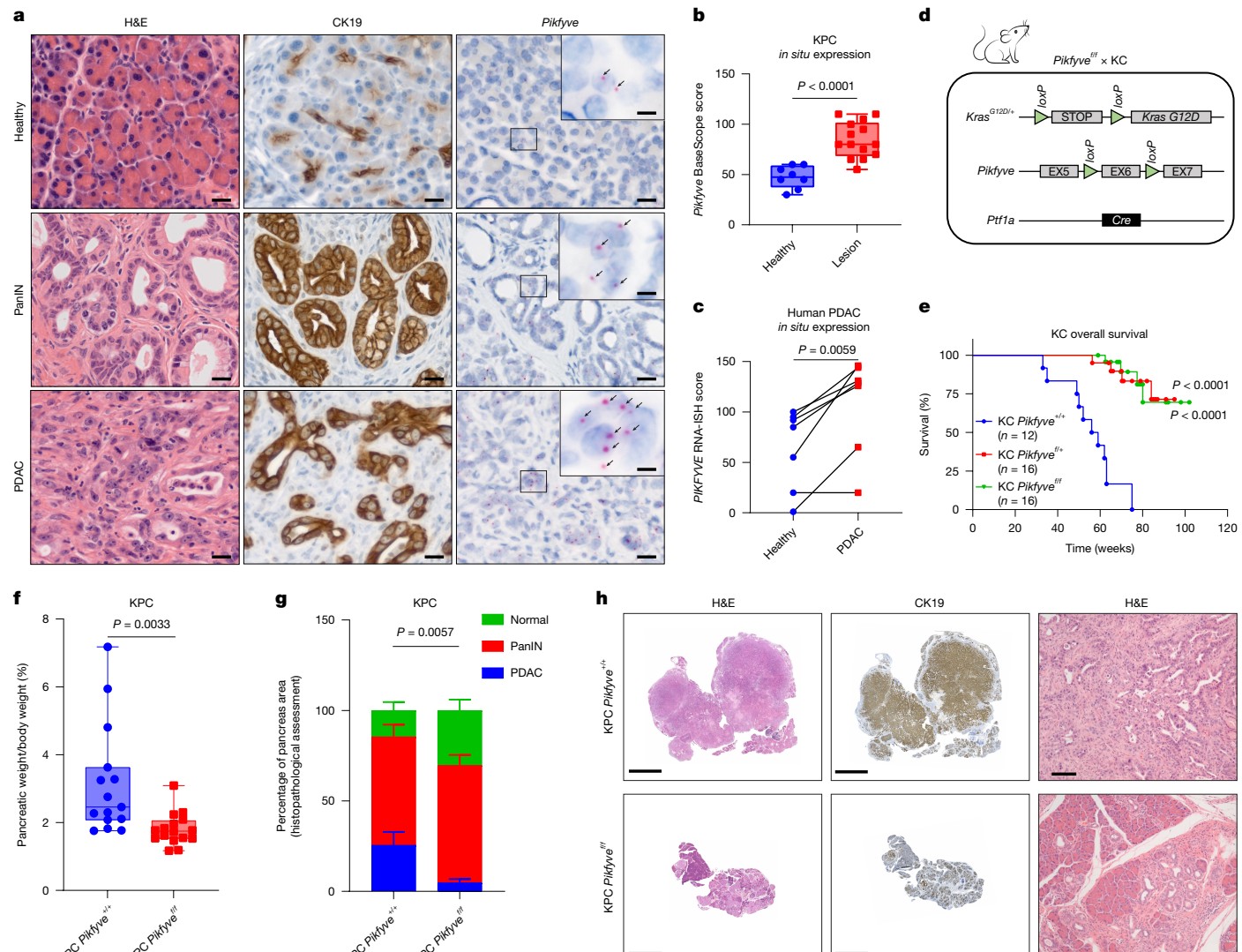

**Fig. 1 | *Pikfyve* is essential for progression of precursor PanIN lesions to PDAC. a**, Representative images of PanIN and PDAC lesions and healthy tissue from a KPC mouse pancreas, showing H&E IHC staining for CK19 and BaseScope for *Pikfyve*. Arrows (insets) indicate areas of BaseScope positivity. **b**, In situ *Pikfyve* levels in KPC mouse pancreas lesion (PanIN or PDAC) versus normal (healthy) tissue, determined by BaseScope RNA-ISH probes targeting *Pikfyve* exon 6. Boxes represent the 25th and 75th percentiles; whiskers represent the range (unpaired two-tailed *t*-tests). **c**, In situ *PIKFYVE* levels in histologically normal or PDAC cells in seven human PDAC patient samples using RNA-ISH (RNAScope) (paired two-tailed *t*-tests). **d**, Breeding design for the generation of KC *Pikfyve*[+/+], KC *Pikfyve*[f/+] and KC *Pikfyve*[f/f] mice. Mouse cartoon adapted from Adobe Stock Image (asset no. 304271210). **e**, Overall survival of KC *Pikfyve*[+/+], KC *Pikfyve*[f/+] and KC *Pikfyve*[f/f] mice (two-sided log-rank test comparing KC *Pikfyve*[+/+] with the other two curves separately). **f**, Pancreas tissue weight normalized to total body weight from KPC *Pikfyve*[+/+] or KPC *Pikfyve*[f/f] mice at death. Boxes represent 25th and 75th percentiles; whiskers represent the range (unpaired two-tailed *t*-tests). **g**, Percentage of pancreas occupied by normal, PanIN or PDAC tissue at death. Bars are ±s.d. (two-way analysis of variance (ANOVA)). **h**, Representative histology showing CK19 IHC and H&E staining of whole pancreatic tissue from KPC *Pikfyve*[+/+] and KPC *Pikfyve*[f/f] mice at 25 weeks. Scale bars: **a**, 20 μm (main images), 5 μm (insets); **h**, 5 mm (left and middle), 100 μm (magnified images, right).

---

(*KRAS*[Q61L]) pCDX, as shown by increased terminal dUTP nick end labelling (TUNEL) staining and PARP cleavage (Extended Data Fig. 3q–s), and induced tumour regression after five days of treatment (Extended Data Fig. 3t). Taken together, these results indicate that pharmacological PIKfyve inhibition, using compounds such as ESK981, suppresses PDAC development and growth and is well tolerated in mice.

## PIKfyve drives PDAC lysosomal processes

To determine the molecular effects of PIKfyve inhibition on PDAC cells, we used a battery of methods to perturb PIKfyve. First, we found that CRISPR interference-mediated *PIKFYVE* knockdown in the human PDAC cell lines MIA PaCa-2 and PANC1 decreased autophagic flux (Extended Data Fig. 4a,b), which is consistent with previous reports[29,30].

Pharmacological inhibition of PIKfyve with apilimod or ESK981 also showed similar effects in 7940B and Panc 04.03 (human PDAC) cells (Extended Data Fig. 4c), as well as in vivo in *UM-2* pCDX tumours (Extended Data Figs. 3r–t, 4d). Using a GFP–LC3-RFP-LC3ΔG autophagic flux probe[35], we found that treatment with apilimod, ESK981 or chloroquine decreased basal autophagic flux, as well as autophagic flux, induced by mTORC inhibition with torin-1 (Extended Data Fig. 4e,f). Finally, as further confirmation of target specificity, we developed a second-generation proteolysis-targeting chimera degrader of PIKfyve, PIK5-33d, which was based on our previously described PIKfyve degrader[36] (Extended Data Fig. 4g). PIK5-33d potently degraded PIKfyve, and this phenocopied the autophagy inhibition and changes in PIP levels elicited by *PIKFYVE* knockdown or its enzymatic inhibition (Extended Data Fig. 4h–m).

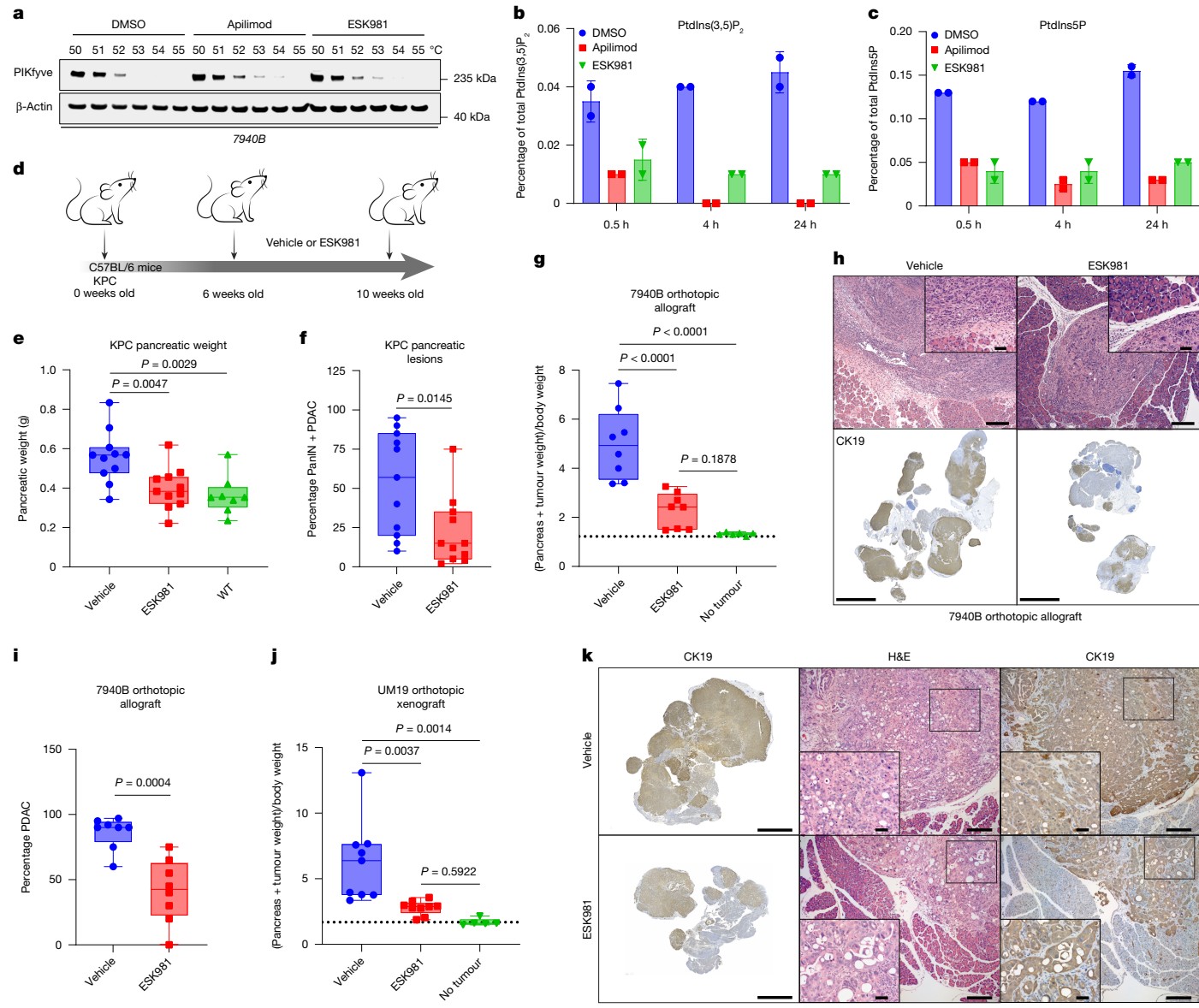

**Fig. 2 | Pharmacological inhibition of PIKfyve blocks pancreatic cancer progression in vivo. a**, Immunoblot analysis demonstrating stabilization of PIKfyve by apilimod (1,000 nM) or ESK981 (1,000 nM) in a cellular thermal shift assay in 7940B cells. **b,c**, Relative abundance of PtdIns(3,5)$P_2$ (**b**) and PtdIns5P (**c**) in PANC1 cells following treatment with apilimod (1,000 nM) or ESK981 (1,000 nM) for the time indicated. Bars show ±s.d. **d**, Schematic of a prophylactic efficacy study of ESK981 on KPC mice. Mouse cartoon adapted from Adobe Stock Image (asset no. 304271210). **e**, Pancreatic tissue weight in vehicle or ESK981-treated KPC mice in comparison with age-matched wild-type (WT) mice. Boxes represent the 25th and 75th percentiles; whiskers represent the range (one-way ANOVA with Dunnett's test). **f**, Quantification of lesions (PanIN or PDAC) in samples from **e**. Boxes represent the 25th and 75th percentiles; whiskers represent the range (unpaired two-tailed *t*-test). **g**, End-point pancreas + tumour weight normalized to total body weight

of mice with 7940B orthotopic tumours. Pancreata of six age-matched non-tumour-bearing C57BL/6 mice were used as references. Boxes represent 25th and 75th percentiles; whiskers represent the range (one-way ANOVA with Tukey's test). **h**, Representative H&E (top) and CK19 (bottom) images of pancreatic and pancreatic tumour tissues from **g**. **i**, Quantification of lesions from the 7940B orthotopic samples in **g**. Boxes represent 25th and 75th percentiles; whiskers represent the range (unpaired two-tailed *t*-tests). **j**, End-point pancreas + tumour weight normalized to total body weight of CB17 SCID mice with UM19 orthotopic tumours. Pancreata of five age-matched non-tumour-bearing CB17 SCID mice were used as references. Boxes represent 25th and 75th percentiles; whiskers represent the range (one-way ANOVA with Tukey's test). **k**, Representative histology images showing CK19 IHC and H&E staining of the samples from **j**. Scale bars: **h**, 200 μm (top, main images), 5 μm (top insets), 5 mm (bottom); **k**, 5 mm (left), 200 μm (middle and right, main images), 50 μm (insets).

Consistent with previous work, treatment with PIKfyve inhibitors or degrader, or *PIKFYVE* knockdown, induced a vacuolization phenotype in PDAC cells[29,30] (Extended Data Fig. 4n,o). PIKfyve perturbation through *PIKFYVE* knockdown also substantially slowed the proliferation of PDAC cells (Extended Data Fig. 4p), and PIKfyve inhibition decreased PDAC cell viability with half-maximal inhibitory concentrations ($IC_{50}$) in the nanomolar ranges for most cell lines (Extended Data Fig. 4q). Lysosome inhibition by chloroquine treatment also decreased PDAC cell

viability (Extended Data Fig. 4q). However, the $IC_{50}$ values were much higher for chloroquine than for apilimod or ESK981 in the same PDAC cell lines (Extended Data Fig. 4r,s). Collectively, these data illustrate that PIKfyve has a crucial role in lysosomal function and, ultimately, in cell proliferation in PDAC.

PDAC is known to use lysosomal processes, including autophagy, to promote iron homeostasis and allow for mitochondrial respiration[14,15], so we investigated whether PIKfyve inhibition decreased PDAC cell

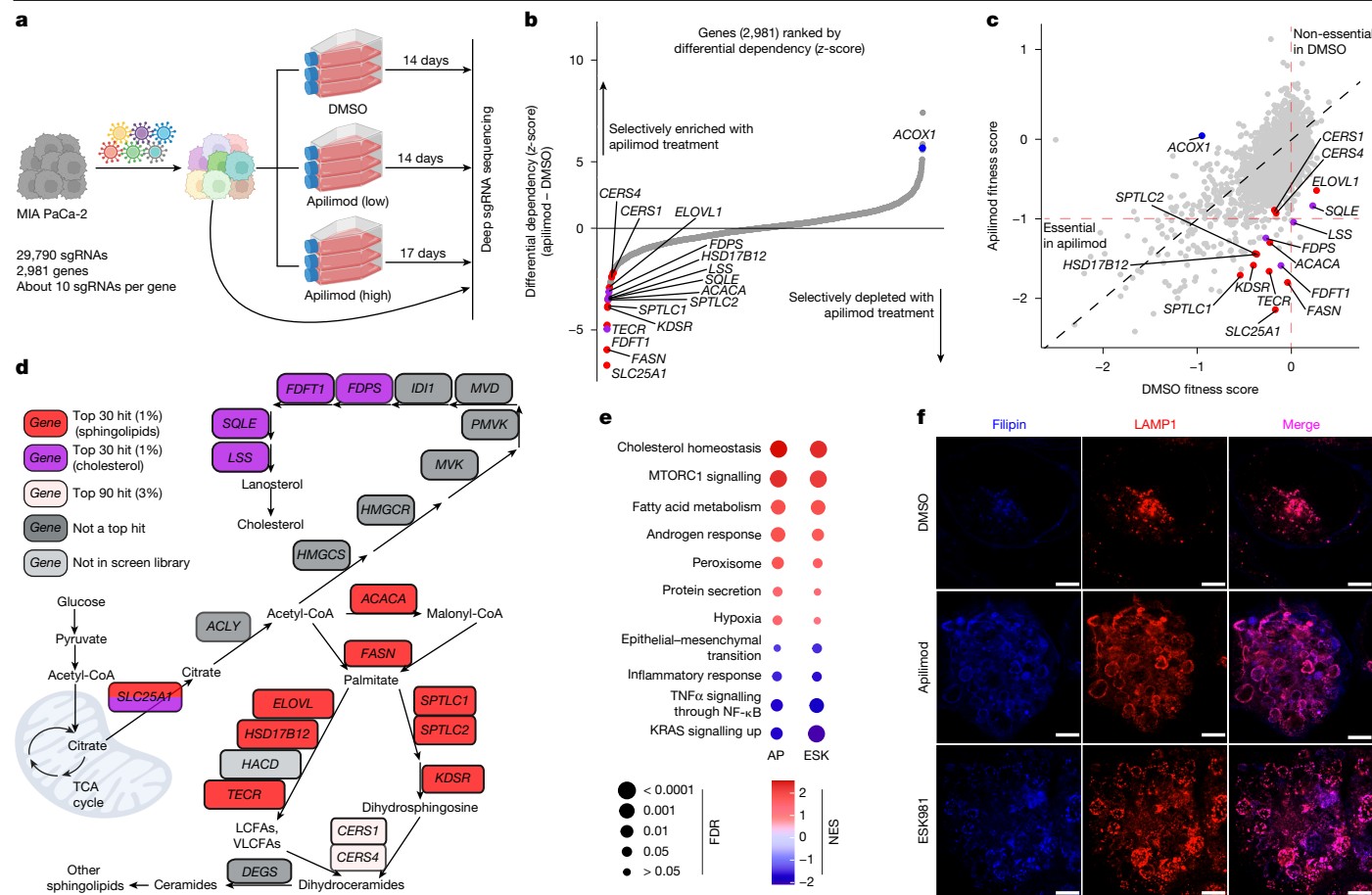

**Fig. 3 | PIKfyve inhibition obligates PDAC cells to stimulate a lipogenic transcriptional and metabolic program. a**, Schematic of the metabolism-focused CRISPR screen in MIA PaCa-2 cells. Created in BioRender. Cheng, C. (2025) https://BioRender.com/d149928. **b**,**c**, Gene enrichment rank plot-based differential sgRNA representation (**b**) and scatter plot of gene fitness scores (**c**) of high dose (2,000 nM) and low dose (100 nM) apilimod-treated versus DMSO-treated end-point populations of the CRISPR screen experiment. Graphs show the top 30 synthetically lethal genes involved in fatty acid and sphingolipid synthesis (red), the top 30 synthetically lethal genes involved in cholesterol synthesis (purple) and genes that confer sensitivity to apilimod (blue). **d**, Metabolic map of sphingolipid and cholesterol synthesis. The figure shows the top 90 synthetically lethal genes involved in fatty acid and sphingolipid

synthesis (red), the top 90 synthetically lethal genes involved in cholesterol synthesis (purple), genes ranked as essential in both DMSO and apilimod conditions (dark grey) and genes not included in the CRISPR screen library (light grey). Created in BioRender. Cheng, C. (2025) https://BioRender.com/yzgsylk. **e**, Pathway enrichment analysis of RNA-seq performed on 7940B cells treated with either apilimod (AP, 25 nM) or ESK981 (ESK, 250 nM) for 8 h. Dot sizes are inversely proportional to the false discovery rate (FDR). The colour scheme reflects the normalized enrichment score (NES). **f**, Immunofluorescence images of PANC1 cells treated with DMSO, apilimod (1,000 nM) or ESK981 (1,000 nM) for 24 h stained with filipin or LAMP1. Scale bars, 5 μm. These are cropped images shown for the sake of focus; the full images are shown in Extended Data Fig. 9.

proliferation through a similar mechanism. PIKfyve inhibition stabilized HIF1α after eight hours of treatment (Extended Data Fig. 4t), consistent with the effect of iron deprivation caused by disrupting autophagy. However, PIKfyve inhibition did not decrease the basal oxygen consumption rate (OCR), contrasting with the activity of chloroquine and bafilomycin A1 (Extended Data Fig. 4u,v). Furthermore, although the antiproliferative effects of bafilomycin were greatly attenuated by the addition of FAC, we did not see a similar effect with PIKfyve inhibitors (Extended Data Fig. 4w–y). Overall, these data indicate that autophagy and lysosomal perturbation through PIKfyve inhibition does not decrease PDAC proliferation by disrupting iron homeostasis and mitochondrial respiration, but rather by a different mechanism.

## Lipogenesis compensates for PIKfyve loss

To assess the functionally relevant metabolic roles of PIKfyve in PDAC in an unbiased manner, we used two parallel metabolism-focused CRISPR screens in MIA PaCa-2 cells using different doses of PIKfyve inhibition (Fig. 3a and Extended Data Fig. 5a). Interestingly, the most significantly

depleted single guide RNAs (sgRNAs) in the high-dose screen targeted genes core to the de novo cholesterol, fatty acid synthesis, fatty acid elongation and sphingolipid synthesis pathways, namely *SLC25A1*, *FDFT1*, *SQLE*, *FDPS*, *LSS*, *FASN*, *ACACA*, *ELOVL1*, *HSD17B12*, *TECR*, *SPTLC1*, *SPTLC2* and *KDSR* (Fig. 3b–d and Supplementary Table 1). Accordingly, the low-dose screen also highlighted the de novo fatty acid synthesis and elongation pathway, including genes such as *FASN*, *ACACA*, *SLC25A1*, *ELOVL1* and *HSD17B12* (Extended Data Fig. 5b,c), indicating that this pathway was probably the most favourable to target alongside PIKfyve. Notably, in both screens, *ACOX1*, which completes the first step of lipid beta-oxidation, was the target of some of the most significantly enriched sgRNAs in the screen (Fig. 3b–d and Extended Data Fig. 5b). Altogether, these data support a synthetic lethal relationship between PIKfyve and de novo lipid synthesis.

To validate this screen, we perturbed several of the identified genes to determine whether loss of their activity rendered cells more sensitive to PIKfyve inhibition. Indeed, CRISPRi-mediated knockdown of *FASN* (Extended Data Fig. 5d,e) sensitized cells to apilimod and the PIKfyve degrader PIK5-33d (Extended Data Fig. 5f,g). Inhibition of ACC1,

the protein product of *ACACA*, with the compound ND646 (Extended Data Fig. 5h) similarly sensitized PDAC cells to apilimod, ESK981 and PIK5-33d (Extended Data Fig. 5i,k). Furthermore, knockdown of *SPTLC1* or *SPTLC2* (Extended Data Fig. 5l,m), which alone did not affect cell viability, sensitized PDAC cells to PIKfyve inhibition (Extended Data Fig. 5n). Finally, YM53601 (a FDFT1 inhibitor) or NB-598 (a SQLE inhibitor) similarly sensitized PDAC cells to PIKfyve inhibition (Extended Data Fig. 5o,p). These data indicate that PDAC cells have an increased dependency on the de novo synthesis of lipids, and ultimately cholesterols and sphingolipids, following PIKfyve inhibition.

## PIKfyve inhibition induces lipogenesis

Given that PIKfyve inhibition obligates PDAC cells to maintain expression and function of the de novo fatty acid-synthesis pathway, we assessed whether PIKfyve perturbation caused upregulation of this pathway. Using RNA-seq in 7940B cells, we determined that eight-hour treatment of apilimod or ESK981 induced remarkably concordant changes in gene expression (Extended Data Fig. 6a), and the most upregulated pathways were related to cholesterol homeostasis, MTORC1 signalling and fatty acid metabolism (Fig. 3e and Extended Data Fig. 6b). Contributing to this signature were many upregulated genes known to be involved in de novo lipogenesis and regulated by sterol regulatory element binding proteins[37] (SREBPs), such as *Fasn*, *Acaca*, *Fdps*, *Sqle*, *Hmgcr* and *Ldlr* (Extended Data Fig. 6c,d). Consistent with this, SREBP1 was activated by PIKfyve inhibition or degradation (Extended Data Fig. 6e), and downstream genes of SREBPs were indeed upregulated following inhibition or knockdown of PIKfyve (Extended Data Fig. 6f–l). Finally, blockade of SREBP processing and activation, using the pharmacological agent fatostatin, partly reversed the upregulation of lipogenic genes (Extended Data Fig. 6m,n). These results were consistently observed in three PDAC cell lines and a normal pancreas line, HPNE, which had lower basal levels of *SREBPF1*, *FASN* and *ACACA* (Extended Data Fig. 6o), and demonstrate that SREBP has a pivotal and conserved role in mediating the activation of the lipogenic gene signature following PIKfyve inhibition.

To determine whether the lipogenic transcriptional program translated to a metabolic phenotype, metabolic analyses of 7940B cells were used. PIKfyve inhibition, using apilimod or ESK981 treatment, induced a similar metabolic landscape (Extended Data Fig. 7a) featuring a decrease in citrate after three hours of treatment (Extended Data Fig. 7b). After eight hours, the citrate level had recovered to levels similar to the DMSO condition (Extended Data Fig. 7b); however, this was associated with a considerable decrease of upstream glycolytic metabolites (Extended Data Fig. 7c,d). Targeted lipidomics revealed that PIKfyve inhibition induced notable changes in the cellular lipid landscape in 7940B cells (Extended Data Fig. 7e), with a large increase in sphingolipids (hexosylceramides, sphingomyelins and ceramides) after 24 h of treatment (Extended Data Fig. 7f–h). Importantly, these shifts were partly dependent on SREBP1 activity (Extended Data Fig. 7i,j). Given that the citrate transporter SLC25A1 was also a top hit in the CRISPR screen (Fig. 3b–d and Extended Data Fig. 5b,c), we proposed that the glycolytic metabolites were being used to generate citrate, which was then shunted into de novo lipid synthesis. We confirmed this hypothesis by using stable isotope tracing with U-$^{13}$C$_6$ glucose and saw a large increase in $^{13}$C incorporation in long-chain ceramides after PIKfyve inhibition compared with the DMSO control (Extended Data Fig. 7k–r). These data indicate that PIKfyve inhibition causes PDAC cells to divert carbon from glucose to synthesize new lipids, particularly sphingolipids.

We sought to identify the mechanism through which PIKfyve regulates lipid homeostasis. Although AMP-activated protein kinase (AMPK) has validated roles in regulating SREBP[38], PIKfyve inhibition did not disrupt AMPK signalling (Extended Data Fig. 8a), and the effect of PIKfyve inhibition on SREBP signalling was not dependent on AMPK

(Extended Data Fig. 8b–e). Similarly, although PIKfyve inhibition disrupts autophagy, inhibition of ULK1, an autophagy initiator, did not affect SREBP signalling (Extended Data Fig. 8f–h). Further substantiating this, PIKfyve inhibition still activated SREBP signalling in autophagy-independent KPC 1361 cells (ATG5 and ATG7 CRISPR knockout)[30] (Extended Data Fig. 8i,j). Similarly, although ACC1 inhibition sensitized cells to apilimod, it did not sensitize cells to SBI to the same degree (Extended Data Fig. 8k–o). Finally, the ATG5 and ATG7 knockout cells were still sensitized to PIKfyve inhibition on ACC1 inhibition (Extended Data Fig. 8p–s). These data indicate that PIKfyve inhibition disrupts lipid homeostasis through a mechanism that does not depend solely on affecting AMPK signalling or autophagic flux.

PIKfyve has a critical role in regulating lysosomal function[7], and lysosomes are known to have an important role in providing cholesterol, as well as sphingoid bases such as sphingosine, to the cell[39]. Consistent with this, chloroquine and bafilomycin, two other non-PIKfyve lysosomal inhibitors, along with U18666A, an inhibitor of the lysosome cholesterol transporter NPC1, also activated SREBP1 signalling (Extended Data Fig. 9a–d). Furthermore, PIKfyve inhibition resulted in enlargement of LAMP1-positive vacuoles and substantial co-localization of filipin, indicating cholesterol accumulation at the lysosomal membranes (Fig. 3f and Extended Data Fig. 9e). By contrast, U18666A treatment resulted in substantial cholesterol accumulation at the lysosomal puncta, without lysosomal vacuolization, consistent with previous reports[40]. Importantly, SREBP activation and cholesterol accumulation at the lysosomal membranes were visible at the four-hour time point (Extended Data Fig. 10a–d), and SREBP activation on PIKfyve inhibition could be reversed by sterol supplementation (Extended Data Fig. 10e,f). Collectively, these data indicate that PIKfyve inhibition causes lysosomal vacuolization and dysfunction, leading to lipid sequestration at the lysosomal membranes and an induced dependency on de novo lipogenesis.

## KRAS–MAPK drives lipogenesis in PDAC

We next sought to identify avenues to leverage the synthetic lethality of PIKfyve and lipid synthesis. We focused specifically on regulation of de novo fatty acid synthesis because that pathway was most strongly nominated as synthetically essential following PIKfyve inhibition by the CRISPR screens. KRAS is known to be a core driver of metabolic homeostasis in PDAC through MAPK signalling and MYC-driven transcriptional regulation[41]. Consistent with previous studies, KRAS inhibition (with MRTX1133, a KRAS[G12D] inhibitor, or AMG510, a KRAS[G12C] inhibitor) or MEK inhibition (using trametinib) decreased expression of the MYC transcript and protein levels in PDAC cells (Extended Data Fig. 11a,b). Furthermore, KRAS or MEK inhibition decreased c-MYC binding to the *FASN* promoter (Extended Data Fig. 11c–e), which is consistent with previous reports[42], indicating that KRAS–MAPK signalling drove the expression of de novo fatty acid-synthesis genes in PDAC. By using an inducible *Kras*[G12D] cell line, iKras 9805, we found that doxycycline withdrawal (Kras OFF) decreased *Fasn* and *Acaca* expression at the transcript and protein levels (Extended Data Fig. 11f,g). Moreover, treatment with MRTX1133, AMG510 (AMG, a KRAS[G12C] inhibitor), or trametinib decreased mRNA and protein levels of FASN and ACACA (Fig. 4a and Extended Data Fig. 11h) in PDAC cell lines, which is concordant with previous reports[43] (Extended Data Fig. 11i,j). Taken together, these data illustrate that KRAS–MAPK signalling regulates the expression of FASN and ACC1 in PDAC.

We next directly assessed the effects of dual inhibition of PIKfyve and KRAS on FASN and ACC1. Whereas PIKfyve inhibition increased the transcript and protein levels of FASN and ACC1, concurrent *Kras* OFF and PIKfyve inhibition led to smaller increases of FASN and ACC1 compared with baseline (Fig. 4b,c). Consistent with this, concurrent PIKfyve inhibition and KRAS or MEK inhibition individually shifted the lipid profile in opposite directions, although the combination of

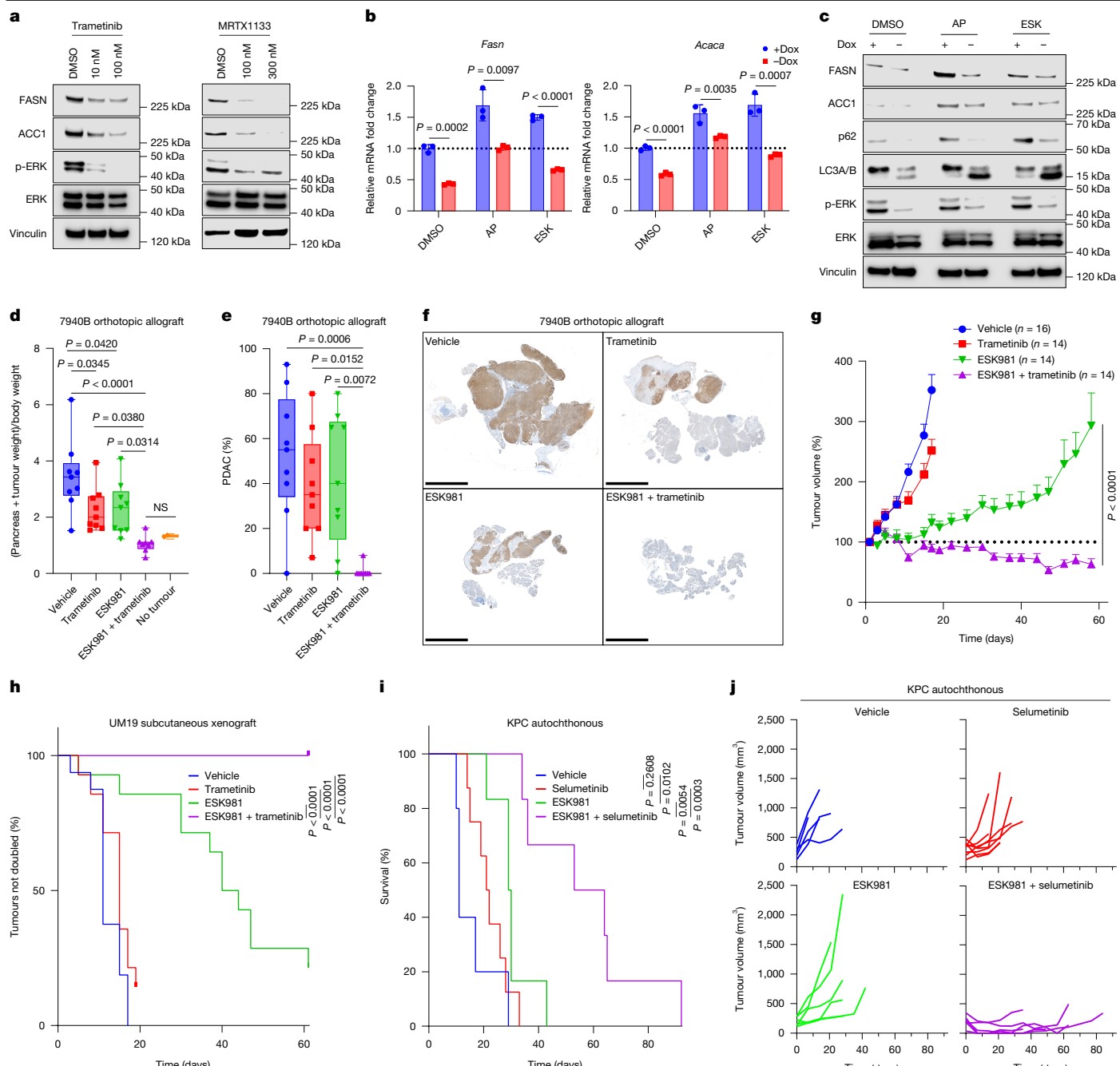

**Fig. 4 | Dual KRAS–MAPK and PIKfyve inhibition results in metabolic crises and synergistic growth suppression in PDAC. a**, Immunoblot analysis of 7940B cells treated with trametinib or MRTX1133 for 48 h. Vinculin served as a loading control. MRTX1133 and DMSO were refreshed every 12 h. **b**, Quantitative PCR (qPCR) of iKRAS 9805 cells after 48 h incubation with or without doxycycline (Dox) and subsequent 8-hour treatment with apilimod (AP, 50 nM), ESK981 (ESK, 300 nM) or DMSO for the genes *Fasn* and *Acaca*. Bars are ±s.d. (multiple unpaired two-tailed *t*-tests). **c**, Immunoblot analysis of iKRAS 9805 cells after 48 h incubation with or without doxycycline and subsequent 24 h treatment with apilimod (50 nM), ESK981 (300 nM) or DMSO. Vinculin served as a loading control. **d**, End-point pancreas + tumour weight normalized to total body weight. Pancreata of six age-matched non-tumour-bearing C57BL/6 mice were used as references. Boxes represent 25th and 75th percentiles; whiskers represent the range (one-way ANOVA with Tukey's test); NS, not significant. **e**, Quantification

of the proportion of PDAC in H&E sections from each tumour in **d**. Boxes represent 25th and 75th percentiles; whiskers represent the range (one-way ANOVA with Tukey's test). **f**, Representative images of CK19 IHC staining of one tumour from each treatment arm in **d**. Scale bar, 5 mm. **g**, Tumour volumes (as a percentage of the initial volume) over the treatment course of the UM19 tumour model treated with trametinib ± ESK981. Bars show s.e.m. (two-way ANOVA with Šidák's correction). The mice in the vehicle-treated and ESK981-treated groups are the same mice shown in Extended Data Fig. 12k–m. **h**, Kaplan–Meier estimates of time to tumour doubling for the mice in **g** (two-sided log-rank tests). **i**, Kaplan–Meier survival curves of KPC mice undergoing the treatments indicated (two-sided log-rank tests). The mice in the vehicle-treated and ESK981-treated groups are the same mice shown in Extended Data Fig. 12o–s. **j**, Tumour volumes of the KPC mice in **i**, measured by ultrasound.

PIKfyve and KRAS/MEK inhibition had a less-extreme shift in either direction (Extended Data Fig. 11k). Specifically, KRAS/MEK inhibition alone decreased sphingolipid levels and attenuated the increase in sphingolipid levels induced by PIKfyve inhibition (Extended Data Fig. 11l). Overall, these data indicate that KRAS–MAPK regulates de novo lipid synthesis and that KRAS–MAPK inhibition prevents PDAC cells

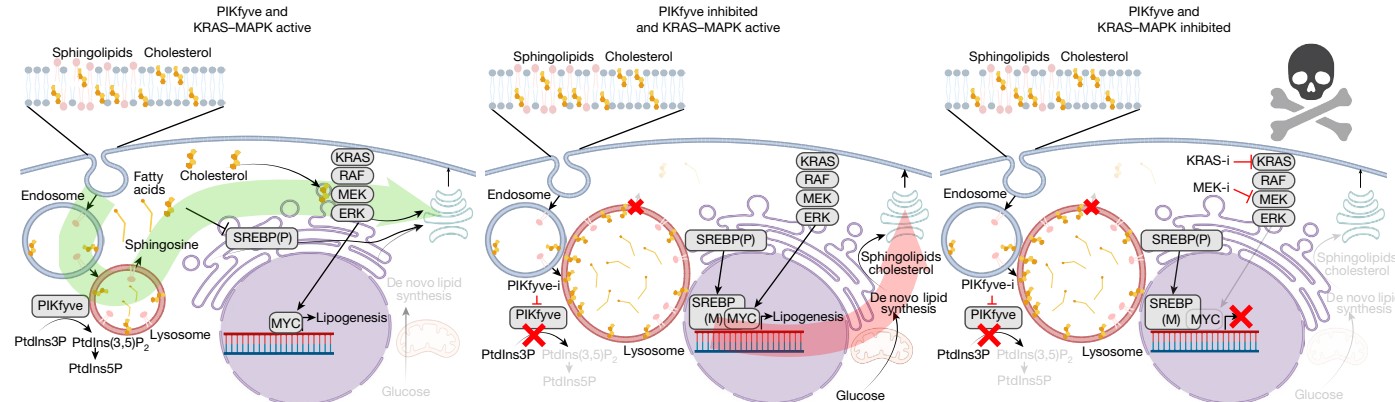

**Fig. 5 | Schematic depicting the effects of PIKfyve inhibition and KRAS–MAPK inhibition.** Left, with functional PIKfyve and KRAS–MAPK signalling, PDAC is at metabolic homeostasis, able to generate lipids through both de novo synthesis and lysosomal acquisition and recycling processes. Middle, with PIKfyve inhibition, lysosomal functions are disrupted, leading to lysosomal swelling or vacuolization and sequestering of lipids on lysosomal membranes.

This leads to a relative decrease in cellular lipid availability, forcing PDAC cells to activate de novo lipid synthesis, activating SREBP and pro-lipogenic transcriptional and metabolic programs. Right, concurrent PIKfyve and KRAS–MAPK inhibition blocks both the lysosomal recycling and acquisition and the de novo synthesis pathways of obtaining lipids, leading to synthetic lethality. Created in BioRender. Cheng, C. (2025) https://BioRender.com/d149928.

from utilizing the de novo synthesis of lipids such as sphingolipids to adapt to PIKfyve inhibition.

## PIKfyve enables protective autophagy

Important recent studies have shown that PDAC cells upregulate and depend on autophagy to maintain metabolic homeostasis following KRAS–MAPK signalling inhibition[19,20]. Using the autophagic flux probe, we confirmed that PDAC cells upregulate autophagy following acute mutant KRAS[G12D] inhibition with MRTX-1133 (Extended Data Fig. 11m). In alignment with our previous data, we found that PIKfyve inhibition can block the autophagic flux induced by KRAS–MAPK (Extended Data Fig. 11n–p). Taken together, these data indicate that concurrent PIKfyve and KRAS–MAPK inhibition drives PDAC into a state of metabolic conflict regarding its regulation of autophagic flux.

## Synergism of PIKfyve and KRAS–MAPK inhibitors

We next sought to assess whether the metabolic crises elicited by simultaneous inhibition of PIKfyve and KRAS–MAPK could be used to inhibit PDAC cell proliferation. Synergy assays confirmed that any combination of PIKfyve inhibition, using apilimod or ESK981, and KRAS–MAPK inhibition, using MRTX1133 or trametinib, resulted in striking synergistic effects, decreasing PDAC cell proliferation and viability (Extended Data Fig. 12a,b).

To determine the efficacy of combining PIKfyve and KRAS–MAPK inhibitors as a therapeutic strategy for PDAC, we first used a syngeneic orthotopic preclinical model (Extended Data Fig. 12c). Importantly, treatment with ESK981 and/or trametinib did not affect mouse body weight throughout the treatment course (Extended Data Fig. 12d). In the end-point analysis, we did not observe any gross evidence of tumour burden in any mice treated with the combination of ESK981 and trametinib. To ensure that microscopic tumour burden was accounted for, tumours and pancreata were weighed together for each of the mice. On completing this analysis, we observed that mice given the combination treatment had substantially lighter pancreata (Extended Data Fig. 12e) than those in age-matched, non-tumour-bearing mice, and the individual treatments had more modest effects compared with vehicle-treated mice (Fig. 4d). Histopathological evaluation with H&E and CK19 corroborated this result, revealing no evidence of PDAC in seven of the eight mice treated with both ESK981 and trametinib, whereas either treatment alone exhibited only marginal effects (Fig. 4e,f and Extended Data Fig. 12f,g). Taken together, these data

illustrate that the combination therapy of a PIKfyve inhibitor and a MEK inhibitor eliminated tumour burden in an immunocompetent orthotopic PDAC model (Extended Data Fig. 12h).

This therapeutic strategy was next tested in a human PDAC model using the UM19 pCDX (Extended Data Fig. 12i). The combination treatment of ESK981 and trametinib induced substantial and durable regression in nearly all tumours, even when the tumours were able to adapt and outgrow ESK981 or trametinib therapy alone (Fig. 4g). At the end point, most of the tumours treated with the combination therapy were still smaller than their original size, with some being almost undetectable (Extended Data Fig. 12j). Ultimately, the combination treatment prevented any tumour from doubling throughout the duration of the experiment, whereas nearly all the tumours from the other treatment groups doubled or more in size (Fig. 4h). The combination of ESK981 and MRTX1133 was also tested in this model (Extended Data Fig. 12k), and it significantly improved the efficacy of either treatment alone throughout the treatment duration and at the end point (Extended Data Fig. 12l–m).

Given the promising results from the previous models, we next used the *Pdx1-Cre;LSL-Kras*[G12D/+]*;LSL-Trp53*[R172H/+] (KPC) autochthonous model to assess the efficacy of combining KRAS–MAPK inhibitors and PIKfyve inhibitors (Extended Data Fig. 12n–o). The combination of ESK981 and selumetinib (an MEK inhibitor) increased the median survival of these mice more than five-fold (58.5 days compared with 11 days) (Fig. 4i). Furthermore, 3D ultrasound imaging revealed that the combination therapy of ESK981 and selumetinib induced tumour regressions in almost every mouse (five out of six), whereas the vehicle and either treatment alone resulted in rapid, progressive tumour growth (Fig. 4j and Extended Data Fig. 12p). Notably, both statistics are similar to the most effective therapy reported in this model[44]. Moreover, although MRTX1133 was highly effective alone in the KPC model, ESK981 also enhanced its effectiveness of inducing tumour regressions and increasing survival (Extended Data Fig. 12q–s).

## Discussion

Targeting lysosomal function and the autophagic pathway as a therapeutic strategy has shown promise preclinically, given the known metabolic vulnerabilities of PDAC. However, HCQ, which is the only clinical-grade compound available to target these pathways, has had limited efficacy[21,22]. In this study, we nominated PIKfyve as a preeminent therapeutic target to disrupt PDAC lysosomal function. We also showed that PIKfyve and KRAS–MAPK have a bidirectional synthetic lethality relationship: first, PIKfyve function is required for

lysosome-dependent lipid homeostasis, and KRAS–MAPK signalling regulates de novo lipogenesis in PDAC cells; second, PIKfyve inhibition results in disruption of lysosomal lipid metabolism, obligating PDAC cells to upregulate and depend on de novo lipogenesis; and third, dual inhibition of PIKfyve and KRAS–MAPK drives PDAC into a metabolic crisis owing to its inability to obtain the lipids needed for cellular functions (Fig. 5). Given the rapidly evolving landscape of mutant KRAS[43,45], pan-(K)RAS[44,46] and MAPK pathway inhibitor development, along with the PIKfyve inhibitor ESK981, which is currently in a multicentre phase 2 clinical trial (NCT05988918), this highlights the combination of PIKfyve and KRAS–MAPK inhibitors as an extremely promising and rapidly translatable therapeutic strategy to treat PDAC.

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

## Methods

### Mouse strains

*Ptf1a-Cre, Ptf1a-Cre;lsl-Kras^{G12D}* and *Ptf1a-Cre;lsl-Kras^{G12D};Trp53^{R172H/+}* mice, which were used for the data in Figs. 1a,b,d–h and 2d–f and Extended Data Figs. 1g–k, 2a–1 and 3d,e, and *Pdx1-Cre;lsl-Kras^{G12D};p53^{R172H/+}* mice, which were used for Fig. 4i,j and Extended Data Fig. 12n–s, have been described previously[47–49]. Conditionally floxed *Pikfyve* (*Pikfyve^{f/f}*) mice were purchased from Jackson Laboratories. PCR genotyping was done for *Ptf1a-Cre* mice (for the *Ptf1a-Cre, Kras^{G12D}*, *Trp53^{R172H/+}* and *Pikfyve^{f/f}* alleles) from DNA isolated from mouse tails using standard methodology. PCR genotyping was done for *Pdx1-Cre* mice (for the *Pdx1-Cre, Kras^{G12D}* and *Trp53^{R172H/+}* alleles) from DNA isolated from an ear punch using standard methodology. Littermate controls were systematically used in all experiments, and the sex ratios for each cohort were balanced. The *Pdx1-Cre* animals used for the autochthonous model studies were bred and studied at the CRUK Scotland Institute as previously described[50]. All experiments involving the *Pdx1-Cre* animals were approved by the University of Glasgow Animal Welfare and Ethical Review Board and were performed under a UK Home Office licence. All other animals used in this study were housed at the University of Michigan in a pathogen-free environment, and all procedures involving these animals were performed in accordance with requirements of the University of Michigan Institutional Animal Care & Use Committee (IACUC). Mice were housed at a maximum of five mice per cage in a pathogen-free animal facility with a 12 h:12 h light:dark cycle, with 30–70% humidity and a temperature of 20–23 °C.

### Cell lines, antibodies and compounds

The cell lines PANC-1, MIA PaCa-2, Panc 04.03, SW1990, Panc 10.05 and HPAF-II were originally obtained from the American Type Culture Collection, and 7940B was provided by Greggory Beatty at Perlman School of Medicine at the University of Pennsylvania. The iKRAS 9805 cell line has previously been described[51]. The UM PDAC primary cell lines (UM2 and UM19) were obtained from surgically resected samples and established through mouse xenograft[52]. KPC1344 and KPC1361 were derived from a KPC mouse in-house by dissociating tumours manually with a sterile blade and then treating them with 1 mg ml$^{-1}$ collagenase II (ThermoFisher Scientific, 17101-015) and 1 mg ml$^{-1}$ DNase (Sigma-Aldrich, 10104159001) for 30 min with shaking at 37 °C. The cells were then strained using a MACS SmartStrainer (30 μm) (Miltenyi Biotec, 130-110-915) and rinsed with PBS before culturing. The KPC1361-sgNC, KPC1361-sgATG5 and KPC1361-sgATG7 monoclonal lines were generated previously[30]. All cells were grown and treated in DMEM (Gibco, 12430) + 10% FBS (ThermoFisher) unless otherwise indicated. All cell lines were genotyped to confirm their identity by Eurofins Genomics and tested biweekly for mycoplasma contamination. The sources of all antibodies and compounds are described in Supplementary Table 2. The synthesis and characterization methods used for PIK5-33d are described in Supplementary File 1.

### Histopathological analyses

The study pathologists conducted a detailed histopathological evaluation of mouse pancreatic tissues on 4-μm-thick H&E-stained formalin-fixed, paraffin-embedded sections. The examination involved checking all collected pancreas samples for the percentage prevalence of normal pancreas, PanIN, either high or low grade, and lesions with atypia or clear evidence of pancreatic ductal adenocarcinoma. The samples were then classified under these three categories and the results tabulated. Finally, the two pathologists reached a consensus to determine the final percentage prevalence.

### *PIKFYVE* RNAScope

RNA-ISH was performed using an RNAscope 2.5 HD Brown kit (Advanced Cell Diagnostics) and a target probe against *PIKFYVE* (1326631 Hs-*PIKFYVE*), according to the manufacturer's instructions. RNA quality was evaluated in each case using a positive control probe against human housekeeping peptidylprolyl isomerase B (PPIB; 313901). The assay background was monitored using a negative control probe against bacillus bacterial gene DapB (310043). Stained slides were evaluated under a light microscope at low- and high-power magnification for RNA-ISH signals in the cancer cells and normal pancreas by authors R.M. and J.H. The expression level was evaluated according to the RNAscope scoring criteria: a score of 0 means no staining or less than 1 dot per 10 cells; 1 means 1–3 dots per cell; 2 means 4–9 dots per cell, and no or very few dot clusters; 3 means 10–15 dots per cell and less than 10% of dots in clusters; 4 means more than 15 dots per cell and more than 10% of dots in clusters. The RNA-ISH score was calculated for each examined tissue section as the sum of the percentage of cells with scores of 0–4 [$(A\% \times 0) + (B\% \times 1) + (C\% \times 2) + (D\% \times 3) + (E\% \times 4)$, $A + B + C + D + E = 100$], using previously published scoring criteria[53].

### *Pikfyve* BaseScope

The BaseScope VS reagent kit (Advanced Cell Diagnostics, 323700), which identifies short targets and splice variants, was used to demonstrate *Pikfyve* on whole-mouse pancreatic tissues. The reagent kit was used with Discovery Ultra automated IHC/ISH slide staining systems by Ventana Medical Systems on a validated protocol utilizing BaseScope VS detection reagents (323710), RNAscope Universal VS sample preparation Reagents v.2 (PN323740) and RNAscope VS accessory kit (320630). BaseScope VS probe BA-Mm-Pikfyve-E6-3zz-st-C1, *Mus musculus* phosphoinositide kinase FYVE type zinc-finger-containing (PIKfyve) transcript variant 2 mRNA targeting exon 6 complimentary to the target mRNA was used (1300097-C1, accession code NM_011086.2, nucleotides 633–771) for the assay as a test probe. BaseScope VS positive control probe, Mm-PPIB-3ZZ, *M. musculus* PPIB mRNA (No701079) and BaseScope VS negative control probe DapB-3ZZ (701019) were used as positive and negative controls, respectively.

All slides were examined for positive signals in lesions and background benign pancreas by R.M. and J.H. The RNA in situ hybridization signal was identified as red punctate dots and the expression level was scored as follows: 0 means no staining or less than 1 dot per 10 cells (at ×40 magnification); 1 means 1 dot per cell (visible at ×20 or ×40); 2 means 2–3 dots per cell; 3 means 4–10 dots per cell (less than 10% in dot clusters) visible at ×20; and 4 means more than 10 dots per cell (more than 10% in dot clusters) visible at ×20. A cumulative RNA ISH product score (BaseScope score) was calculated for each available tissue core as the sum of the individual products of the expression level (0–4) and percentage of cells [0–100; that is, $(A\% \times 0) + (B\% \times 1) + (C\% \times 2) + (D\% \times 3) + (E\% \times 4)$; total range, 0–400].

### Immunohistochemistry

Immunohistochemistry was done on formalin-fixed, paraffin-embedded 4 μm sections of mouse or xenograft tissues. Slides were deparaffinized in xylene, followed by serial hydration steps in ethanol (100%, 95% and 70%) and water for 4 min each. Antigen retrieval was done by boiling slides in citrate buffer (pH 6). Endogenous tissue peroxidase activity was blocked by 3% $H_2O_2$ for 1 h. Slides were blocked in 10% goat serum for 1 h. The slides were then incubated in the primary antibodies. The specifics of the antibodies used are listed in Supplementary Table 2. Visualization of staining was done according to the manufacturer's protocol (Vector Laboratories, SK-4100). Following DAB staining, slides were dehydrated in ethanol (70%, 95% and 100%, 6 min each), xylene (15 min) and mounted using EcoMount (Thermo Fisher, EM897L).

After IHC staining, quantification was done using Fiji (Imagej)[54] (Extended Data Fig. 12g). Images were first subjected to colour deconvolution using the H DAB vector. Subsequently, a manual threshold was set on the basis of the uniform signal intensity of the DAB signal, serving as a cut-off for all images. The ratio of brown signal to total signal was calculated as the percentage CK19-positive area displayed

on the figure. Regions outside the pancreas, such as the spleen, were excluded from the analysis.

## In vivo tumour studies

All animal experiments were done in accordance with the Office of Laboratory Animal Welfare and approved by the University of Michigan IACUC, the CRUK Scotland Institute, or the University of Glasgow. No inclusion or exclusion criteria were used. Both male and female mice were used. Mice were randomized to treatment groups. Sample sizes were determined by preliminary studies and the level of observed effect. Study designs, sample sizes, outcome measures, statistical methods and results are stated in the relevant figure legends. Tumours were measured using digital calipers (two or three times per week) or ultrasound (for the autochthonous model, once a week) in a blinded manner. No experiment exceeded the end points predetermined by IACUC: subcutaneous tumours exceed 2 cm in any direction; tumour ulcerates more than half of its surface area; ulceration that has effusion, appears infected or has haemorrhage; or tumour develops in an area that impairs normal movement or physiological behaviour.

**Subcutaneous tumour studies.** For xenograft studies, CB17 severe combined immunodeficiency (SCID) mice 6–8 weeks old were obtained from the University of Michigan breeding colony. For syngeneic studies, C57BL/6J mice 6–8 weeks old were obtained from Jackson Laboratories. Subcutaneous tumours were established at both sides of the dorsal flank of the mice by injecting $1 \times 10^6$ cells in 100 µl of 50:50 Matrigel and serum-free medium. Tumours were measured 2–3 times per week using digital calipers following the formula $(\pi/6)(L \times W^2)$, where $L$ is the length and $W$ is the width of the tumour. At the end of the studies, the mice were killed and tumours were extracted and weighed.

**Pancreatic orthotopic tumour studies.** The 7940B and UM19 orthotopic models were established according to previously described protocols[13]. In brief, 50,000 (7940B) or 1,000,000 (UM19) cells were implanted directly into the pancreas of C57BL/6J mice (for 7940B) (Jackson Laboratories) or CB17 SCID (for UM19) mice. Tumours were established for 11 days before treatment with the indicated conditions. Mice were killed after three weeks of treatment, and tumours were weighed and preserved for further analyses.

**KPC autochthonous model.** The animals used for the autochthonous model studies were bred and were studied at the CRUK Scotland Institute, as previously described[50]. All experiments were approved by the University of Glasgow Animal Welfare and Ethical Review Board and were performed under a UK Home Office licence.

## In vivo apoptosis evaluation using TUNEL staining

TUNEL staining was performed with an in situ cell death detection kit (Roche Applied Science TMR Red, 12156792910) following the manufacturer's instructions. In brief, fixed sections were deparaffined, rehydrated and subsequently permeabilized using proteinase K. The labelling reaction was done at 37 °C for 1 h by addition of the reaction buffer containing the enzymes. Images were acquired using a Zeiss AxioImager M1 microscope. Quantification was performed using Fiji (ImageJ)[54] (Extended Data Fig. 3r). Signals from TUNEL and DAPI were quantified independently using the same manual threshold for all samples. The TUNEL percentage of positive scores was calculated as a percentage of the TUNEL signal divided by the DAPI signal.

## Immunoblots

Cell lysates were prepared in RIPA buffer (ThermoFisher Scientific) supplemented with Halt protease and phosphatase inhibitor cocktail (ThermoFisher Scientific). Total protein was measured by DC Protein Assay Kit II (BIO-RAD). An equal amount of protein was resolved in NuPAGE 3–8%, Tris-acetate protein gel (ThermoFisher Scientific) or NuPAGE 4–12% Bis-Tris protein gel (ThermoFisher Scientific), blocked with 5% non-fat dry milk in TBS-T and blotted with primary antibodies overnight. After incubation with HRP-conjugated secondary antibodies, membranes were imaged on an Odyssey Fc Imager (LiCOR Biosciences). For immunoblot experiments involving multiple targets overlapping in size, sample lysates were prepared in bulk and loaded on multiple gels as needed. Alternatively, membranes were stripped using Restore western blot stripping buffer (ThermoFisher Scientific) according to the manufacturer's instructions, rinsed, blocked and re-probed. One representative loading control for each experiment was displayed in the figures. For gel source data, see Supplementary Fig. 1.

## CETSA

CETSA was done according to previously described protocols[55]. In brief, 7940B cells were seeded overnight and subsequently treated with DMSO, ESK981 (1,000 nM) or apilimod (1,000 nM) for 2 h. The cells were then collected and used in single-cell suspensions of $1 \times 10^6$ cells in 50 µl PBS containing protease inhibitors. The suspensions were then heated and cooled (two cycles of 3-min heating followed by 3-min cooling at room temperature) using a thermal cycler. Cells were then lysed with three cycles of freeze–thawing in liquid nitrogen. Lysates were then centrifuged at 12,000$g$ for 10 min and the soluble fraction was isolated, denatured and resolved on a NuPAGE 4–12%, Bis-Tris protein gel (ThermoFisher Scientific), blocked with 5% non-fat dry milk in TBS-T and blotted with primary antibodies overnight. After incubation with HRP-conjugated secondary antibodies, membranes were imaged on an Odyssey Fc Imager (LiCOR Biosciences).

## Radioactive inositol labelling and measurement of phosphorylated phosphoinositide lipids

Radioactive phosphoinositide lipid labelling was done as previously described[56]. In brief, PANC-1 cells were grown in 100-mm dishes to 70–80% confluence. Cells were rinsed twice with $1 \times$ PBS, pH 7.4, and were incubated in inositol-free medium with 10 µCi ml$^{-1}$ [$^3$H]inositol for 24 h. The cells were then treated as indicated in Fig. 2b,c, Extended Data Fig. 3a–c or Extended Data Fig. 4i–m at a final concentration of 1 µM and incubated for the indicated time. Note that the small molecules were added into the labelling medium at the beginning of the labelling procedure for the 24-h samples. Following the treatments, myo-[2-$^3$H]inositol-labelled lipids were extracted as described[56] and the resultant glycerophosphoinositides were separated by ion exchange chromatography on a Partisphere 5 µm SAX cartridge column, 250 × 4.6 mm (WVS Hardware, 4621-1505, MAC-MOD Analytical). The raw counts for each peak are presented as a percentage of the total phosphatidylinositol, which is derived from the summation of counts across the six detectable glycero-inositol peaks (PtdIns, PtdIns3P, PtdIns4P, PtdIns5P, PtdIns(3,5)P$_2$ and PtdIns(4,5)P$_2$). Background scintillation counts, determined from adjacent regions, were subtracted from all peaks.

## Cell viability assays and synergy assays

Cells were plated into 96-well plates and incubated overnight at 37 °C in 5% CO$_2$. The next day, a serial dilution of the indicated compounds was prepared in culture medium and added to the plate. The cells were then further incubated for five days for experiments involving MRTX1133 or trametinib, or seven days for all other experiments. Subsequently, a CellTiter-Glo assay (Promega) was then done according to the manufacturer's instructions. The luminescence signal was acquired using an Infinite M1000 Pro plate reader (Tecan) and the data were analysed using GraphPad Prism 10 (GraphPad Software).

To determine the synergism of two different compounds using viability assays, cells were treated with the indicated combinations of the drugs for five days before we did the CellTiter-Glo assay as described above. These experiments were done with five biological replicates, each with ten wells of untreated internal controls for each plate used in

each experiment, which were used for normalization between plates. The data were then expressed as percentage inhibition relative to baseline, and the presence of synergy was determined by the Bliss method using the SynergyFinder+ web application[57].

### Autophagic flux probe generation and assay
Generation of the autophagic flux probe in 7940B, Panc 04.03 and iKRAS 8905 cells was done according to the original author's instructions[35]. In brief, cells were infected with pMRX-IP-GFP-LC3-RFP-LC3ΔG, which was a gift from Noboru Mizushima (Addgene, 84572). After puromycin selection, single-cell clones were expanded and genotyped to ensure the absence of homologous recombination between two LC3 fragments during retrovirus integration.

We seeded 15,000 cells in 96-well plates. After overnight incubation, cells were treated with the indicated compounds for 24 h. For assays assessing the co-treatment of autophagy inhibitors (apilimod, ESK981 or chloroquine) with autophagy inducers (torin-1, trametinib or MRTX1133), the autophagy inhibitor was added 4 h before the inducer. For assays using iKRAS 8905, cells were seeded with or without doxycycline, as indicated, and then treated with compounds in a similar fashion. Fluorescent signals were detected using the Infinite M1000 Pro plate reader (Tecan). Autophagy index was calculated by dividing the RFP signal by the GFP signal from each well, followed by normalization of all RFP/GFP ratios by the average RFP/GFP ratio of the DMSO condition.

### Confluence-based proliferation assays
Cells were seeded in a clear 96-well plate overnight before treatment. For treatment with the indicated compounds, plates were incubated in an Incucyte S3 2022 Rev1 (Sartorious), with ×10 images taken every 4 h, and confluence was analysed to assess for proliferation.

### Oxygen-consumption assays
The OCRs were determined using a Seahorse XF glycolytic rate assay (Agilent), according to the manufacturer's protocol. In brief, 15,000 (7940B) or 25,000 (Panc 04.03) cells were seeded in an Agilent XF96 cell culture microplate 16 h before treatment. Cells were treated with apilimod, ESK981, chloroquine or bafilomycin as indicated for 8 h. Immediately before the assay, cells were washed and then incubated in XF DMEM medium (pH 7.4, Agilent) with 1 mM pyruvate, 2 mM glutamine and 10 mM glucose. The assay was done on an XF96 extracellular flux analyser (Agilent), and the OCR was calculated using Seahorse Wave Controller software (v.2.6.3.5, Agilent). The OCR was normalized to cell number using the CyQUANT NF cell proliferation assay (Invitrogen) according to the manufacturer's instructions.

Real-time monitoring of basal OCR was done using a Resipher (Lucid Scientific). Next, 15,000 7940B cells were seeded in 50 µl of medium in a clear 96-well plate 16 h before treatment. Immediately after treatment with an additional 50 µl of medium (for a total of 100 µl), OCR monitoring was started by placing the Resipher device on the cells, which were incubated at 37 °C and 5% $CO_2$ for 24 h.

### Metabolic CRISPR screen
The human CRISPR metabolic gene knockout library was a gift from David Sabatini (Addgene, 110066)[58]. To achieve at least 1,000-fold coverage of the library for culturing, $75 \times 10^6$ MIA PaCa-2 cells were seeded at a density of $5 \times 10^5$ cells per ml in six-well plates containing 2 ml DMEM, 8 mg ml$^{-1}$ polybrene and the CRISPR screen library virus. Spin infection was done by centrifugation at 1,200$g$ for 45 min at 37 °C. After incubation for 24 h, the medium was replaced with fresh DMEM. After a subsequent 24-h incubation, cells were transferred to T-150 flasks (at a density of three wells into one T150 flask) containing 20 ml DMEM with puromycin at 2 mg ml$^{-1}$. After three days of selection, cells were seeded into 16 T-150 flasks at a density of $5 \times 10^6$ cells per flask in 20 ml DMEM containing either DMSO or 100 nM apilimod. Cells were

passaged every 3–4 days and reseeded back to the original cell density before collection on day 14. For the high-dose CRISPR screen, 2 µM apilimod was used and was refreshed every 3 days until cells became confluent at day 17 and were collected. We collected 15 million cells from each condition for isolation of genomic DNA using the DNeasy blood and tissue kit (Qiagen), according to the manufacturer's protocol.

For each condition, sgRNA was amplified from 50 mg genomic DNA using Herculase II fusion DNA polymerase (Agilent Technologies), column purified using Select-a-Size DNA Clean & Concentrator kit (Zymo Research) and then gel-purified using 6% Novex TBE gel (Thermo Fisher), followed by isolation from the gels with gel breaker tubes and gel filters (BioChain). The resulting PCR products then underwent end-repair and A-tail addition followed by New England Biolabs (NEB) adapter ligation. The final library was prepared by enriching adapter-ligated DNA fragments using 2 × KAPA HiFi HotStart mix and NEB dual-code barcode following the manufacturer's protocol. The libraries were then sequenced on an Illumina NovaSeq 6000 (paired-end 2 × 151 nucleotide read length).

Reads were trimmed to the bare sgRNA sequence using cutadapt 4.1 (ref. 59). Paired-end mates were trimmed separately using a sequence 5′-adjacent to the sgRNA position within the vector (TATATCTTGTG GAAAGGACGAAACACCG), requiring a minimum match of 18 bases to the sequence and followed by truncation to 20 bases (relevant cutadapt command parameters: *-m 18 -O 18 -l 20 --discard-untrimmed*). Trimmed reads were then combined and aligned using bowtie2 2.4.5 (ref. 60) to a reference built from each sgRNA in the library flanked by vector sequences (5′-GTTATCAACTTGAAAAAGTGGCACCG and 3′-CTAGATCTTGAGACAAATGGC). The bowtie2 parameter *--norc* was used to prevent reverse compliment alignment. Counting was then done using MAGeCK 0.5.9.5 (ref. 61). Supplementary Table 1 contains a summary of read counts. sgRNAs with fewer than 100 counts in the initial dataset were removed from downstream analysis. Genes targeted by fewer than six distinct sgRNAs following this filtering were likewise removed. Downstream analyses, including calculation of sgRNA depletion/enrichment scores, gene depletion/enrichment scores and selective dependency, were done according to previously described methods[62]. In brief, normalized sgRNA abundances were calculated by adding a pseudocount of one and then normalized to the total counts of each sample. The sgRNA enrichment/depletion scores were calculated as log$_2$ fold change between the final and initial populations, and the gene scores were calculated as the average log$_2$ fold change of the sgRNAs targeting that gene. To calculate selective essentiality scores, we first scaled gene scores using the medians of non-targeting sgRNAs and sgRNAs targeting core essential genes as references (0 and −1, respectively). Selective essential genes were then identified by taking the $z$-scored difference between the scaled apilimod and DMSO gene scores. Plots were generated using ggplot2 (v.3.4.4).

### RNA isolation and qPCR
Total RNA was isolated from cells using the miRNeasy kit (Qiagen) or DirectZol RNA Miniprep kits (Zymo), and cDNA was synthesized from 1,000 ng of total RNA using a Maxima First Strand cDNA synthesis kit for reverse transcription (RT)-qPCR (Thermo Fisher Scientific). RT-qPCR was done in triplicates using standard SYBR green reagents and protocols on a QuantStudio 5, 6 pro or 7 Real-Time PCR system (Applied Biosystems). The target mRNA expression was quantified using the ΔΔCt method and normalized to *ACTB* (human) or *Actb* (mouse) expression. Data presented represent technical triplicates. All primers were synthesized by Integrated DNA Technologies. Primer sequences are listed in Supplementary Table 2.

### RNA-seq and analysis
RNA-seq libraries were prepared using 800 ng total RNA. Ribosomal RNA was removed by enzymatic digestion of the specific probe-bound duplex rRNA and then fragmented to around 200–300 base pairs with

heat in fragmentation buffer (KAPA RNA Hyper+RiboErase HMR, Roche). Double-stranded cDNA was then synthesized by reverse transcription and underwent end-repair and ligation using New England Biolabs adapters. We did final library preparation by amplification with 2 × KAPA HiFi HotStart mix and NEB dual barcode. Library quality was measured on an Agilent 2100 Bioanalyzer (DNA 1000 chip) for concentration and product size. Paired-end libraries were sequenced using an Illumina NovaSeq 6000, (paired-end 2 × 151-nucleotide read length) with sequence coverage to 30 million to 40 million paired reads. Reads were demultiplexed using Illumina bcl2fastq conversion software (v.2.20). Transcripts were quantified by the alignment-free approach kallisto[63] using index generated from mouse reference genome (mm10) and then summed to obtain gene level counts. Raw transcripts per million values for each gene are shown in Supplementary Table 3. Differential analysis was done using limma-voom[64,65] after TMM normalization[66] of gene level counts with calcNormFactors of edgeR[67]. Genes with mean transcripts per million of less than 1 in both control and treatment groups were considered as low-expressed genes and excluded from differential analysis. Enrichment of the Hallmark and Reactome gene sets downloaded from MSigDB[68] were examined using fgsea[69] with genes ranked by logFC estimated from limma as input.

## Immunofluorescence

Cells were seeded overnight on an eight-chamber glass slide (CELL-TREAT) in 500 µl culture medium. The next day, cells were treated as indicated for each experiment and then fixed using 3.2% paraformaldehyde for 15 min, quenched with 125 mM L-glycine for 10 min and then rinsed twice with PBS. For LAMP1 immunofluorescence, samples were then permeabilized with 0.1% Triton-X 100 for 5 min, rinsed three times with PBS, blocked in 5% BSA for 1 h at room temperature and then incubated in LAMP1 primary antibody (Supplementary Table 2), 1:100 overnight at 4 °C. Samples were rinsed three times with PBS and then incubated in goat anti-rabbit secondary antibody (Alexa Fluor 594), 1:1,000, at room temperature for 1 h. For filipin immunofluorescence, cells were rinsed three times with PBS and incubated in 0.1 mg ml⁻¹ filipin complex for 2 h at room temperature. Samples were then rinsed three times in PBS and mounted with PBS and imaged on a Zeiss LSM900 confocal microscope (filipin, 405 nM channel; LAMP1, 568 nM channel; Airyscan mode) at ×63 magnification.

## Chromatin immunoprecipitation followed by sequencing

All the steps of the chromatin immunoprecipitation followed by sequencing (ChIP-seq) experiments were done as previously described[70] using the ideal ChIP-seq kit (Diagenode) with the following specifications. First, 5 million MIA PaCa-2 cells were treated as indicated for 8 h. The sonication cycle used was 30 s on followed by 30 s off, easy mode, for four cycles (Bioruptor, Diagenode) to achieve an average fragment size of 200 base pairs. We used 4 µg of c-MYC antibody for immunoprecipitation of fragmented chromatin. ChIP DNA was then de-crosslinked, purified and prepared for sequencing, as previously described[70].

Libraries were sequenced on a NovaSeq 6000, producing 150-base pair end reads. Reads were trimmed using Trimmomatic 0.39 (ref. 71) with options PE ILLUMINACLIP:TruSeq3-PE-2.fa:2:30:10. Trimmed paired reads were aligned to the human reference genome using bwa[72] with options -5 -S -P -T 0. The GRCh38/hg38 reference sequence was obtained from UCSC. Alignments were sorted and filtered for mapping quality >= 20 using samtools 1.9 (ref. 73). Read duplicates were removed using Picard MarkDuplicates 2.26.0 (http://broadinstitute.github.io/picard/). Non-primary alignments were removed using samtools view (option -F 0×900) and converted to BED format using bedtools bamtobed 2.27 (ref. 74). Peaks were called from these alignments using MACS2 2.2.7 (ref. 75) with default settings and the -B option to generate bedGraph coverage files. Peaks were then filtered using the ENCODE Unified GRCh38 exclusion list (https://www.encodeproject.org/files/ ENCFF356LFX/). Coverages captured by MACS2 were converted to bigWig using wigToBigWig 2.4 (ref. 76). FIMO 5.5.6 from the MEME[77] was used to find the JASPAR Myc motif MA0147.3 in called peaks[78]. Motif enrichment was done using findMotifsGenome.pl (HOMER 5.1 (ref. 79)) with options -size 200 -len 8 on filtered peaks (score >60).

## Generation of CRISPRi-mediated knockdown cell lines

The sgRNA sequences used were taken from a previously validated Perturb-seq library[80]. The sgRNAs were cloned into the backbone pLV hU6-sgRNA hUbC-dCas9-KRAB-T2a-Puro (Addgene: plasmid 71236; http://n2t.net/addgene:71236; RRID: Addgene_71236)[81] using the Golden Gate reaction. The generated plasmids were then expanded, verified by Sanger sequencing and packaged into lentiviruses by the University of Michigan Vector Core. Cells were seeded, infected with viruses along with polybrene (10 mg ml⁻¹) and then selected with puromycin (2 µg ml⁻¹ for MIA PaCa-2 and 5 µg ml⁻¹ for PANC-1) before further analysis. Given the notable impact of *PIKFYVE* and *FASN* knockdown on PDAC cells, new CRISPRi knockdown cell lines were generated before each experiment. The sgRNA sequences are listed in Supplementary Table 2.

## Generation of CRISPR-mediated gene knockout cell lines

The sgRNA sequences used are provided in Supplementary Table 2 and were inserted into the lentiCRIPSRv2 backbone[82] using the Golden Gate reaction, amplified and verified by Sanger sequencing. Plasmids were transfected into cells using Lipofectamine 3000 (ThermoFisher Scientific) according to manufacturer's protocol. Cells were then selected using puromycin as described above before further analysis. Of note, to knock out AMPKα, two sgRNAs targeting AMPKα1 (*PRKAA1*) and 2 sgRNAs targeting AMPKα2 (*PRKAA2*) were transfected together (625 ng each, totalling 2.5 µg of plasmid).

## Knockdown of *PRKAA1*/*PRKAA2* using siRNA

ON-TARGETplus Human SMARTpool siRNA (Horizon) targeting *PRKAA1* and *PRKAA2* or a non-targeting control were transfected into PANC1 cells using Lipofectamine RNAiMAX (ThermoFisher Scientific) at a concentration of 25 nM according to the manufacturer's protocol. One day after transfection, cells were trypsinized and re-seeded. After a subsequent overnight incubation, cells underwent a second round of transfection before further analysis.

## ESK981, trametinib, selumetinib and MRTX1133 formulation

ESK981 was added to ORA-PLUS and sonicated until completely dissolved. Trametinib was added to corn oil and sonicated until completely dissolved. Aliquots were frozen at −20 °C to prevent freeze–thaw cycles. Selumetinib was suspended in 0.5% hydroxypropyl methylcellulose (HPMC) + 0.1% Tween-80 in water and kept suspended by continuous stirring at 4 °C for up to one week. The MRTX1133 used for the subcutaneous study was added to 10% Captisol in 50 mM citrate (pH 5.0), sonicated until completely dissolved as previously described[43], and kept at 4 °C in the dark for a maximum of five days. The MRTX1133 used for the KPC autochthonous model was dissolved in hydroxypropyl-B-cyclodextrin (10% w/v in 50 mM citrate, pH 5.0) and kept with continuous stirring at 4 °C for up to one week. The ESK981, trametinib and selumetinib were delivered by oral gavage. The MRTX1133 was delivered by intraperitoneal injection.

## Targeted metabolomics

Polar metabolites from samples treated in biological triplicates were extracted using 80% (v/v) methanol/water and normalized using protein quantification from an additional sample from each condition. Equal estimated amounts of metabolites were dried using a SpeedVac vacuum concentrator, reconstituted in 50% (v/v) methanol in water, and analysed by liquid chromatography–tandem mass spectrometry (LC–MS/MS), as previously described[83]. Data were analysed as previously described[83] using Agilent MassHunter Workstation Quantitative

Analysis for QQQ v.10.1, build 10.1.733.0. However, metabolite abundance levels were not divided by the median levels across the samples. No post-detection normalization was done to avoid assuming linearity of signal. Raw values of each metabolite measured are provided in Supplementary Table 4. Heatmaps were generated using the Morpheus Matrix Visualization and analysis tool (https://software.broadinstitute.org/morpheus).

### Targeted lipidomics

Experiments with results presented in Extended Data Fig. 7e–h used the following methods.

**Sample preparation.** Samples for lipidomics analyses were prepared according to the automatic dual-metabolite/lipid sample-preparation workflow described in the Agilent application note 5994-5065EN. In brief, 1 million cells were washed in PBS and lysed with 1:1 trifluoroethanol:water at room temperature. Lysates were transferred to microcentrifuge tubes, incubated for 10 min and centrifuged at 250$g$ for 30 s. Samples were dried with a vacuum concentrator and resuspended in 1:1 trifluoroethanol:water. After transferring the samples to a 96-well plate, lipids were selectively isolated on a Bravo automated liquid handler platform (Agilent) operated by a VWorks automation protocol as described (5994-5065EN).

**LC–MS/MS analysis.** Samples were analysed on an Agilent 1290 Infinity II Bio LC ultra-high-performance liquid chromatography system with the Agilent Standardized Omics LC configuration, consisting of a high-pressure binary pump, multicolumn thermostat and a temperature-controlled multi-sampler. Samples were injected in randomized order on an Agilent 6495 C triple quadrupole mass spectrometer equipped with an Agilent Jet Stream Dual ESI ion source. Samples were analysed using the reverse-phase LC–MS/MS method reported in the Agilent application note 5994-3747EN. After acquisition, datasets were processed using MassHunter Quantitative Analysis 12.0 software and subsequently imported into Mass Profiler Professional (MPP) for chemometric analysis. No post-detection normalization was done to avoid assuming linearity of signal. Raw values of each lipid measured are provided in Supplementary Table 5.

Experiments with results presented in Extended Data Figs. 7i,j and 11k,l used the following methods.

After cell treatment as indicated, cells were pelleted by centrifugation at 1,000$g$ for 4 min at 4 °C. Next, 1 ml of chilled 10:3 methyl *tert*-butyl ether:methanol was added to the cell pellets, vortexed for 20 s and incubated at 4 °C for 5 min. Then 188 μl of MS-grade water was added and the sample was vortexed for 20 s. The samples were then centrifuged at 14,000$g$ at room temperature for 2 min, then 700 μl of the lipid fraction (top) was then isolated and moved to a separate tube. Using an independent biological replicate of all conditions, total protein was measured using the DC Protein Assay Kit II (BIO-RAD) as previously described. Using the protein quantity, the volumes of lipid extracts were normalized and then dried.

Dry lipid extracts were reconstructed with 20 μl of 9:1 methanol:chloroform, vortexed for 2 min and centrifuged for 10 min at 13,000 rpm at 20 °C to pellet insoluble material. Supernatants were transferred to analytical vials containing glass inserts and analysed by LC–MS/MS. Lipids were separated by reverse-phase C18 chromatography on an Agilent 1290 Infinity II BioLC with Agilent standard omics configuration coupled with an Agilent 6495D iFunnel triple quadrupole mass spectrometer. Details of the LC–MS/MS acquisition method are reported in the Agilent Application note 5994-3747EN. Samples were injected in randomized order and the raw data were processed using Agilent Mass Hunter Quantitative Analysis 12.0. Lipid signals were exported as CSV files for chemometric analysis. Raw values of each lipid measured are provided in Supplementary Table 5.

Changes in lipid class abundance in 7940B cells following treatment with apilimod (100 nM) or ESK981 (1,000 nM) relative to treatment with DMSO were estimated from linear mixed models with random intercepts to adjust for the baseline differences across the lipid classes. A separate model for each treatment (apilimod or ESK981) comparison against DMSO was built using the R package lme4 (v.1.1-35.1)[84].

### Stable isotope tracing

**Target compound confirmation.** C20 ceramide (d18:1/20:0) $C_{38}H_{57}NO_3$ (Cayman Chemicals) and C22 ceramide (d18:1/22:0) $C_{40}H_{79}NO_3$ (Cayman Chemicals) standards were dissolved in methanol at a concentration of 0.1 μg μl$^{-1}$ and were used to confirm the identity of each species. $(M + H)^+$, $(M\text{-}H_2O + H)^+$, $(M + Na)^+$, $(M + NH_4)^+$ and $(M\text{-}H)^-$ were used to confirm the identity of each lipid species with mass accuracies of 5 ppm tolerance.

**Samples preparation and data analysis.** After allowing cells to attach overnight, the culture medium was changed to DMEM − glucose with either 4.5 g l$^{-1}$ of U-$^{13}C_6$ isotopically labelled glucose or $^{12}C_6$ glucose, as indicated. Cells were also treated with PIKfyve inhibitors or DMSO as indicated. After 24 h, lipid extracts were isolated using methyl *tert*-butyl ether following the procedure described above. Protein lysate was extracted and quantified from a separate biological replicate of each condition for normalization. Samples were then normalized, dried and dissolved in 50 μl methanol. A 2 μl sample was then run on a Thermo Scientific IQX Orbitrap LC–MS system with an A YMC Accura Triart C8 (12 nm, 1.9 μm) 150 × 2.1 mm ID Column and a Phenomenex High Pressure column protection filter for separation. Skyline (v.24.1) was used to analyse the ion counts for the mass isotopologues for each species. Raw values of each lipid measured are provided in Supplementary Table 5.

### Statement on the use of human samples

Patient tissues from biopsies of pancreatic tumours were acquired from the University of Michigan pathology archives. These tissues were used for RNA Scope (RNA-ISH) experiments to assess for *PIKFYVE* expression in tumour or adjacent healthy pancreatic cells. The use of clinical formalin-fixed paraffin-embedded specimens from the archives was approved by the University of Michigan Institutional Review Board and did not require patient consent.

### Statistical analyses

No statistical methods were used to predetermine sample sizes. For all in vivo experiments, animals were randomly assigned into treatment cohorts. Tumour measurements were done by digital calipers or ultrasound (for the autochthonous model) in a blinded manner. For all in vitro experiments, cells were seeded from the same pool, so there was no requirement for randomization. All samples were analysed equally and simultaneously to eliminate bias. GraphPad Prism software (v.10) and R (v.4.3.2) were used for statistical calculations. Specific R packages used for individual analyses were included in their specific section in the Methods.

### Statistics and reproducibility

Figure 1. In **b**, the normal group had $n$ = 8 samples from individual animals and the lesion group had $n$ = 14 samples from individual animals; 8 animals were shared between the two groups. $P$-value, $3.6 \times 10^{-5}$. **c**, Biopsy samples were taken from five independent PDAC patients: two patients donated two samples each from distinct biopsies. Scores were determined as described above. **e**, $P$-values: KC *Pikfyve*$^{+/+}$ versus KC *Pikfyve*$^{f/+}$, $P = 7.4 \times 10^{-8}$; KC *Pikfyve*$^{+/+}$ versus KC *Pikfyve*$^{f/f}$, $P = 1.4 \times 10^{-9}$. **f**, $n$ = 15 individual KPC *Pikfyve*$^{+/+}$ animals; $n$ = 16 individual KPC *Pikfyve*$^{f/f}$ animals. **g**, $n$ = 15 individual KPC *Pikfyve*$^{+/+}$ animals; $n$ = 16 individual KPC *Pikfyve*$^{f/f}$ animals. **h**, These images are representative of $n$ = 15 KPC *Pikfyve*$^{+/+}$ and $n$ = 16 KPC *Pikfyve*$^{f/f}$.

Figure 2. **a**, This experiment was performed once. **b,c**, One biological replicate of each condition was analysed in two independent experiments for a total of $n$ = 2 for each group. **e**, Vehicle, $n$ = 11; ESK981, $n$ = 11; WT, $n$ = 8 animals. **f**, $n$ = 11 individual animals for both groups. **g**, The

data for the 'no tumour' group was also used as a reference in Fig. 4d ($n$ = 8 tumours for the vehicle and ESK981 groups, and $n$ = 6 for the 'no tumour' group, all from individual animals). $P$-values: vehicle versus ESK981, $7.4 \times 10^{-5}$; vehicle versus no tumour, $3.9 \times 10^{-6}$; ESK981 versus no tumour, 0.19. **i**, $n$ = 8 animals for each group. **j**, Vehicle, $n$ = 9; ESK981, $n$ = 9; no tumour, $n$ = 5 animals. **k**, These images are representative of $n$ = 9 for each group.

Figure 3. **f**, These images are representative of $n$ = 2 images.

Figure 4. **a**, These experiments were each performed independently twice with similar results. **b**, $n$ = 3 technical replicates for each group. These experiments were performed independently three times, each with similar results. $P$-values: *Fasn*: ESK981, +Dox versus −Dox, $1.8 \times 10^{-5}$; *Acaca*: DMSO, +Dox versus −Dox: $8.8 \times 10^{-5}$. **c**, This experiment was performed twice with similar results. **d**, $n$ = 9 individual animals for vehicle, trametinib and ESK981 groups; $n$ = 8 animals for the ESK981 + trametinib group; $n$ = 6 for the no tumour group. $P$-values for vehicle versus ESK981 + trametinib, $P = 8.4 \times 10^{-6}$. **e**, $n$ = 9 animals for vehicle, trametinib and ESK981 groups; $n$ = 8 individual animals for the ESK981 + trametinib group. **g**, $n$ = 16 tumours from 8 animals for the vehicle group; $n$ = 14 tumours from 7 animals for the trametinib, ESK981 and ESK981 + trametinib groups. $P$-value: less than $1 \times 10^{-15}$. **h**, $P$-values: ESK981 + trametinib versus vehicle, $7.15 \times 10^{-175}$; ESK981 + trametinib versus trametinib, $6.7 \times 10^{-137}$; ESK981 + trametinib versus ESK981, $2.8 \times 10^{-54}$. **i**, $n$ = 5 animals for the vehicle group; $n$ = 8 animals for the selumetinib group; $n$ = 6 for ESK981 and ESK981 + selumetinib groups. **j**, $n$ = 5 individual animals for the vehicle group; $n$ = 8 animals for the selumetinib group; $n$ = 6 for the ESK981 and ESK981 + selumetinib groups.

## Reporting summary

Further information on research design is available in the Nature Portfolio Reporting Summary linked to this article.

## Data availability

All data and raw gel images are included with the paper. All materials are available from the authors on reasonable request. All raw next-generation sequencing data, such as DNA sequencing for the CRISPR screen and ChIP-seq or RNA-seq, have been deposited in the Gene Expression Omnibus repository at NCBI with the accession numbers GSE255378 and GSE277832. Processed sequencing data, such as sgRNA counts and RNA-seq (in transcripts per million, TPM), are included as Supplementary Tables 1 and 3, respectively. Raw data for metabolomics and lipidomics experiments are included as Supplementary Tables 4 and 5, respectively). Other publicly available datasets used were human genome assembly, GRCh38; mouse genome assembly, GRCm38; GSE201412 (ref. 43). Source data are provided with this paper.

## Code availability

No custom codes were developed for this study.

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

**Acknowledgements** We thank R. Pakkan, A. Yin, N. H. Kim, A. K. Körkaya, S. Yee, X. Jiang, B. Jackson, H. Zheng, D. Sutton, N. Das, S. Simko and A. Delekta for technical assistance; E. J. Bristow for help with phosphoinositide lipid measurements; F. Su and R. Wang for generating sequencing libraries; L. McMurry, A. Miller, C. Caldwell-Smith, Y. Cheng and S. Li for processing histological samples; J. C.-Y. Tien for help with coordinating genetically engineered mouse model breeding colonies; P. Sajjakulnkit, H. Wong and A. Andren for help with metabolomics and lipidomics experiments; D. Klionsky, E. Young, S. Kang, B. Chen, A. M. Elhossiny, M. Radyk, S. A. Kerk, L. Lin and J. E. Choi for experimental and analytical guidance; D. Frigo, J. Xu, A. Aguirre and S. Chugh for help with procuring reagents; Agilent Technologies for help with lipidomics experiments and chemometrics data analysis; S. Miner for manuscript editing and preparation; and members of the Chinnaiyan and Lyssiotis laboratories and the staff of the Rogel & Blondy Pancreatic Cancer Center, University of Michigan, for support and discussions. This work was supported as follows: a National Cancer Institute (NCI) Outstanding Investigator Award R35-CA231996 (to A.M.C.); NCI Early Detection Research Network U2C-CA271854 (A.M.C.); NCI P30-CA046592, National Institute of Neurological Diseases and Stroke R01 1NS129198 (L.S.W.); Department of Defense Idea Development Award HT9425-23-1-0084 (Y.Q.); CRUK core funding A17196 and A31287 (to the CRUK Scotland Institute) and A29996 (J.P.M.); a CRUK Precision Panc grant (A25233); the CRUK Scotland Centre (CTRQQR-2021\100006) (J.P.M.). C.A.L. was supported by the NCI (R37-CA237421, R01-CA248160 and R01-CA244931). C.A.L. and M.P.d.M. were supported by NIH grants U54 CA274371 and U01 CA274154. Y.M.S. was supported by the NCI (R01CA148828

and R01CA245546). N.J.R. was supported by an NIH T32 training grant (T32 GM150581). C.C. was supported by an NCI F30 fellowship (F30CA288093) and NIH T32 training grants (CMB 5T32-GM145470 and MSTP T32GM00786). A.M.C. is a Howard Hughes Medical Institute investigator, an A. Alfred Taubman scholar and an American Cancer Society professor.

**Author contributions** C.C., Y.Q., C.A.L. and A.M.C. conceived the study, designed the experiments and wrote the manuscript. Y.Q., C.A.L. and A.M.C. supervised the project. C.C., J.P.W., S.P., Y. Zheng, L.X., D.A., Y.B., and S.M. did in vitro experiments. C.C. and Y.Q. did all the in vivo experiments with help from T.H. and Y. Zheng. P.M. did lipidomics experiments and analysis with help from R.B. Bioinformatics analyses were done by C.C., R.B., N.J.R., B.M. and Y. Zhang. J.H., R. Mannan and R. Mehra did all the histopathological evaluations and did the immunohistochemistry with help from C.C. and Y.Q. X.C. generated next-generation sequencing libraries and did the sequencing. C.L., Z.W. and K.D. developed the PIKfyve degrader. S.A.K., D.J.M. and J.P.M. did the KPC autochthonous model efficacy studies. H.K. and L.S.W. determined the levels of the phosphorylated phosphoinositide lipids. L.Z. and C.C. did the ceramide stable-isotope tracing and data analysis. V.S., M.P.d.M., and Y.M.S. helped with manuscript writing and organization. M.P.d.M., Y.M.S., C.A.L. and A.M.C. provided resources and funding.

**Competing interests** A.M.C. is a co-founder and serves on the scientific advisory board of Esanik Therapeutics, which owns proprietary rights to the clinical development of ESK981. Esanik Therapeutics did not fund or approve the conducting of this study. A.M.C. is a co-founder and serves on the scientific advisory board of Medsyn Bio, Lynx Dx and Flamingo Therapeutics. A.M.C. serves as an advisor to Tempus, Proteovant, Aurigene Oncology, RAPTTA Therapeutics and Ascentage Pharmaceuticals. A.M.C., Y.Q., C.A.L., C.C., Y.B., K.D., Z.W. and C.L. are listed as inventors on the following patents pertaining to the development of methodologies and compounds targeting PIKfyve in diseases: PCT: PCT/US2021/057022 (A.M.C. and Y.Q.); PCT: PCT/US2024/017088 (A.M.C. and Y.Q.); PCT: PCT/CN2024/087809 (A.M.C., Y.Q., Z.W., K.D. and C.L.), US patent no: 63/537,996 (A.M.C., Y.Q. and Y.B.), US patent no: PCT/CN2024/078381 (C.A.L., A.M.C., K.D., Y.Q., Z.W. C.L., and C.C.). In the past three years, C.A.L. has consulted for Astellas Pharmaceuticals, Odyssey Therapeutics, Third Rock Ventures and T-Knife Therapeutics, and he is an inventor on patents pertaining to KRAS-regulated metabolic pathways, redox control pathways in pancreatic cancer and targeting the GOT1-ME1 pathway as a therapeutic approach (US patent no. 2015126580-A1, 2015; US patent no. 20190136238, 2019; and international patent no. WO2013177426-A2, 2015). The remaining authors declare no competing interests.

**Additional information**
**Correspondence and requests for materials** should be addressed to Yuanyuan Qiao, Costas A. Lyssiotis or Arul M. Chinnaiyan.

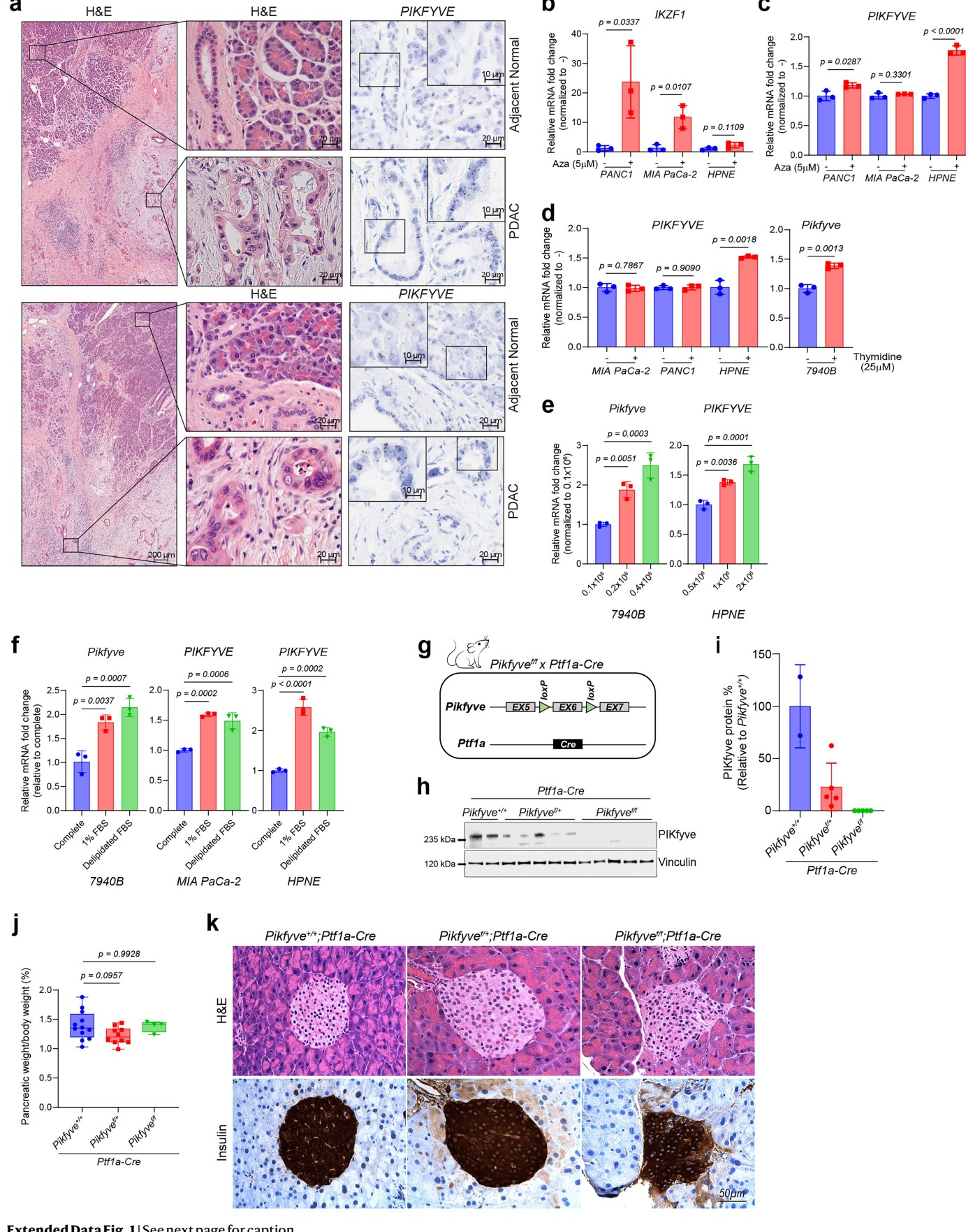

**Extended Data Fig. 1** | See next page for caption.

**Extended Data Fig. 1 | PIKfyve is overexpressed in PDAC, regulated by nutrient availability, and dispensable for normal pancreatic development and function. a**. Representative image of two human PDAC patient samples showing H&E (left and middle) or *PIKFYVE* RNA-ISH (right). Scalebars are 200 µm (left), 20 µm (middle), 20 µm (right, low magnification), and 10 µm (right inset, high magnification). **b-c**. Quantitative PCR (qPCR) showing changes of *IKZF1* (b) and *PIKFYVE* (c) in PANC1 (PDAC), MIA PaCa-2 (PDAC), and HPNE (hTERT-immortalized pancreatic epithelial) cells, treated with 5-aza-2-cytidine (5 µM) for 5 days, refreshed on day 3. CT levels were normalized first to β-actin and then to the "-" conditions for each cell line which were not treated with 5-aza-2-cytidine. Bars are +/-SD. (Unpaired two-tailed t-tests.). **d**. qPCR showing changes in *PIKFYVE* in MIA PaCa-2, PANC1, HPNE, or 7940B (murine KPC cell line) upon treatment with thymidine (25µM) for approximately 1 cell doubling time (48 hours for MIA PaCa-2, PANC1, and HPNE, and 24 hours for 7940B). CT values were normalized first to β-actin and then to the "-" conditions for each cell line which were not treated with thymidine. Bars are +/-SD. (Unpaired two-tailed t-tests.). **e**. qPCR showing changes in *PIKFYVE* in 7940B or HPNE cells upon overnight incubation after being seeded at different cell densities. CT values were normalized first to β-actin and then to the condition with the lowest cell density. Bars are +/-SD. (One way ANOVA with Dunnett's.) **f**. qPCR showing changes in *PIKFYVE* in 7940B, MIA PaCa-2, or HPNE cells upon 24-hour incubation with the indicated media conditions. Complete= DMEM + 10% fetal bovine serum (FBS) + 1% penicillin/streptomycin (P/S), 1% FBS = DMEM + 1% FBS + 1% P/S, Delipidated FBS = DMEM + 10% lipid-depleted FBS + 1% P/S. Bars are +/-SD. (One way ANOVA with Dunnett's.). **g**. Breeding design for the generation of *Pikfyve* specific deletion in *Ptf1a-Cre* mice. Mouse cartoon was adapted from Adobe Stock Image (Asset #304271210). **h**. Immunoblot analysis of pancreatic tissue from 12-week-old *Ptf1a-Cre; Pikfyve*^+/+^, *Ptf1a-Cre*; *Pikfyve*^f/+^, and *Ptf1a-Cre*; *Pikfyve*^f/f^ mice showing changes in PIKfyve protein levels. Vinculin was used as a loading control. **i**. Densitometry analyses of immunoblot displayed in **(h.)** PIKfyve protein % was calculated by dividing the densitometry values for each PIKfyve band by the average value from the *Pikfyve*^+/+^ group. Bars are +/-SD. (One-way ANOVA with Dunnett's.). **j**. Pancreas tissue weight normalized to total body weight for *Ptf1a-Cre;Pikfyve*^+/+^, *Ptf1a-Cre;Pikfyve*^f/+^, and *Ptf1a-Cre;Pikfyve*^f/f^ mice. Box represents 25th and 75th percentiles; whiskers represent the range. (One-way ANOVA with Dunnett's.). **k**. Representative images of H&E and insulin IHC staining from the pancreas tissue of *Ptf1a-Cre;Pikfyve*^+/+^, *Ptf1a-Cre;Pikfyve*^f/+^, and *Ptf1a-Cre;Pikfyve*^f/f^ mice. Scalebar = 50 µm. **Statistics and reproducibility: b**. n = 3 technical replicates each condition per cell line. **c**. n = 3 technical replicates each condition per cell line. **d**. n = 3 technical replicates each condition per cell line. **e**. n = 3 technical replicates each condition per cell line. **f**. n = 3 technical replicates each condition per cell line. P-value: complete vs Delipidated FBS for HPNE:1.4e-5. **i**. Pikfyve + /+ n = 2, Pikfyvef/+ n = 5, Pikfyvef/f n = 5 individual animals). **j**. *Ptf1a-Cre;Pikfyve*^+/+^ n = 12, *Ptf1a-Cre;Pikfyve*^f/+^ n = 10, *Ptf1a-Cre;Pikfyve*^f/f^ n = 4 individual animals.

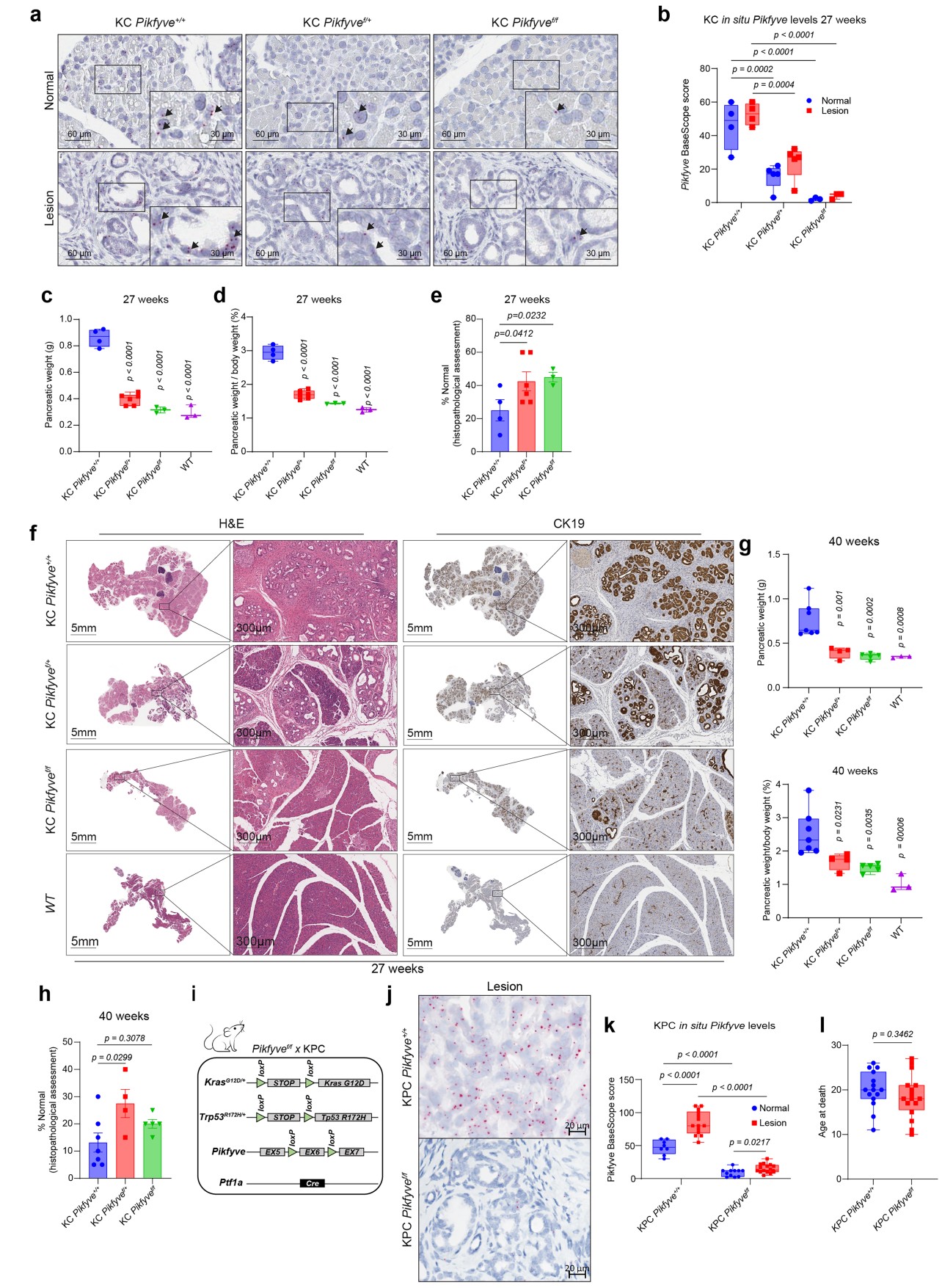

**Extended Data Fig. 2** | See next page for caption.

**Extended Data Fig. 2 | PIKfyve is critical for progression of precursor PanIN lesions to PDAC. a**. Representative images of PIKfyve BaseScope staining from pancreas tissue of 27-week-old KC *Pikfyve*$^{+/+}$, KC *Pikfyve*$^{f/+}$, and KC *Pikfyve*$^{f/f}$ mice. Scalebar = 60 μm for zoomed-out images; 30 μm for zoomed-in images. **b**. *Pikfyve* levels as determined by BaseScope of KC *Pikfyve*$^{+/+}$, KC *Pikfyve*$^{f/+}$, and KC *Pikfyve*$^{f/f}$ murine pancreas tissue separated by normal and lesional areas. Box represents 25th and 75th percentiles; whiskers represent the range. (One way ANOVA with Dunnett's.). **c**. Pancreas tissue weight from 27-week-old KC *Pikfyve*$^{+/+}$, KC *Pikfyve*$^{f/+}$, KC *Pikfyve*$^{f/f}$, and age-matched wild-type (WT) mice. Box represents 25th and 75th percentiles; whiskers represent the range. (One-way ANOVA with Dunnett's.). **d**. Pancreas tissue weight normalized to total body weight from KC *Pikfyve*$^{+/+}$, KC *Pikfyve*$^{f/+}$, KC *Pikfyve*$^{f/f}$ or age-matched wild-type (WT) mice at 27 weeks of age. Box represents 25th and 75th percentiles; whiskers represent the range. (One-way ANOVA with Dunnett's.). **e**. Percentage of pancreas occupied by normal tissue as determined by histological analyses in KC *Pikfyve*$^{+/+}$, KC *Pikfyve*$^{f/+}$, and KC *Pikfyve*$^{f/f}$ mice at 27 weeks of age. Bars are +/- SEM. (One-way ANOVA with Dunnett's.). **f**. Representative histological images showing H&E and CK19 staining on pancreatic tissue of KC *Pikfyve*$^{+/+}$, KC *Pikfyve*$^{f/+}$, and KC *Pikfyve*$^{f/f}$ mice at 27 weeks of age. Scalebar = 5 mm for the whole-pancreas images, 300 μm for the zoomed-in images. **g**. Raw pancreas tissue weight (top) and pancreatic weight normalized to total body weight (bottom) from 40-week-old KC *Pikfyve*$^{+/+}$, KC *Pikfyve*$^{f/+}$, and KC *Pikfyve*$^{f/f}$, and age-matched wild-type (WT) mice. Box represents 25th and 75th percentiles; whiskers represent the range. (One-way ANOVA with Dunnett's.). **h**. Percentage of pancreas occupied by normal tissue as determined by histological analyses of KC *Pikfyve*$^{+/+}$, KC *Pikfyve*$^{f/+}$, and KC *Pikfyve*$^{f/f}$ mice at 40 weeks of age. Box represents 25th and 75th percentiles; whiskers represent the range. (One-way ANOVA with Dunnett's.). **i**. Breeding design for the generation of KPC *Pikfyve*$^{+/+}$ and KPC *Pikfyve*$^{f/f}$ mice. Mouse cartoon was adapted from Adobe Stock Image (Asset #304271210). **j**. Representative images of PIKfyve BaseScope staining from pancreas tissue of 25-week-old KPC *PIKfyve*$^{+/+}$ and KPC *PIKfyve*$^{f/f}$ mice.

Scalebar = 20 μm. **k**. In situ *Pikfyve* levels in KPC *Pikfyve*$^{+/+}$ and KPC *Pikfyve*$^{f/f}$ murine pancreas lesion vs normal tissue as determined by BaseScope. The KPC *Pikfyve*$^{+/+}$ scores used as a reference are the same as those used in Fig. 1b. The two cohorts were stained and analyzed in the same batch. Box represents 25th and 75th percentiles; whiskers represent the range. (Multiple unpaired two-tailed t-test.). **l**. The age at death of mice in KPC *PIKfyve*$^{+/+}$ and KPC *PIKfyve*$^{f/f}$ cohorts that were analyzed in Fig. 1f,g. Box represents 25th and 75th percentiles; whiskers represent the range. (Unpaired two-tailed t-test.). **Statistics and reproducibility: b**. n = 4 sections taken from individual KC *Pikfyve*$^{+/+}$ animals analyzed for both normal and lesional areas; n = 5 sections taken from individual KC *Pikfyve*$^{f/+}$ animals analyzed for both normal and lesional areas; n = 3 KC *Pikfyve*$^{f/f}$ sections taken from individual animals analyzed for normal and lesional areas. P-values: KC *Pikfyve*$^{+/+}$ normal vs KC *Pikfyve*$^{f/f}$ normal:1.1e-5; KC *Pikfyve*$^{+/+}$ lesion vs KC *Pikfyve*$^{f/f}$ lesion:3.1e-6. **c**. KC *Pikfyve*$^{+/+}$ n = 4, *KC Pikfyve*$^{f/+}$ n = 6, KC *Pikfyve*$^{f/f}$ n = 3, WT n = 3 individual animals. P-values: KC *Pikfyve*$^{+/+}$ vs KC *Pikfyve*$^{f/+}$:1.4e-8; KC *Pikfyve*$^{+/+}$ vs KC *Pikfyve*$^{f/f}$:1.5e-8; KC *Pikfyve*$^{+/+}$ vs WT:1.0e-8. **d**. KC *Pikfyve*$^{+/+}$ n = 4, KC *Pikfyve*$^{f/+}$ n = 6, KC *Pikfyve*$^{f/f}$ n = 3, WT n = 3 individual animals. P-values: KC *Pikfyve*$^{+/+}$ vs KC *Pikfyve*$^{f/+}$:2.1e-8; KC *Pikfyve*$^{+/+}$ vs KC *Pikfyve*$^{f/f}$:1.6e-8; KC *Pikfyve*$^{+/+}$ vs WT:4.2e-9. **e**. KC *Pikfyve*$^{+/+}$ n = 4, KC *Pikfyve*$^{f/+}$ n = 6, KC *Pikfyve*$^{f/f}$ n = 3, WT n = 3 individual animals). **f**. These images are representative of n = 4 KC *Pikfyve*$^{+/+}$; n = 6 KC *Pikfyve*$^{f/+}$, n = 3 KC *Pikfyve*$^{f/f}$; n = 3 WT. **g**. KC *Pikfyve*$^{+/+}$ n = 7, *KC Pikfyve*$^{f/+}$ n = 4, KC *Pikfyve*$^{f/f}$ n = 5, WT = 3 individual animals. **h**. KC *Pikfyve*$^{+/+}$ n = 7, *KC Pikfyve*$^{f/+}$ n = 4, KC *Pikfyve*$^{f/f}$ n = 5, WT = 3 individual animals. **k**. The KPC normal had n = 8 individual animals; the KPC lesion n = 14 individual animals; 8 animals were shared between these two groups. The KPC *Pikfyve*$^{f/f}$ normal group had n = 11 individual animals, and the KPC *Pikfyve*$^{f/f}$ lesion n = 16 individual animals; 11 animals were shared between these two groups. P-values: KPC *Pikfyve*$^{+/+}$ normal vs KPC *Pikfyve*$^{+/+}$ lesion:3.6e-5; KPC *Pikfyve*$^{+/+}$ normal vs KPC *Pikfyve*$^{f/f}$ normal:1.9e-8; KPC *Pikfyve*$^{+/+}$ lesion vs KPC *Pikfyve*$^{f/f}$ lesion:1.6e-14. **l**. KPC *PIKfyve*$^{+/+}$ n = 15, KPC *PIKfyve*$^{f/f}$ n = 16 individual animals.

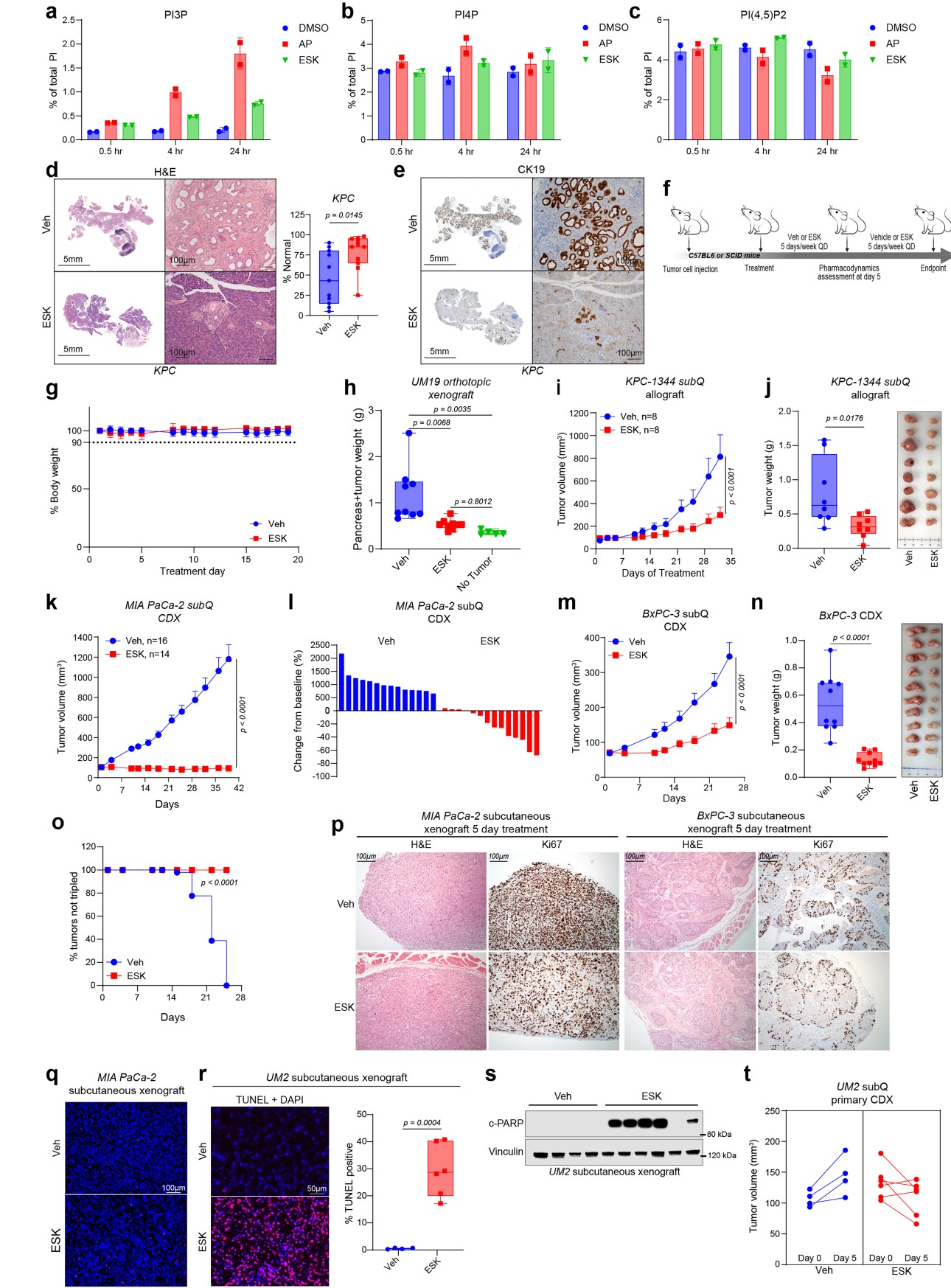

**Extended Data Fig. 3 |** See next page for caption.

**Extended Data Fig. 3 | Pharmacological inhibition of PIKfyve blocks pancreatic cancer progression and tumor growth. a-c.** Relative abundance of PI3P (**a.**), PI4P (**b.**), and PI(4,5)P2 (**c.**) in PANC1 cells upon treatment with apilimod (1000 nM) or ESK981 (1000 nM) for the indicated times. Bars are +/-SD. **d.** Representative H&E staining of whole pancreatic tissue from vehicle and ESK981 treated mice (left). Scalebar = 5 mm for the whole-pancreas images, 100 µm for the zoomed-in images. Quantification of histologically normal pancreatic tissue in vehicle or ESK981 treated mice (right). Box represents 25th and 75th percentiles; whiskers represent the range. (Unpaired two-tailed t-test.). **e.** Representative CK19 IHC staining of whole pancreatic tissue from vehicle or ESK981 treated mice (left). Scalebar = 5 mm for the whole-pancreas images, 100 µm for the zoomed-in images. **f.** Schematic of in vivo efficacy studies utilizing cell-derived xenograft (CDX) or allograft models. ESK981 was dosed at 30 mg/kg per day for 5 days/week (PO) in all studies. Mouse cartoons were adapted from Adobe Stock Image (Asset #304271210). **g.** Relative body weight of C57BL/6 mice used in Fig. 2g–i measured on the indicated treatment day. Bars are +/-SEM. **h.** Endpoint raw pancreas + tumor weight of CB17 SCID mice bearing UM19 orthotopic tumors. Pancreata of 5 age-matched non-tumor bearing CB17 SCID mice were used as references. Box represents 25th and 75th percentiles; whiskers represent the range. (One-way ANOVA with Tukey's.). **i.** Tumor volumes of subcutaneous allograft model using KPC-derived KPC-1344 cells in response to vehicle or ESK981 in C57BL/6 mice. Bars are +SEM. (Two-way ANOVA.). **j.** Tumor weights (left) and images (right) of KPC-1344 model tumors at study endpoint. Box represents 25th and 75th percentiles; whiskers represent the range. (Unpaired two-tailed t-test.). **k.** Tumor volumes of subcutaneous CDX model using MIA PaCa-2 cells in response to vehicle or ESK981 in SCID mice. Bars are +SEM. (Two-way ANOVA.). **l.** Waterfall plot displaying changes in tumor volume comparing endpoint to baseline in response to vehicle or ESK981 treatment in MIA PaCa-2 CDX model. **m.** Tumor volumes of subcutaneous CDX model derived from BxPC-3 cells in response to vehicle or ESK981 in SCID mice. Data plotted are mean tumor volumes +SEM. (Two-way ANOVA.). **n.** Individual weights (left) and images (right) of tumors from CDX model derived from BxPC-3 cells at endpoint. Box represents 25th and 75th percentiles; whiskers represent the range. (Unpaired two-tailed t-test.). **o.** Kaplan–Meier estimates of time to tumor tripling of BxPC-3 CDX tumors after vehicle or ESK981 treatment. (Two-sided log-rank test.). **p.** Representative images of H&E and Ki67 IHC staining in MIA-PaCa-2 (left) and BxPC-3 (right) CDX models post vehicle or ESK981 treatment. **q.** Representative image of TUNEL staining from MDA-PaCa-2 CDX tumors after 5 days of treatment of vehicle or ESK981. **r.** Left panel, representative images of TUNEL staining from primary UM-2 CDX tumors after 5 days of treatment of vehicle or ESK981. Scalebar = 50 µm. Right panel, quantification of TUNEL positivity in indicated groups. Box represents 25th and 75th percentiles; whiskers represent the range. (Unpaired two-tailed t-test.). **s.** Immunoblot analysis of primary UM2 CDX tumors after 5 days treatment of vehicle or ESK981 showing changes in apoptosis marker cleaved PARP (c-PARP). Vinculin was used as a loading control. **t.** Individual tumor volumes of a PDAC primary CDX UM-2 model before and after 5 days treatment of vehicle or ESK981. **Statistics and reproducibility: a-c.** One biological replicate of each condition was analyzed together, in two independent experiments for a total of n = 2 for each group. **d.** n = 11 individual animals for both groups. **e.** This image is representative of n = 11 individual animals for both groups. **g.** n = 8 individual animals for each group. **h.** vehicle n = 9, ESK981 n = 9, No Tumor n = 5 individual animals. **i.** n = 8 from 4 mice for each cohort, P-value: 4.4e-6. **j.** n = 8 tumors from 4 mice for each group. **k.** n = 16 from 8 mice for vehicle, n = 14 from 7 mice for ESK981. P-value: 5.8e-157. n = 14 from 7 mice for each group as 1 mouse in the vehicle reached humane endpoint prior to endpoint analysis. **m.** n = 12 tumors from 6 individual animals for vehicle, n = 10 tumors from 5 individual animals for ESK981. P-value: 5.7e-32. **n.** n = 10 tumors from 5 animals for each cohort as 1 mouse in the vehicle cohort reached humane endpoint prior to endpoint analysis. **o.** n = 12 tumors from 6 individual animals for vehicle, n = 10 tumors from 5 individual animals for ESK981. P-value = 1.7e-28. **p.** These images are representative of images from n = 4 tumors from individual animals for MIA PaCa-2 vehicle and ESK981 groups, n = 4 for BxPC-3 vehicle group, and n = 6 BxPC-3 ESK981 group. **q.** These images are representative of three. **r.** n = 4 (vehicle) or n = 6 (ESK981) individual tumors (from n = 2 or n = 3 animals, respectively) and each represent the mean of 5 representative images per tumor. **s.** This experiment was performed once. **t.** Vehicle n = 4 tumors from 2 animals; ESK981 n = 6 tumors from 3 animals.

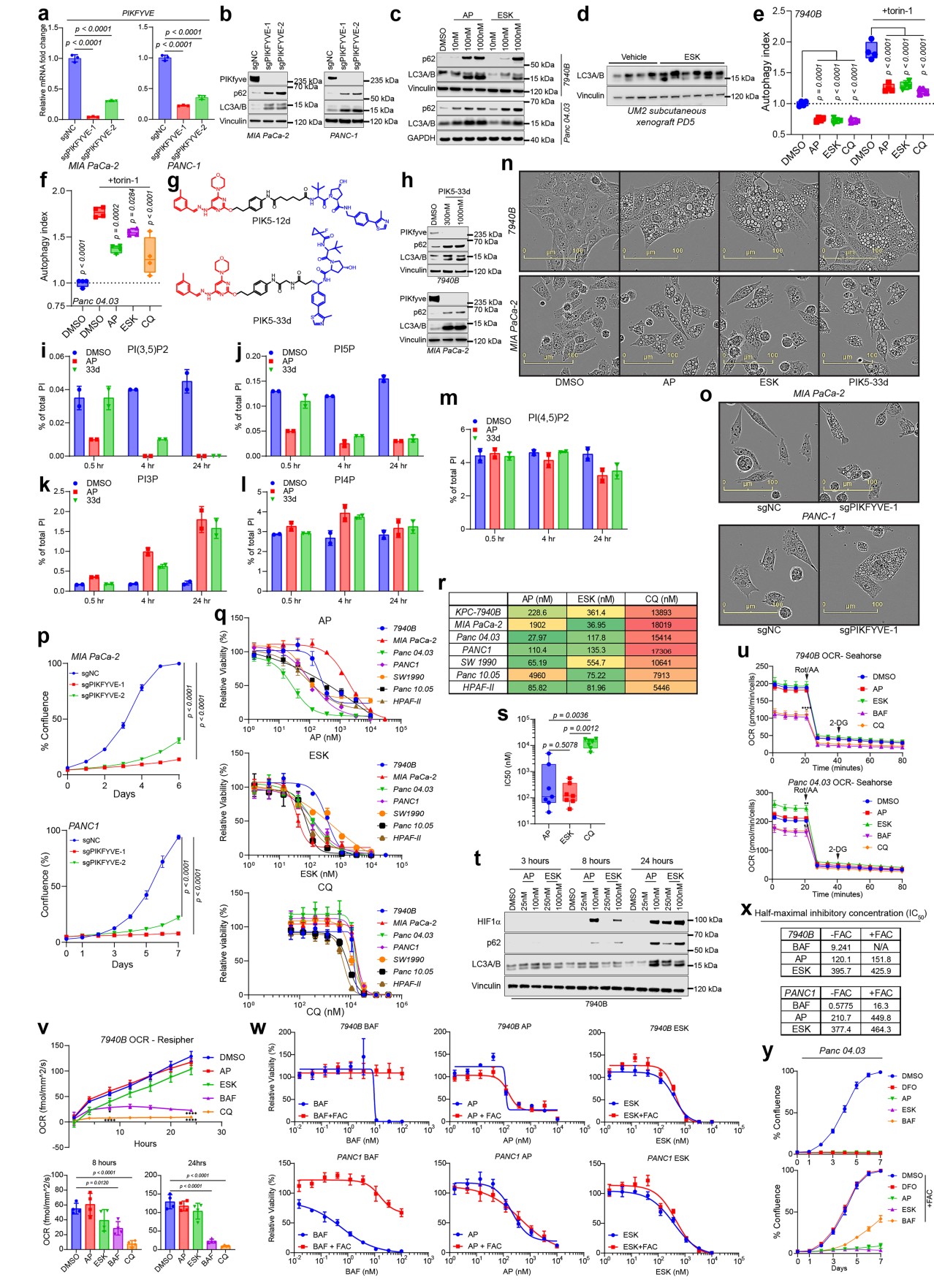

**Extended Data Fig. 4** | See next page for caption.

**Extended Data Fig. 4 | Perturbation of PIKfyve disrupts autophagy, induces lysosomal vacuolization, and decreases PDAC cell growth through an iron-independent mechanism. a.** qPCR of MIA PaCa-2 (left) or PANC-1 (right) cells upon CRISPRi-mediated knockdown of *PIKFYVE* with two independent sgRNAs (sgPIKFYVE-1 and sgPIKFYVE-2) validating target knockdown compared to control (sgNC). Bars are +/-SD. (One way ANOVA with Dunnett's.). **b.** Immunoblot analysis of MIA PaCa-2 (left) and PANC-1 (right) cells upon CRISPRi-mediated knockdown of *PIKFYVE* showing changes in PIKfyve, p62 (SQSTM1), and LC3A/B. Vinculin was used as a loading control for all blots. **c.** Immunoblot analysis of p62 and LC3A/B upon treatment with PIKfyve inhibitors apilimod or ESK981 in 7940B and Panc 04.03 cell lines. Vinculin or GAPDH were used as loading controls. **d.** Immunoblot of UM-2 primary cell-derived xenograft tumors described in Extended Data Fig. 3r,s. showing changes in LC3A/B. Vinculin was used as a loading control. **e.** Tandem fluorescent autophagic flux reporter assay in 7940B cells after 24-hour treatment with apilimod (100 nM), ESK981 (1000 nM), and chloroquine (CQ, 50 μM) with or without mTORC1/mTORC2 inhibitor torin-1 (100 nM). Box represents 25th and 75th percentiles; whiskers represent the range. (One-way ANOVA with Šidák's.). **f.** Tandem fluorescent reporter assay in Panc 04.03 cells showing changes in autophagic flux upon 4-hour pre-treatment with DMSO, apilimod (300 nM), ESK981 (1000 nM), or chloroquine (30 mM) and subsequent treatment of torin-1 (100 nM) or DMSO for 24 hours. Box represents 25th and 75th percentiles; whiskers represent the range. (One-way ANOVA with Dunnett's.). **g.** Chemical structures of PIK5-12d and PIK5-33d. Red indicates protein of interest ligand; black indicates chemical linker; blue indicates VHL E3 ligase ligand. **h.** Immunoblot of 7940B (top) and MIA PaCa-2 (bottom) cells treated with PIK5-33d at the indicated doses for 24 hours showing changes in p62 and LC3A/B. Vinculin was used as a loading control. **i-m.** Relative abundance of PI(3,5)P2 **(i)**, PI5P **(j)**, PI3P **(k)**, PI4P **(l)**, and PI(4,5)P2 **(m)** in PANC1 cells upon treatment with apilimod (1000 nM) or PIK-5-33d (33 d) (1000 nM) for the indicated times. The DMSO and apilimod conditions are from the same data used in Fig. 2b,c and Extended Data Fig. 3a–c and are used as comparisons for the effectiveness of PIK5-33d. **n.** 20x bright-field images of 7940B (top) MIA-PaCa-2 (bottom) cells upon treatment with PIKfyve inhibitors apilimod or ESK981 or PIK5-33d for 8 h. **o.** 20x bright-field images of MIA-PaCa-2 (top) and PANC-1 (bottom) cells upon CRISPRi-mediated knockdown of *PIKFYVE* (sgPIKFYVE-1) or control (sgNC). **p.** Confluence assay of MIA PaCa-2 (top) and PANC1 (bottom) cells upon CRISPRi-mediated knockdown of *PIKFYVE* (sgPIKFYVE) or control (sgNC). Bars are +/-SEM (Two-way ANOVA with Dunnett's.). **q.** Dose-response curve series of indicated PDAC cell lines treated with apilimod (top), ESK981 (middle), or chloroquine (bottom) for seven days using CellTiter-Glo assays. Bars are +/-SD. **r.** $IC_{50}$ values of each drug in each cell line tested. **s.** Box-and-whisker plot displaying $IC_{50}$ values of apilimod, ESK981, and chloroquine (CQ) in 7 human and mouse PDAC cell lines from **(q-r)**. Box represents 25th and 75th percentiles; whiskers represent the range. Statistics were performed using a Repeated Measures one-way ANOVA with Reisser-Greenhouse correction and with Tukey's multiple comparisons

test with individual variances computed for each comparison. **t.** Immunoblot of 7940B cells treated as indicated assessing changes in HIF1α, p62, and LC3A/B. Vinculin was used as a loading control. **u.** Oxygen consumption rate (OCR) in 7940B (top) and Panc 04.03 (bottom) cells upon treatment with apilimod (100 nM), ESK981 (1000 nM), bafilomycin (BAF, 100 nM), or CQ (100 μM) for 8 hours. Automated addition of Rot/AA and 2-DG were performed at the indicated time points. Bars are +/-SEM. (One-way ANOVA with Dunnett's.). **v.** (top) Real-time oxygen consumption rate monitoring by Resipher on 7940B cells upon treatment with apilimod (100 nM), ESK981 (1000 nM) bafilomycin (100 nM), or chloroquine (100 μM). (bottom) OCR at 8 hours and 24 hours by Resipher measurements from the same experiment. (One-way ANOVA with Dunnett's.). **w.** Dose-response curve series of 7940B (top) and PANC-1 (bottom) cells treated with bafilomycin, apilimod, or ESK981 with or without ferric ammonium citrate (FAC). All conditions containing FAC were also treated with ferrostatin-1 (1 μM) to block incidental ferroptosis. Bars are +/-SD. **x.** $IC_{50}$ values of each treatment in cell lines with or without FAC co-treatment. **y.** Confluence assay of Panc 04.03 cells undergoing treatment with deferoxamine (DFO) (100 μM), apilimod (300 nM), ESK981 (1000 nM), or bafilomycin (100 nM), without (top) or with (bottom) FAC (100 μg/mL) and ferrostatin-1 (1 μM). DFO, an iron chelator, was used as a positive control. Bars are +/-SEM. **Statistics and reproducibility: a.** Data shown are technical replicates from one of three independent experiments. P-values: sgNC vs sgPIKFYVE-1:8.7e-8; sgNC vs sgPIKFYVE-2:5.8e-7. **b.** This experiment was performed twice independently. **c.** This experiment was run twice in 7940B cells and once in Panc 04.03 cells. **d.** This experiment was performed once. **e.** n = 4 biological replicates for each condition. This represents one of three independent experiments. P-values: DMSO vs apilimod 4.4e-6; DMSO vs ESK981:1.7e-5; DMSO vs chloroquine:1.4e-5; DMSO+torin-1 vs apilimod+torin-1:1.2e-11; DMSO+torin-1 vs ESK981+torin-1:5.4e-11; DMSO+torin-1 vs chloroquine+torin-1:2.3-12. **f.** P-values: torin-1 vs DMSO: 5.6e-8; torin-1 vs torin-1+chloroquine: 2.1e-5. This experiment was performed independently thrice. **h.** These experiments were performed twice independently. **i-m.** One replicate of each condition was analyzed together, in two independent experiments (total n = 2 for each group). Bars are +/-SD. **n-o.** These images are representative of n = 3 each. **p.** n = 4 biological replicates. P-values: MIA PaCa-2: sgPIKFYVE-1 vs sgNC:1.7e-13; sgPIKFYVE-2 vs sgNC:1.7e-13; PANC1: sgPIKFYVE-1 vs sgNC:4.9e-14; sgPIKFYVE-2 vs sgNC:4.9e-14. These experiments were performed thrice each. **q.** n = 6 biological replicates. **t.** This experiment was performed twice. **u.** n = 5 biological replicates. Bars are +/-SEM. This experiment was performed twice independently. P-values: 7940B: DMSO vs chloroquine:1.9e-9; DMSO vs bafilomycin:9.1e-7. Panc 04.03: DMSO vs chloroquine:1.0e-3; DMSO vs bafilomycin:8.2e-3; DMSO vs ESK981:4.9e-3. **v.** Data shown are mean +/-SEM from 4 biological replicates. P-value: 8 hours DMSO vs chloroquine:6.6e-5; 24 hours DMSO vs bafilomycin:1.0e-7; 24 hours DMSO vs chloroquine:2.0e-8. **w.** n = 6 independent biological samples for each condition. **y.** n = 6 biological replicates.

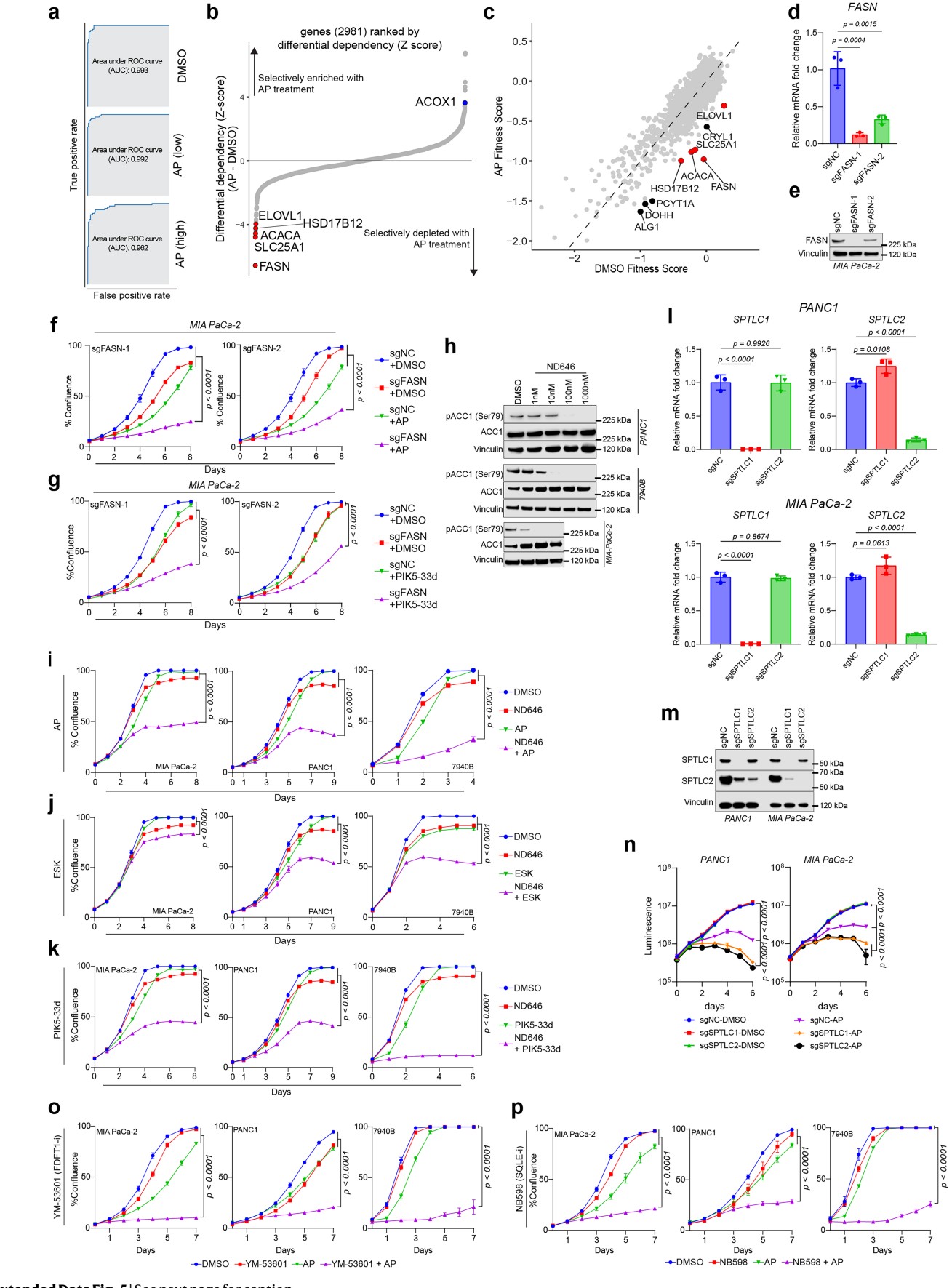

**Extended Data Fig. 5 |** See next page for caption.

**Extended Data Fig. 5 | De novo lipid synthesis is synthetically critical upon PIKfyve inhibition. a**. Receiver operator characteristic (ROC) curves for the prediction of core essential genes using datasets from MIA PaCa-2 CRISPR screens. **b**. Gene enrichment rank plot based differential sgRNA representation in (low dose, 100 nM) apilimod- treated versus DMSO-treated endpoint populations of the CRISPR screen experiment. Lipid synthesis-related genes ranked at either extreme are highlighted. **c**. Scatter plot of gene fitness scores in (low dose, 100 nM) apilimod-treated versus DMSO-treated endpoint conditions in metabolic CRISPR screen. Top 10 hits are labeled, and 5 lipid synthesis-related genes are highlighted. **d**. qPCR showing changes in mRNA levels of *FASN* upon CRISPRi-mediated knockdown of *FASN* in MIA PaCa-2 cells. (One-way ANOVA with Dunnett's.). **e**. Immunoblot of MIA PaCa-2 upon CRISPRi-mediated knockdown of *FASN* in MIA PaCa-2 cells. Vinculin was used as a loading control for the immunoblot. **f**. MIA PaCa-2 cell confluence assays upon *FASN* knockdown (sgFASN) or control (sgNC) treated with apilimod (100 nM) or DMSO. The sgNC DMSO and sgNC AP conditions are shared between the two graphs. Bars are +/-SEM. (Two-way ANOVA with Dunnett's.). **g**. Confluence assays of MIA PaCa-2 cells upon *FASN* knockdown or control upon treatment with PIKfyve degrader PIK5-33d (100 nM) or DMSO. The sgNC DMSO and sgNC PIK5-33d conditions are shared between the two graphs. Bars are +/-SEM. (Two-way ANOVA with Dunnett's.). **h**. Immunoblots of PANC-1, 7940B, and MIA PaCa-2 cells upon treatment with ND646 at indicated doses for 24 hours showing changes in labeled targets. **i**. Confluence assays of MIA PaCa-2 (left), PANC1 (middle), and 7940B (right) cells upon treatment with ND646 (100 nM for MIA-PaCa2, 1000 nM for PANC-1 and 7940B), apilimod (100 nM for MIA PaCa-2, 50 nM for PANC1 and 7940B), or both. Bars are +/-SEM. (Two-way ANOVA with Dunnett's.) The data for DMSO and ND646 are also utilized in Extended Data Fig. 5j (MIA PaCa-2) or 5j-k (PANC1 and 7940B), as they were generated in the same experiment. **j**. Confluence assays of MIA PaCa-2, PANC-1, and 7940B cells upon treatment with ND646 (100 nM for MIA-PaCa2, 1000 nM for PANC-1 and 7940B), ESK981 (30 nM for MIA PaCa-2, 100 nM for PANC-1 and 7940B), or both. Bars are +/-SEM. (Two-way ANOVA with Dunnett's.) The data for DMSO and ND646 are also utilized in Extended Data Fig. 5i (MIA PaCa-2) or 5i,k (PANC1 and 7940B), as they were generated in the same experiment. **k**. Confluence assays of MIA PaCa-2, PANC-1, and 7940B cells upon treatment with ND646 (100 nM for MIA-PaCa2, 1000 nM for PANC-1 and 7940B), PIK5-33d (100 nM for MIA PaCa-2 and PANC-1, 1000 nM for 7940B), or both. Bars are +/-SEM. (Two-way ANOVA with Dunnett's.) The data for DMSO and ND646 are also utilized in Extended Data Fig. 5i–k (PANC1 and 7940B), as they were generated in the same experiment. **l**. qPCR of PANC1 (top) and MIA PaCa-2 (bottom) upon CRISPRi-mediated knockdown of *SPTLC1*, *SPTLC2*, or control showing changes in *SPTLC1* (left) and *SPTLC2* (right). Bars are +/-SD. (One-way ANOVA with Dunnett's.). **m**. Immunoblot of PANC1 and MIA PaCa-2 upon CRISPRi-mediated knockdown of *SPTLC1*, *SPTLC2*, or control showing changes in SPTLC1 and SPTLC2. Vinculin was used as a loading control. **n**. Luminescence values from CellTiter-Glo analyses on PANC1 (left) or MIA PaCa-2 (right) cells upon knockdown of *SPTLC1*, *SPTLC2*, or control, and treatment with apilimod (3000 nM) or DMSO for the indicated duration. Bars are +/-SEM. (Two-way ANOVA with Dunnett's.). **o**. Confluence assays of MIA PaCa-2, PANC1, and 7940B upon treatment with DMSO, apilimod (300 nM for MIA PaCa-2, 100 nM for PANC1, and 50 nM for 7940B, YM-53601 (5000 nM), or both. Bars are +/-SEM. (Two-way ANOVA with Dunnett's.). **p**. Confluence assays of MIA PaCa-2, PANC1, and 7940B upon treatment with DMSO, apilimod (300 nM for MIA PaCa-2, 100 nM for PANC1, and 50 nM for 7940B), NB-598 (5000 nM), or both. Bars are +/-SEM. (Two-way ANOVA with Dunnett's.). **Statistics and reproducibility: d**. Data shown are technical triplicates from one of two independent experiments. **e**. This experiment was performed twice. **f**. n = 4 biological replicates. P-values: sgFASN-1+apilimod vs sgNC+DMSO:1.1e-16; sgFASN-1+apilimod vs sgFASN-1 + DMSO:1.8e-13; sgFASN-1+apilimod vs sgNC+apilimod:1.8e-13; sgFASN-2+apilimod vs sgNC+DMSO:1.8e-13; sgFASN-2+apilimod vs sgFASN-2 + DMSO:1.8e-13; sgFASN-2+apilimod vs sgNC+apilimod:1.8e-13. This experiment was performed twice. **g**. n = 4 biological replicates. P-values: sgFASN-1 + PIK5-33d vs sgNC+DMSO:1.8e-13; sgFASN-1 + PIK5-33d vs sgFASN-1 + DMSO:1.8e-13; sgFASN-1 + PIK5-33d vs sgNC+PIK5-33d:1.8e-13; sgFASN-2 + PIK5-33d vs sgNC+DMSO:1.8e-13; sgFASN-2 + PIK5-33d vs sgFASN-2 + DMSO:1.8e-13; sgFASN-2 + PIK5-33d vs sgNC+PIK5-33d:1.8e-13. This data is representative of two independent experiments. **h**. These experiments were performed once each. **i**. n = 6 biological replicates for MIA PaCa-2. n = 4 biological replicates per group for PANC1 and 7940B. P-values: MIA PaCa-2: ND646+apilimod vs DMSO:2.9e-13; vs ND646:2.9e-13; vs apilimod:2.9e-13; PANC1: ND646+apilimod vs DMSO: < 1.0e-15; vs ND646: < 1.0e-15; vs apilimod:<1.0e-15; 7940B: ND646+ apilimod vs DMSO:1.5e-13; vs ND646:1.5e-13; vs apilimod:1.5e-13. **j**. n = 6 biological replicates for MIA PaCa-2. n = 4 biological replicates for PANC1 and 7940B. P-values: MIA PaCa-2: ND646 + ESK981 vs DMSO:2.9e-13; vs ND646:2.9e-13; vs ESK981:2.9e-13; PANC1: ND646 + ESK981 vs DMSO: < 1.0e-15; vs ND646: < 1.0e-15; vs ESK981: < 1.0e-15; 7940B: ND646 + ESK981 vs DMSO:2.2e-13; vs ND646:2.2e-13; vs ESK981:2.2e-13. These experiments were performed thrice each. **k**. n = 4 biological replicates. P-values: MIA PaCa-2: ND646 + PIK5-33d vs DMSO:1.8e-13; vs ND646:1.8e-13; vs PIK5-33d:1.8e-13; PANC1: ND646 + PIK5-33d vs DMSO: < 1.0e-15; vs ND646: < 1.0e-15; vs PIK5-33d:<1.0e-15; 7940B: ND646 + PIK5-33d vs DMSO:2.2e-13; vs ND646:2.2e-13; vs PIK5-33d:2.2e-13. These experiments were performed thrice each. **l**. n = 3 technical triplicates. P-values: PANC1-SPTLC1: sgNC vs sgSPTLC1:2.4e-5; PANC1-SPTLC2: sgNC vs sgSPTLC2:1.7e-5; MIA PaCa-2-SPTLC1: sgNC vs sgSPTLC1:4.6e-7; MIA PaCa-2-SPTLC2: sgNC vs sgSPTLC2:1.8e-5. **m**. This experiment was performed twice. **n**. n = 4 biological replicates. P-values: PANC1: sgNC+apilimod vs sgNC+DMSO: < 1.0e-15; sgNC+apilimod vs sgSPTLC1+DMSO: < 1.0e-15; sgNC+apilimod vs sgSPTLC2+DMSO: < 1.0e-15; sgNC+apilimod vs sgSPTLC1+apilimod:<1.0e-15; sgNC+apilimod vs sgSPTLC2+apilimod:<1.0e-15. MIA PaCa-2: sgNC+apilimod vs sgNC+DMSO: < 1.0e-15; sgNC+apilimod vs sgSPTLC1+DMSO: < 1.0e-15; sgNC+apilimod vs sgSPTLC2+DMSO: < 1.0e-15; sgNC+apilimod vs sgSPTLC1+apilimod: <1.0e-15; sgNC+apilimod vs sgSPTLC2+apilimod:<1.0e-15. **o**. n = 3 biological replicates. P-values: MIA PaCa-2: YM-53601+apilimod vs DMSO:3.0e-14; vs YM-53601:3.0e-14; vs apilimod:3.0e-14; PANC1: YM-53601+apilimod vs DMSO: 3.0e-14; vs YM-53601:3.0e-14; vs apilimod:3.0e-14; 7940B: YM-53601+apilimod vs DMSO: 3.0e-14; vs YM-53601:3.0e-14; vs apilimod:3.0e-14. **p**. n = 3 biological replicates. P-values: MIA PaCa-2: NB-598+apilimod vs DMSO:3.0e-14; vs NB-598:3.0e-14; vs apilimod:3.0e-14; PANC1: NB-598+apilimod vs DMSO: < 1.0e-15; vs NB-598: < 1.0e-15; vs apilimod:<1.0e-15; 7940B: NB-598+apilimod vs DMSO:3.0e-14; vs NB-598:3.0e-14; vs apilimod:3.0e-14.

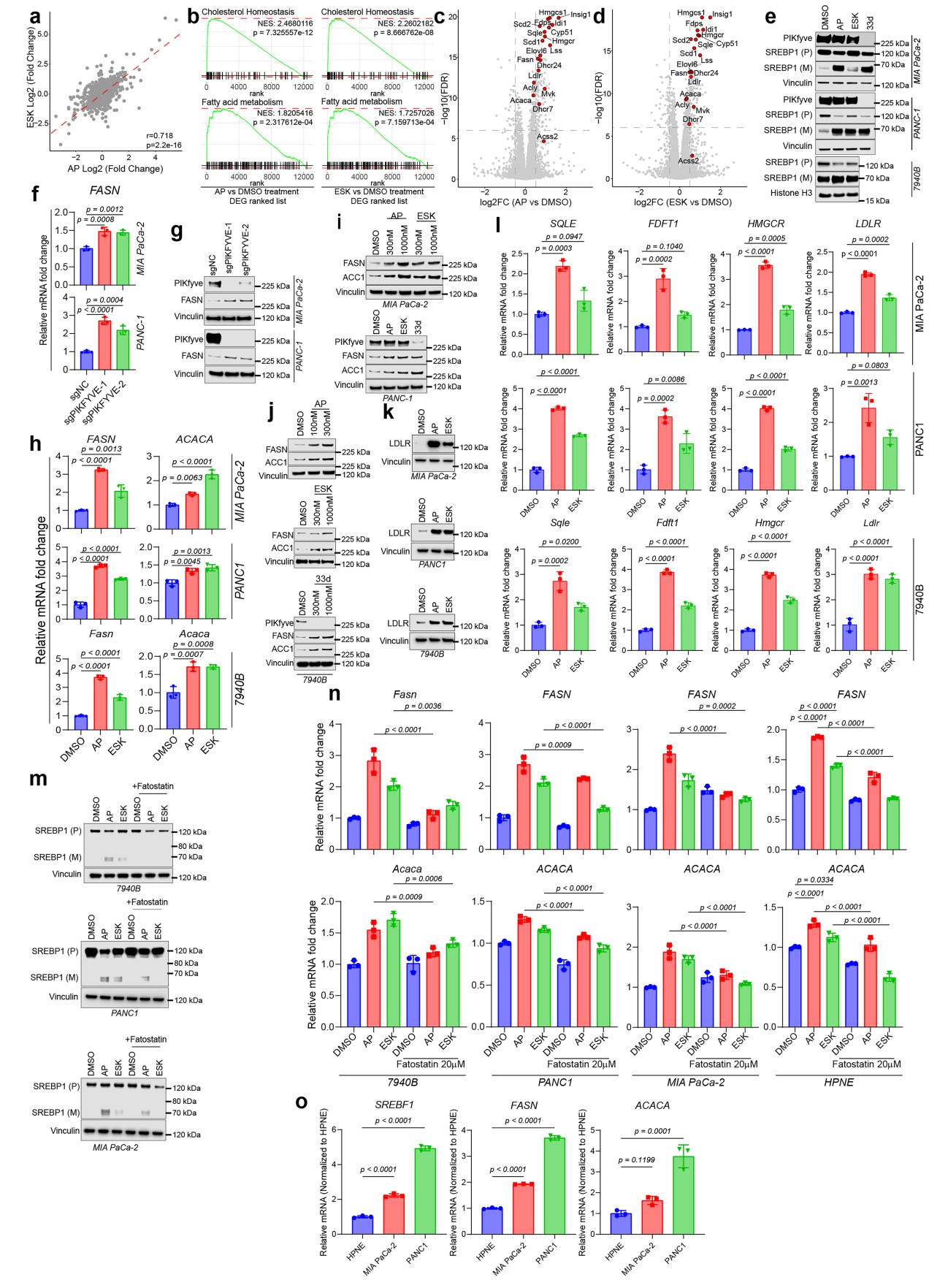

**Extended Data Fig. 6** | See next page for caption.

**Extended Data Fig. 6 | PIKfyve inhibition activates an SREBP-dependent lipogenic transcriptional signature. a.** Scatter plot of log2 fold change in gene expression upon 8-hour treatment with apilimod (100 nM) vs DMSO (x-axis) and ESK981 (1000 nM) vs DMSO (y-axis). A linear regression was calculated with r- and p-values displayed. **b.** GSEA plots of cholesterol homeostasis and fatty acid metabolism using the fold change rank-ordered gene signature from the 7940B cells treated with apilimod or ESK981 for 8 hours. (GSEA enrichment test.). **c-d.** Volcano plot using RNA-seq analysis on 7940B cells treated with apilimod (25 nM, **c.**) or ESK981 (250 nM, **d.**) for eight hours highlighting SREBP-1 target genes. Vertical dashed lines indicate log2 fold change = +/-0.5. Horizontal dashed line indicates FDR = $10^{-6}$. **e.** Immunoblot showing PIKfyve, premature SREBP1 (SREBP1 (P)), and mature SREBP1 (SREBP1 (M)) in MIA PaCa-2, PANC-1, and 7940B cells upon treatment with PIKfyve inhibitors or degrader PIK5-33d (33 d) for 8 hours. Vinculin or histone H3 were used as loading controls. The drug doses used were as follows: MIA-PaCa-2 and PANC-1: apilimod=300 nM, ESK981 = 1000 nM, PIK5-33d = 1000 nM; 7940B: apilimod=100 nM, ESK981 = 1000 nM, PIK5-33d = 1000 nM. **f.** qPCR of MIA PaCa-2 and PANC1 showing changes in RNA levels of *FASN* upon CRISPRi-mediated knockdown of *PIKFYVE*. Bars are +/-SD. (One-way ANOVA with Dunnett's.). **g.** Immunoblot analysis of MIA PaCa-2 and PANC1 showing changes in protein levels of FASN upon CRISPRi-mediated knockdown of *PIKFYVE* using two independent sgRNAs targeting *PIKFYVE* relative to control. Vinculin was used as a loading control. **h.** qPCR of MIA PaCa-2, PANC-1, and 7940B showing changes in RNA levels of labeled genes upon treatment with PIKfyve inhibitors for 8 hours. The drug doses used were as follows: MIA PaCa-2 and PANC-1: apilimod = 300 nM, ESK981 = 1000 nM. 7940B: apilimod = 100 nM, ESK981 = 1000 nM. Bars are +/-SD. (One-way ANOVA with Dunnett's.). **i-j.** Immunoblot analysis of MIA PaCa-2, PANC1 **(i.)**, and 7940B **(j.)** cells showing changes in protein levels of labeled genes upon treatment with PIKfyve inhibitors for 24 hours. Vinculin was used as a loading control. The drug doses used are indicated on the figure or as follows for PANC-1: apilimod = 300 nM, ESK981 = 1000 nM, PIK5-33d = 1000 nM. **k.** Immunoblot of MIA PaCa-2, PANC1, and 7940B cells treated with apilimod (100 nM for 7940B, 300 nM for MIA PaCa-2 and PANC1) or ESK981 (1000 nM) for 24 hours showing changes in LDLR. Vinculin was used as a loading control. **l.** qPCR showing changes in RNA levels of *SQLE*, *FDFT1*, *HMGCR*, or *LDLR* upon treatment with apilimod (100 nM for 7940B, 300 nM for MIA PaCa-2 and PANC1), or ESK981 (1000 nM) for 8 hours. Bars are +/-SD. (One-way ANOVA with Dunnett's.). **m.** Immunoblot analysis of 7940B, PANC1, and MIA PaCa-2 cells upon treatment with apilimod (100 nM for 7940B, 300 nM for PANC1, and MIA PaCa-2), ESK981 (1000 nM), fatostatin (20 µM), apilimod + fatostatin, or ESK981 + fatostatin as indicated for eight hours showing changes in premature SREBP1 (P) and mature SREBP1 (M). Vinculin was used as a loading control for all blots. **n.** qPCR showing changes in *FASN* (top) or *ACACA* (bottom) upon treatment with apilimod (100 nM for 7940B, 300 nM for PANC1 and MIA PaCa-2), ESK981 (1000 nM), fatostatin (20 µM), apilimod + fatostatin, or ESK981 + fatostatin as indicated for eight hours. Bars are +/-SD. (One way ANOVA with Šídák's multiple comparisons test.). **o.** qPCR of HPNE, MIA PaCa-2, and PANC1 cells showing differential baseline expressions of *SREBF1* (left), *FASN* (middle), and *ACACA* (right). Ct values were normalized to β-actin first and then to HPNE. Bars are +/-SD. (One-way ANOVA with Dunnett's.). **Statistics and reproducibility: e.** These experiments were performed twice with similar results. **f.** n = 3 technical triplicates. P-value: PANC1 *FASN*: sgNC vs sgPIKFYVE-1:5.5e-5. This experiment was performed thrice in MIA PaCa-2 cells with similar results and once in PANC1 cells. **g.** These experiments were performed once each. **h.** n = 3 technical triplicates. P-values: MIA PaCa-2 *FASN*: DMSO vs apilimod:2.0e-5; *ACACA*: DMSO vs ESK981:2.3e-5; PANC1 *FASN* DMSO vs apilimod:3.8e-7; DMSO vs ESK981:4.5e-6; 7940B *Fasn* DMSO vs apilimod:9.3e-7; DMSO vs ESK981:7.3e-5. These experiments were performed three independent times each with similar results. **i.** This experiment was performed once in each cell line. **j.** This experiment was performed twice with similar results. **k.** These experiments were performed once each. **l.** n = 3 technical triplicates per group. P-values: MIA PaCa-2: *HMGCR*: DMSO vs apilimod:5.3e-7; *LDLR*: DMSO vs apilimod:8.9e-7; PANC1: *SQLE*: DMSO vs apilimod:3.2e-8; DMSO vs ESK981:1.0e-6; *HMGCR*: DMSO vs apilimod:3.2e-8; DMSO vs ESK981:2.0e-5; 7940B: *Fdft1*: DMSO vs apilimod:1.2e-7; DMSO vs ESK981:2.0e-5; *Hmgcr*: DMSO vs apilimod:2.0e-7; DMSO vs ESK981:7.4e-6; *Ldlr*: DMSO vs apilimod:4.3e-5; DMSO vs ESK981:7.7e-5. **m.** These experiments were performed twice in each cell line. **n.** n = 3 technical triplicates per group. P-values: 7940B: *Fasn*: apilimod vs apilimod + fatostatin:2.4e-7; PANC1: *FASN*: ESK981 vs ESK981 + fatostatin:2.5e-6; *ACACA*: apilimod vs apilimod + fatostatin:3.2e-5; ESK981 vs ESK981 + fatostatin:1.8e-5; MIA PaCa-2: *FASN*: apilimod vs apilimod + fatostatin:1.1e-7; *ACACA*: apilimod vs apilimod + fatostatin:3.4e-5; ESK981 vs ESK981 + fatostatin:1.7e-5. HPNE: *FASN*: DMSO vs apilimod:2.8e-11; DMSO vs ESK981:2.6e-7; apilimod vs apilimod + fatostatin:6.9e-10; ESK981 vs ESK981 + fatostatin:6.8e-9; ACACA: DMSO vs apilimod:3.0e-5; apilimod vs apilimod + fatostatin:8.5e-5; ESK981 vs ESK981 + fatostatin:1.0e-7. **o.** n = 3 technical triplicates per group. P-values: SREBF1: HPNE vs MIA PaCa-2:1.5e-5; HPNE vs PANC1:1.5e-8; FASN: HPNE vs MIA PaCa-2:1.1e-6; HPNE vs PANC1:1.9e-9.

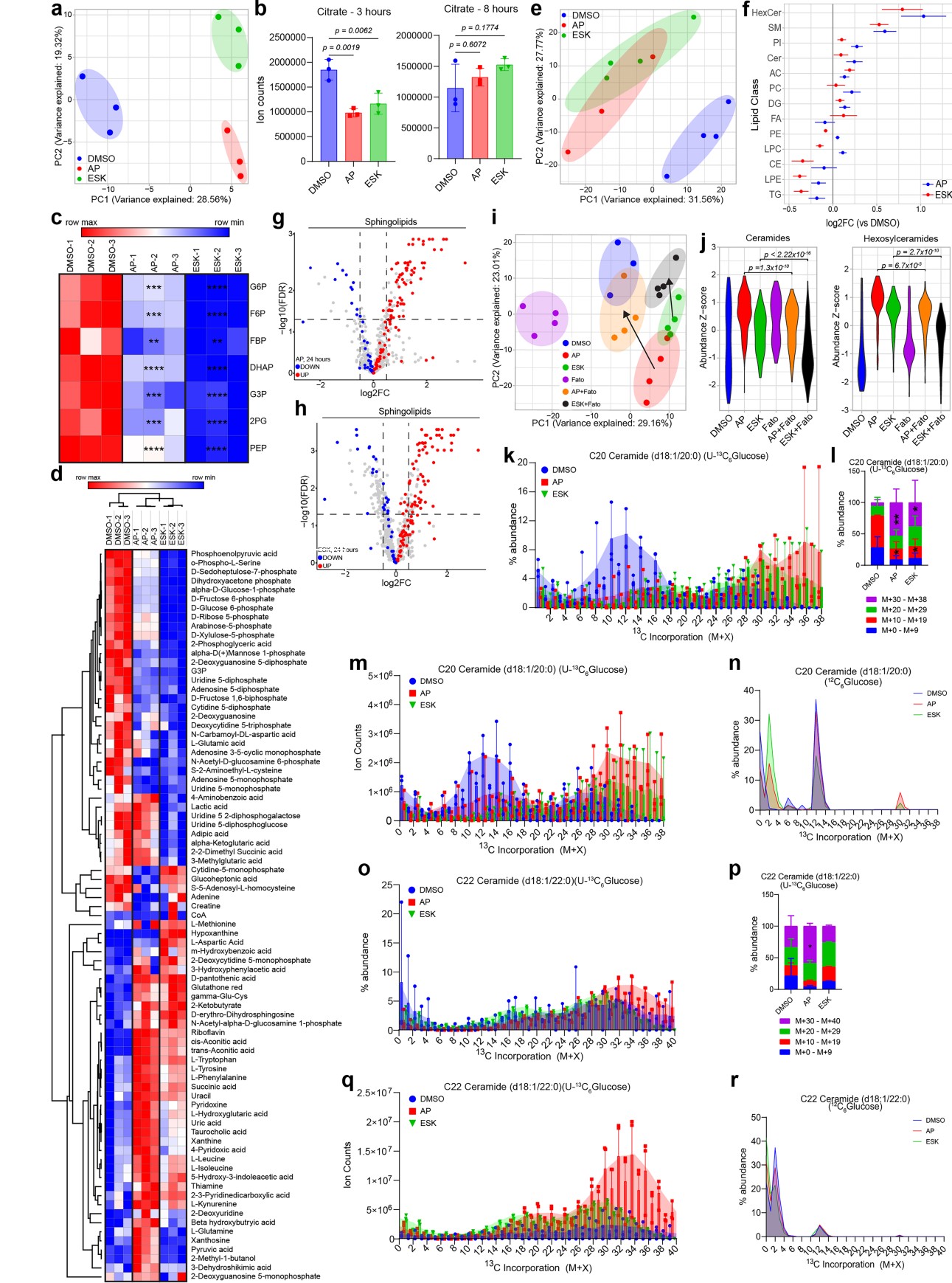

**Extended Data Fig. 7 |** See next page for caption.

**Extended Data Fig. 7 | PIKfyve inhibition reprograms PDAC cell metabolism to favor lipogenesis. a**. Principal component analysis (PCA) of targeted metabolomics experiment on 7940B cells treated with DMSO, apilimod (100 nM), or ESK981 (1000 nM) for eight hours. **b**. Citrate levels as detected by LC-MS-based metabolomics on 7940B cells treated with apilimod (100 nM) or ESK981 (1000 nM) for either three or eight hours, as indicated. Bars are +/-SD. (One-way ANOVA with Dunnett's.). **c**. Heatmap of glycolytic metabolite abundance in 7940B cells treated with DMSO, apilimod (100 nM), or ESK981 (1000 nM) for eight hours. G6P = glucose 6-phosphate; F6P = fructose 6-phosphate; FBP = fructose 1,6-bisphosphate; DHAP = dihydroxyacetone phosphate; G3P = glyceraldehyde 3-phosphate; 2PG = 2 phosphoglycerate; PEP = phosphoenolpyruvate. (One-way ANOVA with Dunnett's.) The same data is represented in **(d)**. **d**. Heatmap of significantly changed metabolites as determined by unpaired two-tailed t-test (apilimod vs DMSO or ESK981 vs DMSO, $p < 0.05$ in at least one of the two comparisons). Biological triplicates are displayed. **e**. PCA of targeted lipidomics experiment on 7940B cells treated with DMSO, apilimod (100 nM), or ESK981 (1000 nM) for 24 hours. Of note, one sample in the apilimod group was removed as an outlier. **f**. Forest plot indicating changes in lipid class abundance in 7940B cells upon treatment with DMSO, apilimod (100 nM), or ESK981 (1000 nM) for 24 hours. HexCer = hexosylceramide; SM = sphingomyelin; PI = phosphatidylinositol; Cer = ceramide; AC = acylcarnitine; PC = phosphatidylcholine; DG = diacylglyceride; FA = fatty acid; PE = phosphatidylethanolamine; LPC = lysophosphatidylcholine; CE = cholesteryl ester; LPE = lipophosphatidylethanolamine; TG = triacylglyceride. Effect sizes are in log2 scale of lipid abundance estimated from separate linear model for each treatment (apilimod or ESK981) compared to DMSO, adjusting for lipid classes with random intercept. Bars are 95% CI. **g-h**. Volcano plot showing differentially abundant lipid species in 7940B cells upon treatment with apilimod (100 nM, **g**) or ESK981 (1000 nM, **h**) for 24 hours. Highlighted in red are upregulated sphingolipid species. Highlighted in blue are downregulated sphingolipid species. (Two-tailed t-test using BH procedure to calculate FDR.). **i**. PCA of targeted lipidomics experiment on 7940B cells treated with DMSO, apilimod (100 nM), ESK981 (1000 nM), fatostatin (20 μM), apilimod + fatostatin, or ESK981 + fatostatin for 24 hours. Arrows are provided to highlight the change in lipid profile upon addition of fatostatin to apilimod or ESK981. Of note, data from one sample of the DSMO group was removed due to being a statistical outlier. **j**. Relative abundance of ceramides (left) and hexosylceramides (right) in 7940B cells upon treatment with apilimod (100 nM), ESK981 (1000 nM), fatostatin (20 μM), apilimod + fatostatin, or ESK+ fatostatin for 24 hours. (Unpaired two-tailed t-tests.). **k**. C20 ceramide (d18:1/20:0) isotopologue distribution in 7940B cells upon incubation with U-$^{13}$C$_6$ glucose and treatment with DMSO, apilimod (100 nM), or ESK91 (1000 nM) for 24 hours. Bars represent the range. **l**. Fractional enrichment pattern in C20 ceramide (d18:1/20:0) in 7940B cells upon incubation with U-$^{13}$C$_6$ glucose and treatment with DMSO, apilimod (100 nM), or ESK981 (1000 nM) for 24 hours, binned as indicated. Bars are +/-SD. (Two-way ANOVA with Dunnett's with DMSO as the baseline.). **m**. Total ion counts of individual isotopologues of C20 ceramide (d18:1/20:0) in 7940B cells upon incubation with U-$^{13}$C$_6$ glucose and treatment with DMSO, apilimod (100 nM), or ESK981 (1000 nM). This is the pre-normalized data from **(l)**. Bars represent the range. **n**. C20 ceramide (d18:1/20:0) isotopologue distribution in 7940B cells upon incubation with $^{12}$C$_6$ glucose and treatment with DMSO, apilimod (100 nM), or ESK91 (1000 nM) for 24 hours. Extraneous peaks were detected at M + 12-14 and M + 30-32 which are suspected to be contaminating species that were co-eluted with our target species. **o**. Fractional enrichment pattern in C22 ceramide (d18:1/22:0) in 7940B cells upon incubation with U-$^{13}$C$_6$ glucose and treatment with DMSO, apilimod (100 nM), or ESK91 (1000 nM) for 24 hours. Bars represent the range. **p**. Data from **(o)** binned as indicated. Bars are +/-SD. (Two-way ANOVA with Dunnett's.). **q**. Total ion counts of individual isotopologues of C22 ceramide (d18:1/22:0) in 7940B cells upon incubation with U-$^{13}$C$_6$ glucose and treatment with DMSO, apilimod (100 nM), or ESK981 (1000 nM). This is the pre-normalized data from **(o)**. Bars represent the range. **r**. C22 ceramide (d18:1/22:0) isotopologue distribution in 7940B cells upon incubation with $^{12}$C$_6$ glucose and treatment with DMSO, apilimod (100 nM), or ESK981 (1000 nM) for 24 hours. Extraneous peaks were detected at M + 12-14 which is suspected to be a contaminating species that were co-eluted with our target species. **Statistics and reproducibility: b**. n = 3 biological replicates. **c**. * indicates p < 0.05, ** indicates p < 0.01, *** indicates p < 0.001, **** indicates p < 0.0001. P-values: G6P: DMSO vs apilimod:9.0e-4; DMSO vs ESK981:2.6e-5; F6P: DMSO vs apilimod:5.5e-4; DMSO vs ESK981:2.6e-5; FBP: DMSO vs apilimod:0.007; DMSO vs ESK981:0.002; DHAP: DMSO vs apilimod:6.8e-6; DMSO vs ESK981:3.1e-7; G3P: DMSO vs apilimod:1.2e-4; DMSO vs ESK981:4.1e-5; 2PG: DMSO vs apilimod:8.2e-4; DMSO vs ESK981:7.2e-5; PEP: DMSO vs apilimod:6.9e-6; DMSO vs ESK981:1.5e-7. **f-h**. n = 4 biological replicates for DMSO and ESK981; n = 3 biological replicates for apilimod. **i**; n = 4 biological replicates per group for AP, ESK, Fato, AP+Fato, and ESK+Fato groups, n = 3 for the DMSO group. **j**. n = 4 biological replicates per group for AP, ESK, Fato, AP+Fato, and ESK+Fato groups, n = 3 for the DMSO group. P-values: Ceramides: ESK981 vs ESK981 + fatostatin:2.6e-39. **k**. n = 3 biological replicates per group. **l**. n = 3 biological replicates per group. P-values: M + 10-M + 19: DMSO vs AP:0.031; DMSO vs ESK:0.040; M + 30-M + 38: DMSO vs AP:0.0021; DMSO vs ESK:0.034. **m**. n = 3 biological replicates each. **o**. n = 3 biological replicates per group. **p**. n = 3 biological replicates per group. P-value: M + 30-M + 40: apilimod vs DMSO:0.013. **q**. n = 3 biological replicates per group.

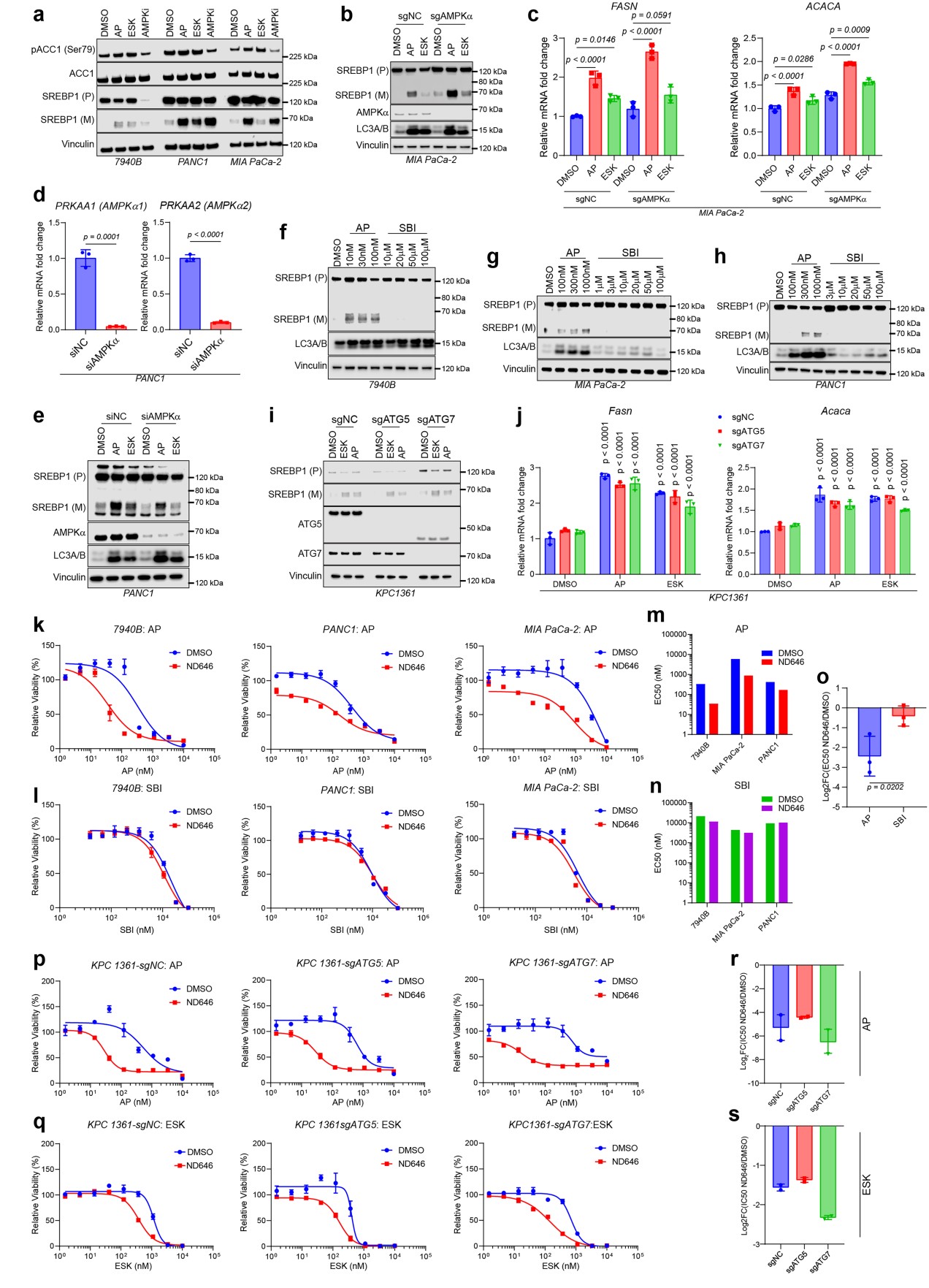

**Extended Data Fig. 8** | See next page for caption.

**Extended Data Fig. 8 | PIKfyve inhibition does not activate SREBP-dependent transcriptional signatures primarily through AMPK or autophagic pathways. a**. Immunoblot analysis of 7940B, PANC1, and MIA PaCa-2 cells upon treatment with apilimod (100 nM for 7940B, 300 nM for PANC1 and MIA PaCa-2), ESK981 (1000 nM), or Compound C (dorsomorphin, labeled "AMPKi", 20 μM) for eight hours showing changes in pACC1 (Ser 79), total ACC1, SREBP1 (P), and SREBP1 (M). Vinculin was used as loading control. **b**. Immunoblot analysis of MIA PaCa-2 cells upon CRISPR knockout of *PRKAA1* and *PRKAA2* (AMPKα1 and AMPKα2) or control and treatment with apilimod (300 nM) or ESK981 (1000 nM) for eight hours showing changes of SREBP1 (P), SREBP1 (M), AMPKα, and LC3A/B. Vinculin was used as a loading control. **c**. qPCR analysis showing changes in *FASN* (left) or *ACACA* (right) of MIA PaCa-2 upon CRISPR knockout of AMPKα1/2 and treatment with apilimod (300 nM) or ESK981 (1000 nM) for eight hours. All conditions were normalized to sgNC-DMSO condition. Bars are +/-SD. (One way ANOVA with Šídák's multiple comparisons test.) **d**. qPCR of PANC1 cells upon siRNA knockdown of *PRKAA1* and *PRKAA2* (AMPKα1 and AMPKα2) or control showing changes in *PRKAA1* and *PRKAA2* (AMPKα1 and AMPKα2). Bars are +/-SD. (Unpaired two-tailed t-test.). **e**. Immunoblot analysis of PANC1 cells upon siRNA knockdown of AMPKα1/2 or control and treatment with apilimod (300 nM) or ESK981 (1000 nM) for eight hours showing changes in SREBP1 (P), SREBP1 (M), AMPKα, and LC3A/B. Vinculin was used as a loading control. **f**. Immunoblots of 7940B cells treated with apilimod or SBI-0206965 at the indicated doses for eight hours showing changes in SREBP1 (P), SREBP1 (M), and LC3 A/B. Vinculin was used as a loading control. This experiment was performed twice with similar results. **g**. Immunoblots of MIA PaCa-2 cells treated with apilimod or SBI-0206965 at the indicated doses for eight hours showing changes in SREBP1 (P), SREBP1 (M), and LC3 A/B. Vinculin was used as a loading control. This experiment was performed twice with similar results. **h**. Immunoblots of PANC1 cells treated with apilimod or SBI-0206965 at the indicated doses for eight hours showing changes in SREBP1 (P), SREBP1 (M), and LC3 A/B. Vinculin was used as a loading control. This experiment was performed twice with similar results. **i**. Immunoblot analysis of KPC 1361-sgNC, KPC 1361-sgATG5, or KPC 1361-sgATG7 cells treated with DMSO, ESK981 (1000 nM), or apilimod (300 nM) for eight hours showing changes in SREBP1 (P), SREBP1 (M), ATG5, and ATG7. Vinculin was used as a loading control. This experiment was performed twice with similar results. **j**. qPCR of KPC 1361-sgNC, KPC 1361-sgATG5, or KPC 1361-sgATG7 cells treated with DMSO, apilimod (300 nM), or ESK981 (1000 nM) for eight hours showing changes in *Fasn* (left) or *Acaca* (right). Bars are +/-SD. (Two-way ANOVA with Dunnett's multiple comparison test, setting DMSO as the reference for each cell line.). **k-l**. Dose-response curves of PANC1, MIA PaCa-2, or 7940B cells treated with either apilimod (top) or SBI-0206965 (SBI, ULK inhibitor) at the indicated doses and co-treated with either DMSO or ND646 (100 nM for MIA PaCa-2, 1000 nM for PANC1 and 7940B) for seven days. Relative viabilities were calculated by normalizing all values to the DMSO condition. Bars are +/-SD. **m**. EC50 of apilimod in 7940B, MIA PaCa-2, or PANC1 cells upon co-treatment with DMSO or ND646. **n**. EC50 of SBI-0206965 in 7940B, MIA PaCa-2, or PANC1 cells upon co-treatment with DMSO or ND646. **o**. Log₂ of the fold change in EC50 of apilimod or SBI-0206965 upon co-treatment with ND646 for 7940B, MIA PaCa-2, and PANC1 cells. Bars are +/-SD. (Paired two-tailed t-test.). **p-q**. Dose-response curves of KPC 1361-sgNC, KPC 1361-sgATG5, or KPC 1361-sgATG7 treated with indicated doses of apilimod and co-treated with either DMSO or ND646 (1000 nM) for seven days. Relative viabilities were calculated by normalizing all values to the DMSO condition. Bars are +/-SD. **r**. Dose-response curves of KPC 1361-sgNC, KPC 1361-sgATG5, or KPC 1361-sgATG7 treated with indicated doses of ESK981 and co-treated with either DMSO or ND646 (1000 nM) for seven days. Relative viabilities were calculated by normalizing all values to the DMSO condition. Bars are +/-SD. **s**. Log₂ of the fold change in IC50 of apilimod (top) or ESK981 (bottom) treatment upon co-treatment with ND646 (1000 nM) versus DMSO as calculated from two independent replicates of the experiments using KPC1361-sgNC, KPC1361-sgATG5, and KPC1361-sgATG7 displayed in Extended Data Fig. 8p,q. Bars are +/-SD. **Statistics and reproducibility a**. This experiment was performed once. **b**. This experiment was performed twice with similar results. **c**. n = 3 technical replicates per group; P-values: *FASN*: sgNC: DMSO vs apilimod:2.3e-5; sgAMPKα: DMSO vs apilimod:3.0e-7; *ACACA*: sgNC: DMSO vs apilimod:5.7e-5; sgAMPKα: DMSO vs apilimod:9.9e-8. This experiment was performed twice with similar results. **d**. n = 3 technical replicates. P-values: *PRKAA1*: siNC vs sgAMPKα:1.4e-4; *PRKAA2*: siNC vs sgAMPKα:6.0e-6. This experiment was performed twice with similar results. **e**. This experiment was performed twice with similar results. **f**. This experiment was performed twice with similar results. **g**. This experiment was performed twice with similar results. **h**. This experiment was performed twice with similar results. **i**. This experiment was performed twice with similar results. **j**. n = 3 biological replicates per group. P-values: *Fasn*: sgNC: DMSO vs apilimod:2.4e-12; DMSO vs ESK981:4.9e-10; sgATG5: DMSO vs apilimod:4.8e-10; DMSO vs ESK981:5.1e-8; sgATG7: DMSO vs apilimod:1.5e-10; DMSO vs ESK981:2.8e-6; *Acaca*: sgNC: DMSO vs apilimod:2.0e-10; DMSO vs ESK981:1.2e-9; sgATG5: DMSO vs apilimod:4.5e-7; DMSO vs ESK981:2.3e-8; sgATG7: DMSO vs apilimod:2.0e-6; DMSO vs ESK981:7.4e-5. This experiment was performed thrice with similar results. **k-l**. n = 3 biological replicates per condition. **o**. n = 3 cell lines per group. **p-q**. n = 3 biological replicates per condition. These experiments were performed twice each with similar results. **r-s**. n = 2 independent experiments, represented by one data point for each condition.

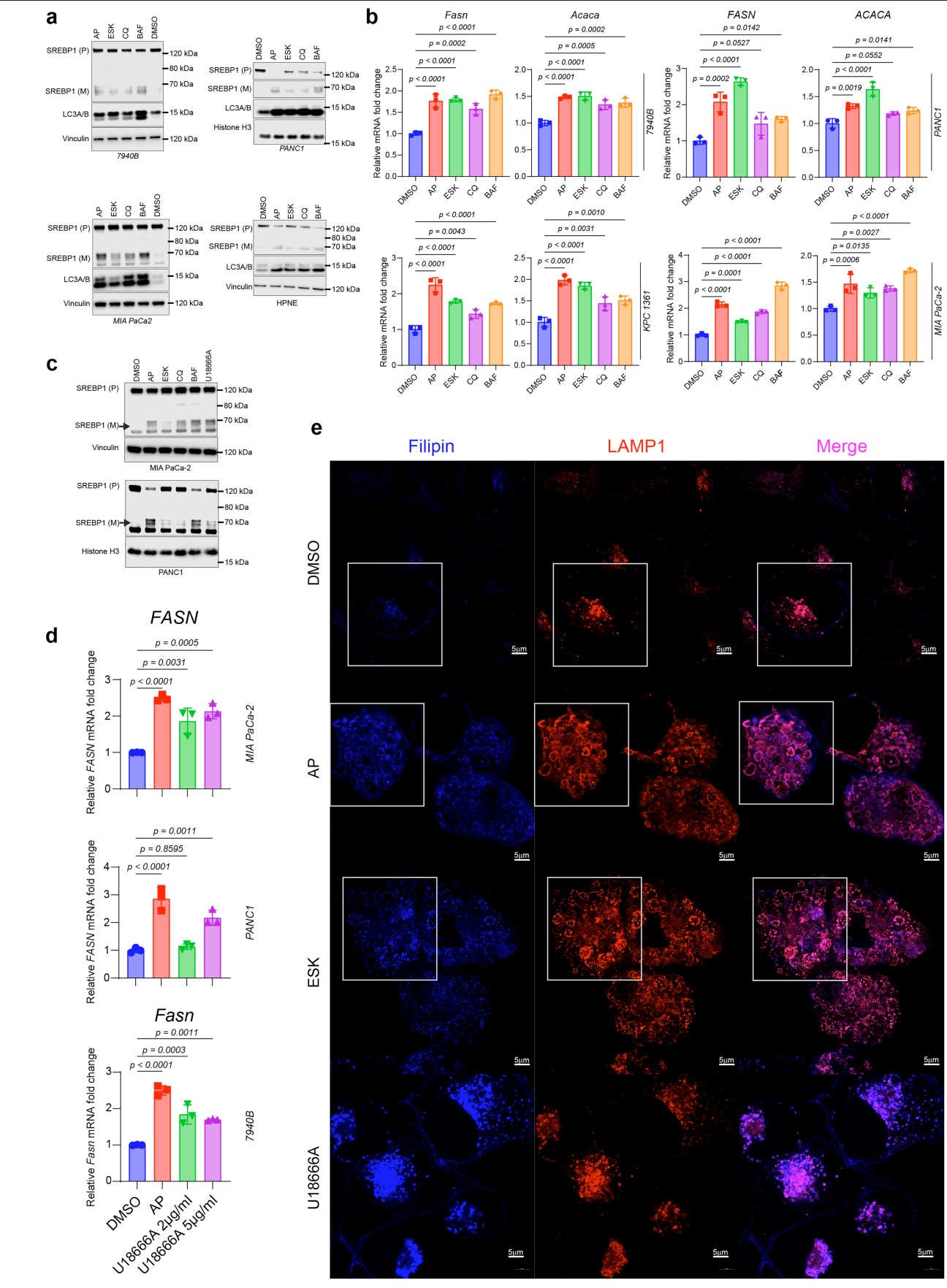

**Extended Data Fig. 9 |** See next page for caption.

**Extended Data Fig. 9 | Inhibition of PIKfyve activates SREBP-dependent transcriptional signatures through disrupting lysosomal lipid homeostasis pathways. a**. Immunoblots of 7940B, PANC1, MIA PaCa-2, and HPNE cells upon treatment with apilimod (100 nM for 7940B, 300 nM for PANC1, MIA PaCa-2, and HPNE), ESK981 (1000 nM), chloroquine (30 μM), or bafilomycin A1 (30 nM) for eight hours showing changes in SREBP1 (P), SREBP1 (M), and LC3A/B. Vinculin or histone H3 were used as loading controls. **b**. qPCR of 7940B, PANC1, MIA PaCa-2, or KPC1361 cells upon treatment with apilimod (100 nM for 7940B, 300 nM for PANC1, MIA PaCa-2, KPC 1361), ESK981 (1000 nM), chloroquine (CQ, 30 μM), or bafilomycin A1 (BF, 30 nM) for eight hours showing changes in *FASN* or *ACACA* mRNA levels as indicated. Bars are +/-SD. (Two-way ANOVA with Dunnett's multiple comparison test, setting DMSO as the reference for each target for each cell line.). **c**. Immunoblots of PANC1 and MIA PaCa-2 cells treated with apilimod (300 nM), ESK981 (1000 nM), chloroquine (30 μM), bafilomycin (30 nM), or U18666A (5 μg/mL) for 8 hours showing changes in SREBP1 (P) and SREBP1 (M). Histone H3 or vinculin are used as loading controls. **d**. qPCR of MIA PaCa-2, PANC1, and 7940B cells assessing changes in RNA of *FASN* upon treatment with apilimod (100 nM for 7940B,

300 nM for PANC1, MIA PaCa-2) or U18666A (5 μg/mL) for 8 hours. Bars are +/-SD. (One-way ANOVA with Dunnett's, with DMSO as the baseline.). **e**. Immunofluorescence images of PANC1 cells treated with DMSO, apilimod (1000 nM), ESK981 (1000 nM), or U18666A (5 μg/mL) for 24 hours stained with filipin or LAMP1. White squares indicate cropped area used for Fig. 3f. Scalebars= 5 μm. **Statistics and reproducibility: a**. This experiment was performed twice in MIA PaCa-2 cells with similar results and once in all other cells. **b**. n = 3 technical replicates per group. P-values: 7940B: *Fasn*: DMSO vs apilimod:1.6e-5; DMSO vs ESK981:1.2e-5; DMSO vs bafilomycin:3.0e-6; *Acaca*: DMSO vs apilimod:2.7e-5; DMSO vs ESK981:2.0e-5; KPC 1361: *Fasn*: DMSO vs apilimod:5.9e-7; DMSO vs ESK981:4.3e-5; DMSO vs BAF:7.8e-5; *Acaca*: DMSO vs apilimod:3.8e-6; DMSO vs ESK981:1.4e-5 PANC1: *FASN*: DMSO vs ESK981:5.8e-6; *ACACA*: DMSO vs ESK981:6.3e-6 MIA PaCa-2: *FASN*: DMSO vs apilimod:6.3e-8; DMSO vs chloroquine:9.1e-7; DMSO vs bafilomycin:5.9e-10; *ACACA*: DMSO vs bafilomycin:1.5e-5. **c**. These experiments were performed once each. **d**. n = 3 technical replicates per group. P-values: *FASN*: MIA PaCa-2: DMSO vs apilimod:1.5e-4; PANC1: DMSO vs apilimod:4.4e-5; *Fasn*: 7940B: DMSO vs apilimod:4.4e-6. **e**. These images are representative of n = 2 images each.

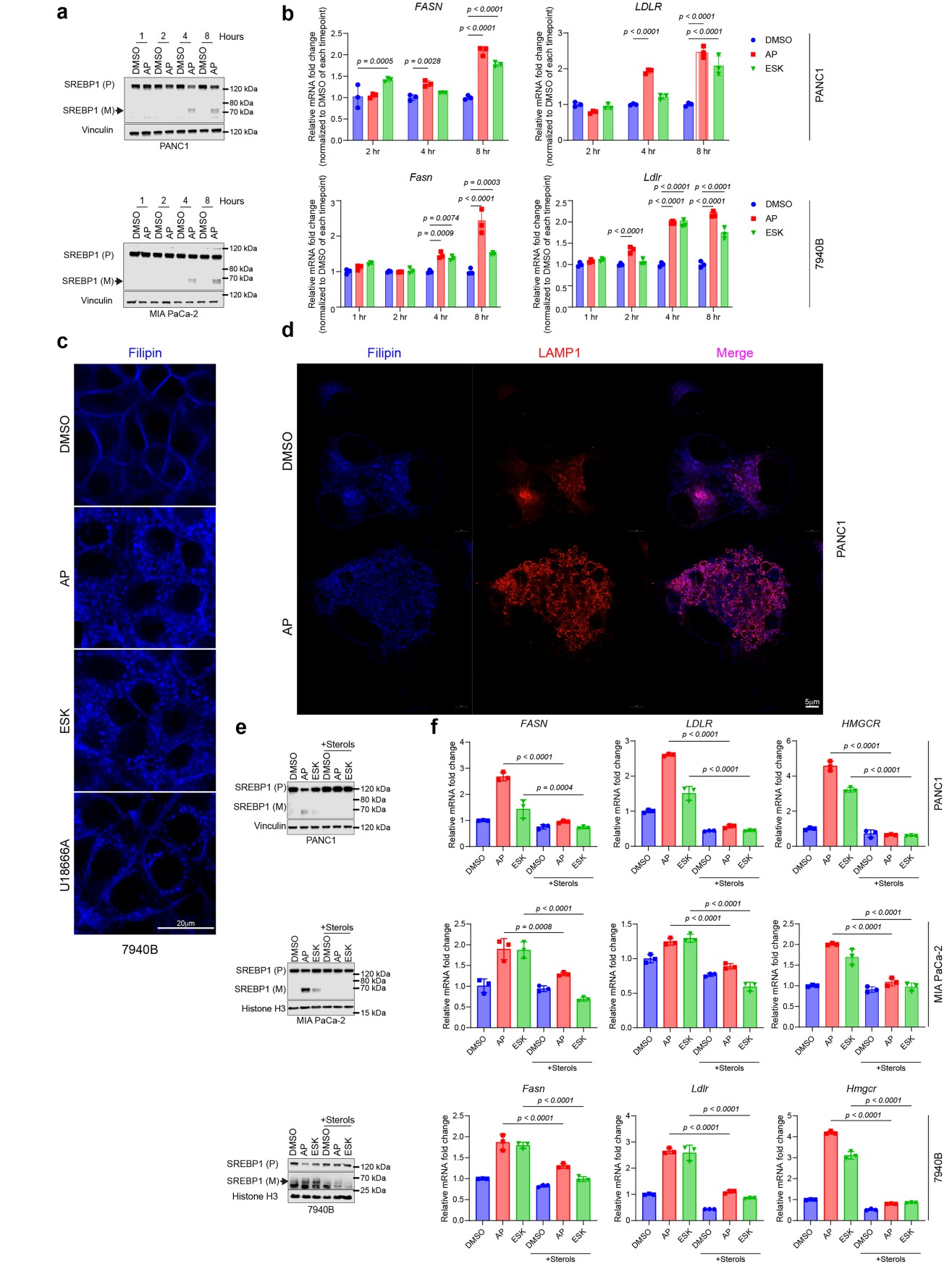

**Extended Data Fig. 10 | PIKfyve inhibition activates SREBP through causing lysosomal lipid sequestration. a**. Immunoblots of PANC1 and MIA PaCa-2 cells upon treatment with DMSO or apilimod (300 nM) for the indicated number of hours detecting changes of SREBP1 (P) and SREBP1 (M) protein levels. Vinculin was used as a loading control. **b**. qPCR showing changes in RNA levels of *FASN* (left) or *LDLR* (right) upon treatment with apilimod (300 nM for PANC1 and 100 nM for 7940B) or ESK981 (1000 nM) for the indicated number of hours. Bars are +/-SD. (Two-way ANOVA with Dunnett's with the DMSO condition for each timepoint set as the baseline and normalized to 1.). **c**. Immunofluorescence images of 7940B cells upon treatment with DMSO, apilimod (100 nM), ESK981 (1000 nM), or U18666A (5 μg/mL) for 4 hours stained with filipin. Scalebars=20 μm. **d**. Immunofluorescence images of PANC1 cells upon treatment with DMSO or apilimod (300 nM) for 4 hours stained with filipin and LAMP1. Scalebars=5 μm. **e**. Immunoblot of PANC1, MIA PaCa-2, and 7940B cells upon treatment with DMSO, apilimod (300 nM for PANC1 and MIA PaCa-2; 100 nM for 7940B), or ESK981 (1000 nM) with or without cholesterol supplementation (10 μg/mL cholesterol and 1 μg/mL 25-hydroxycholesterol) for 8 hours detecting changes of SREBP1 (P) and SREBP1 (M) protein levels. Vinculin or histone H3 were used as a loading control. **f**. qPCR showing changes in RNA levels of *FASN*, *LDLR*, or *HMGCR* in PANC1, 7940B, or MIA PaCa-2 cells upon treatment with DMSO, apilimod (100 nM for 7940B, 300 nM for MIA PaCa-2 and PANC1), or ESK981 (1000 nM) with or without cholesterol supplementation (10 μg/mL cholesterol and 1 μg/ mL 25-hydroxycholesterol) for 8 hours. Bars are +/-SD. (One way ANOVA with Šídák's multiple comparisons test.). **Statistics and reproducibility: a**. These experiments were performed once each. **b**. n = 3 technical triplicates per group. P-values: PANC1: *FASN*: 8 hour: DMSO vs apilimod:4.7e-10; DMSO vs ESK981:9.4e-8; *LDLR*: 4 hour: DMSO vs apilimod:2.5e-8; 8 hour: DMSO vs apilimod:1.7e-11; DMSO vs ESK981:2.3-9; 7940B: *Fasn*: 8 hour: DMSO vs apilimod:2.2e-12; *Ldlr*: 2 hour: DMSO vs apilimod:1.3e-5; 4 hour: DMSO vs apilimod:7.0e-15; DMSO vs ESK981:1.1e-14; 8 hour: DMSO vs apilimod:<1.0e-15; DMSO vs ESK981:9.9e-12. **c**. These images are representative of n = 2 each. **d**. These images are representative of n = 3 each. **e**. These experiments were performed once each. **f**. n = 3 technical triplicates per group. P-values: PANC1 *FASN*: apilmod vs apilimod + sterols:4.5e-8; *LDLR*: apilimod vs apilimod+sterols:3.9e-12; ESK981 vs ESK91+sterols:7.7e-9; *HMGCR*: apilimod vs apilimod+sterols:1.4e-12; ESK981 vs ESK981+sterols:1.6e-10; MIA PaCa-2: *FASN*: ESK981 vs ESK981+sterols:1.3e-6; *LDLR*: apilimod vs apilmod+sterols:4.5e-6; ESK981 vs ESK981+sterols:2.5e-9; *HMGCR*: apilimod vs apilimod+sterols:2.4e-7; ESK981 vs ESK981+sterols:3.0e-6; 7940B: *Fasn*: apilimod vs apilimod+sterols: 1.0e-5; ESK981 vs ESK981+sterols:2.3e-7; *Ldlr*: apilimod vs apilimod+sterols: 1.1e-8; ESK981 vs ESK981+sterols:3.4e-9; *Hmgcr*: apilimod vs apilimod+sterols: 4.0e-15; ESK981 vs ESK981 + sterols:5.1e-13.

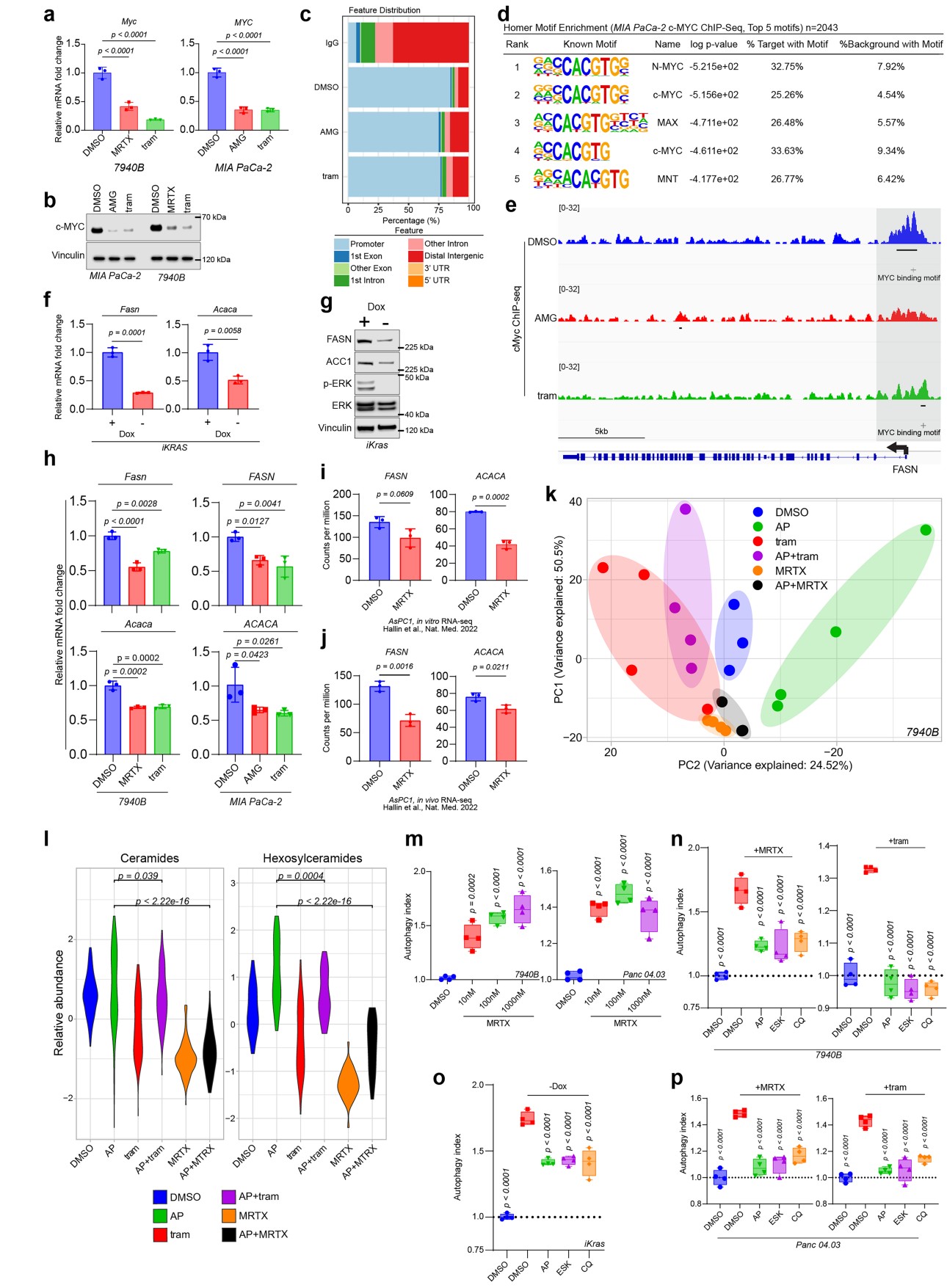

**Extended Data Fig. 11 |** See next page for caption.

**Extended Data Fig. 11 | KRAS-MAPK opposes PIKfyve in its regulation of lipogenesis and autophagy in PDAC cells. a**. qPCR of 7940B cells upon treatment with MRTX1133 (MRTX, KRAS$^{G12D}$ inhibitor) or trametinib (tram, MEK inhibitor) and MIA PaCa-2 cells upon treatment with AMG510 (AMG, KRAS$^{G12C}$ inhibitor) or trametinib for eight hours showing changes in *MYC* expression. Bars are +/-SD. (One-way ANOVA with Dunnett's.). **b**. Immunoblot of 7940B cells upon treatment with MRTX1133 or trametinib and MIA PaCa-2 upon treatment with AMG510 or trametinib for eight hours showing changes in c-MYC levels. Vinculin was used as a loading control. **c**. ChIP-seq read feature distribution upon pulldown of IgG (DMSO) or c-MYC from MIA PaCa-2 cell lysates upon treatment with DMSO, AMG510 (100 nM), or trametinib (10 nM) for eight hours. **d**. Top five known motifs (ranked by p-value, HOMER, hypergeometric test) enriched upon pulldown of c-MYC from MIA PaCa-2 cells treated with DMSO. **e**. ChIP-seq track of *FASN* gene locus showing peaks called by MACS2 and MYC binding motif called by FIMO in MIA PaCa-2 cells upon treatment with DMSO, AMG510, or trametinib. **f**. qPCR of iKRAS (doxycycline-inducible KRAS$^{G12D}$) 9805 cells showing changes in RNA levels of labeled genes upon presence or absence of doxycycline (Dox) for 48 hours. Bars are +/-SD. (Unpaired two-tailed t-test.). **g**. Immunoblot analysis of iKRAS 9805 cells showing changes in protein levels of FASN and ACC1 upon presence or absence of doxycycline for 72 hours. Vinculin was used as a loading control. **h**. qPCR of 7940B and MIA PaCa-2 cells treated with KRAS$^{G12D}$ inhibitor MRTX1133 (100 nM), KRAS$^{G12C}$ inhibitor AMG510 (100 nM for MIA PaCa-2), or MEK inhibitor trametinib (10 nM for MIA PaCa-2, 30 nM for 7940B) for eight hours. Bars are +/-SD. (One-way ANOVA with Dunnett's.). **i**. Counts per million (CPM) from RNA-seq analysis on AsPC1 cells treated with MRTX1133 (100 nM) for 24 hours. Data plotted are biological triplicates from publicly available RNA-seq data[43]. (Unpaired two-tailed t-test.) Bars are +/-SD. **j**. CPM from RNA-seq analysis on AsPC1 cell-derived xenograft model. Mice were dosed with 30 mg/kg of MRTX1133 6 hours prior to tumor collection. Data plotted are taken from independent tumors from publicly available RNA-seq data[43]. (Unpaired two-tailed t-test.) Bars are +/-SD. **k**. Principal component analysis of targeted lipidomics data showing shifts in 7940B global lipid profiles upon treatment with DMSO, apilimod (100 nM, 24 hours), trametinib (100 nM every 24 hours for 48 hours), MRTX1133 (1000 nM every 8 hours for 48 hours), apilimod and trametinib, or apilimod and MRTX. Of note, one sample in the DMSO group was removed as a statistical outlier. **l**. Relative abundance of ceramides (left) and hexosylceramides (right) in 7940B cells upon treatment with apilimod (100 nM for 24 hours), trametinib (100 nM every 24 hours for 48 hours), MRTX1133 (1000 nM every eight hours for 48 hours), apilimod + trametinib, or apilimod + MRTX. (Unpaired two-tailed t-tests.). **m**. Tandem fluorescent reporter assay in 7940B (left) or Panc 04.03 (right) cells showing changes in autophagic flux upon 24-hour treatment with labeled doses of MRTX1133. Data shown are four biological replicates from one of three independent experiments. Box represents 25th and 75th percentiles; whiskers represent the range. (One-way ANOVA with Dunnett's.). **n**. Tandem fluorescent reporter assay on 7940B cells showing changed autophagic flux

upon 4-hour pretreatment with apilimod (100 nM), ESK981 (1000 nM), or chloroquine (50 μM) and subsequent treatment with MRTX1133 (300 nM, left) or trametinib (25 nM, right) for 24 hours. (One-way ANOVA with Dunnett's.). **o**. Tandem fluorescent reporter assay on iKRAS 9805 cells showing changed autophagic flux upon doxycycline withdrawal for 24 hours and subsequent treatment with apilimod (100 nM), ESK981 (1000 nM), or chloroquine (10 μM) for 24 hours. Box represents 25th and 75th percentiles; whiskers represent the range. (One way ANOVA with Dunnett's.). **p**. Tandem fluorescent reporter assay in Panc 04.03 cells showing changes in autophagic flux upon four-hour pretreatment with apilimod (300 nM), ESK981 (1000 nM), or chloroquine (CQ, 30 mM) followed by 24-hour treatment with MRTX1133 (300 nM, left) or trametinib (25 nM). Data shown are four biological replicates from one of three independent experiments. Box represents 25th and 75th percentiles; whiskers represent the range. (One-way ANOVA with Dunnett's.). **Statistics and reproducibility: a**. n = 3 technical triplicates per group. P-values: 7940B: *Myc*: DMSO vs MRTX1133:9.0e-5; DMSO vs trametinib:1.3e-5; MIA PaCa-2: *MYC*: DMSO vs AMG510:1.4e-5; DMSO vs trametinib: 1.3e-5. This experiment was performed thrice each with similar results. **b**. This experiment was performed twice with similar results. **f**. n = 3 technical triplicates. This experiment was performed thrice with similar results. **g**. This experiment was performed once. **h**. n = 3 technical replicates. These experiments were performed thrice each with similar results. P-values: 7940B: *Fasn*: DMSO vs MRTX1133:5.9e-5. **k**. n = 4 for apilimod, trametinib, apilimod+trametinib, MRTX1133, and apilimod+ MRTX1133 groups; n = 3 for DMSO. **l**. n = 4 for apilimod, trametinib, apilimod+ trametinib, MRTX1133, and apilimod+MRTX1133 groups; n = 3 for DMSO. P-values: Ceramides: apilimod vs apilimod+MRTX1133:1.1e-109; Hexosylceramides: apilimod vs apilimod + MRTX1133:1.3e-46. **m**. n = 4 biological relicates each. P-values: 7940B: DMSO vs MRTX1133 100 nM: 5.6e-6; DMSO vs MRTX1133 1000 nM:1.5e-6; Panc04.03: DMSO vs MRTX1133 10 nM:4.7e-6; DMSO vs MRTX1133 100 nM:6.5e-7; DMSO vs MRTX1133 1000 nM:1.2e-5. **n**. n = 4 independent biological samples for each group. P-values: DMSO vs MRTX1133:9.4e-8; MRTX1133 vs MRTX1133+apilimod: 2.0e-5; MRTX1133 vs MRTX1133 + ESK981:1.5e-5; MRTX1133 vs MRTX1133+ chloroquine:5.8e-5; DMSO vs trametinib:8.4e-10; trametinib vs trametinib+ apilimod:3.1e-10; trametinib vs trametinib+ESK981:1.4e-10; trametinib vs trametinib+chloroquine:1.7e-10. This data is representative of three independent experiments each. **o**. n = 4 independent biological samples for each group. P-values: DMSO-Dox vs DMSO+Dox:2.4e-11; DMSO-Dox vs apilimod:1.5e-6; DMSO-Dox vs ESK981: 2.8e-6; DMSO-Dox vs chloroquine: 1.5e-6. This data is representative of three independent experiments each. **p**. P-values: Panc 04.03: MRTX1133 vs DMSO:2.0e-8; MRTX1133 vs MRTX1133+ apilimod:2.0e-7; MRTX1133 vs MRTX1133 + ESK981:5.3e-7; MRTX1133 vs MRTX1133+chloroquine:4.7e-6 trametinib vs DMSO:9.3e-9; trametinib vs trametinib+apilimod: 5.2e-8 trametinib vs trametinib+ESK981:7.5e-8; trametinib vs trametinib+chloroquine:2.0e-6.

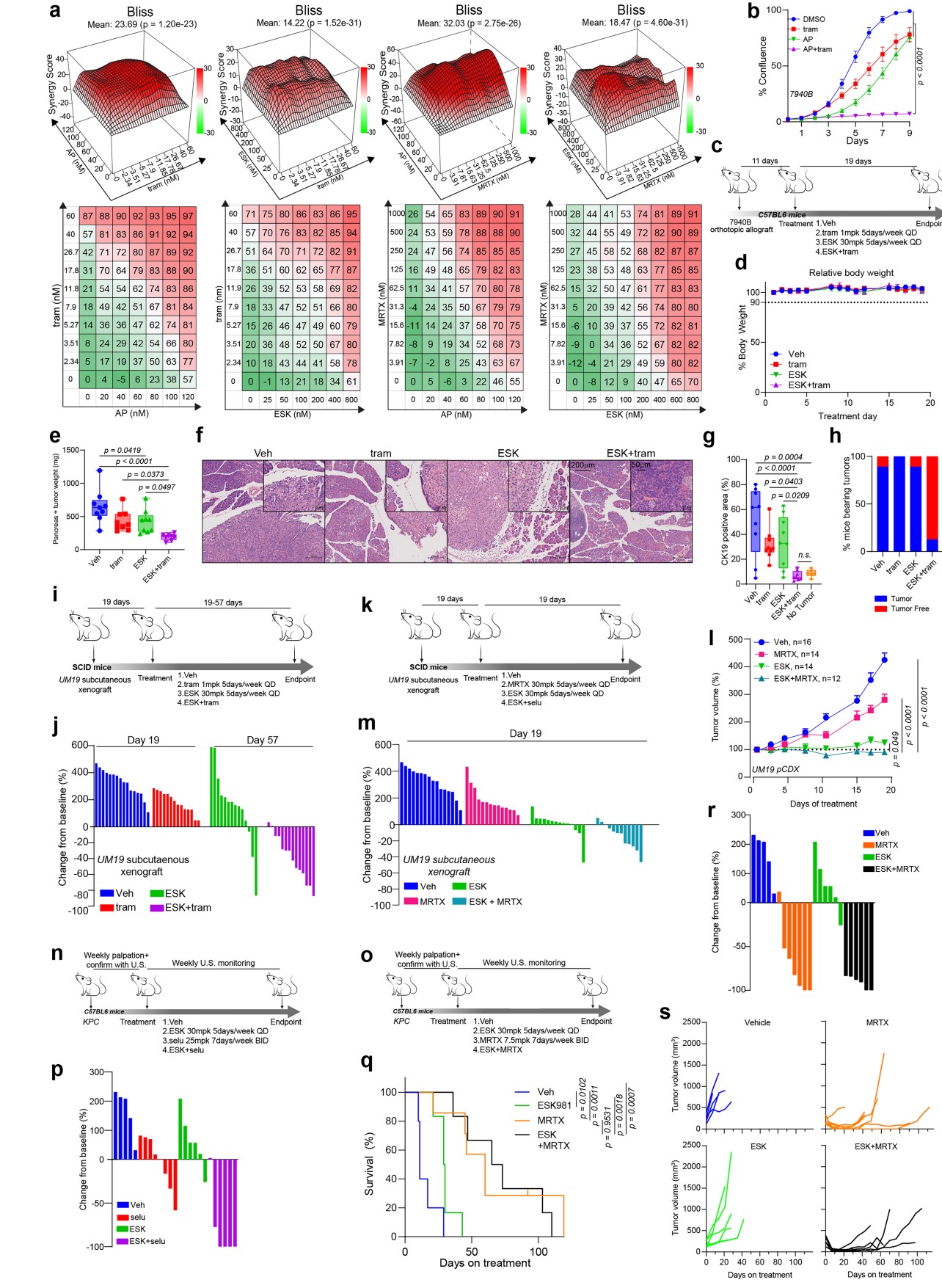

**Extended Data Fig. 12** | See next page for caption.

**Extended Data Fig. 12 | Dual inhibition of PIKfyve and KRAS-MAPK results in synergistic efficacy in PDAC cell and tumor models. a.** 3D synergy plots (top) and corresponding effect value heatmaps (bottom) in 7940B cells treated with apilimod and trametinib (left), ESK981 and trametinib (middle left), apilimod and MRTX1133 (middle right), and ESK981 and MRTX1133 (right). Red peaks in the 3D plots indicate synergism, and the overall average synergy score is listed above. **b.** Confluence assays of 7940B cells treated with DMSO, trametinib (20 nM), apilimod (50 nM), or both trametinib and apilimod. Bars are +/-SEM. (Two-way ANOVA with Dunnett's with the combination as the baseline.). **c.** Schematic outlining syngeneic orthotopic model of 7940B for C57BL/6 mice assessing in vivo efficacy of ESK981 (ESK, 30 mg/kg, QD, PO), trametinib (tram, 1 mg/kg QD, PO), or ESK981 and trametinib (ESK + tram). **d.** Relative body weight (compared to day 1) of mice harboring 7940B orthotopic tumors undergoing indicated treatment. Bars are +/-SEM. **e.** Raw pancreas + tumor weights collected at endpoint of 7940B syngeneic orthotopic model. Box represents 25th and 75th percentiles; whiskers represent the range. (One-way ANOVA with Tukey's.). **f.** Representative images of H&E of one tumor from each treatment arm from 7940B syngeneic orthotopic model. Scalebar = 200 µm for the zoomed-out H&E images; 50 µm for the zoomed-in images. **g.** Quantification of CK19 positive area compared to hematoxylin counterstain on a section from each tumor of the 7940B syngeneic orthotopic model. Box represents 25th and 75th percentiles; whiskers represent the range. (One-way ANOVA with Tukey's). **h.** Barplot of tumor presence or absence based on gross and histological evidence. **i.** Schematic outlining efficacy study using subcutaneous model of UM19 pCDX treated with vehicle, trametinib (tram, 1 mg/kg, QD, PO), ESK981 (ESK 30 mg/kg, QD, PO), or ESK981 + trametinib (ESK + tram). **j.** Waterfall plot displaying change in tumor volume at treatment end point compared to baseline of the UM19 CDX model treated with trametinib +/- ESK981. The endpoint displayed of the vehicle and trametinib arms are day 19. The endpoint displayed of the ESK981 and ESK981 +/-trametinib group are day 57. The mice in the vehicle- and ESK981- treated groups were the same mice shown in **(l-m). k.** Schematic outlining efficacy study using subcutaneous model of UM-19 primary cell-derived xenograft (CDX) treated with vehicle, MRTX1133 (MRTX, 30 mg/kg, QD, IP), ESK981 (ESK 30 mg/kg, QD, PO), or ESK981 + MRTX1133 (ESK + MRTX). **l.** Tumor volumes as a percentage of the initial volume measured by calipers over treatment course of the UM19 primary CDX (pCDX) model treated with MRTX1133 + /-ESK981. The mice in the vehicle and ESK981 groups are shared with the mice displayed in (Fig. 4g,h, Extended Data Fig. 12j), as they were all treated in the same experiment. Bars are +SEM. **m.** Waterfall plot displaying change in tumor volume at treatment end point (day 19) compared to day 1. The mice in the vehicle and ESK981 groups are shared with the mice displayed in Fig. 4g,h, and Extended Data Fig. 12j, as they were all treated in the same experiment. **n.** Schematic outlining efficacy study using KPC autochthonous model assessing the combinatorial effects of ESK981 and selumetinib (MEK inhibitor). U.S. = ultrasound. **o.** Schematic outlining efficacy study using KPC autochthonous model assessing the combinatorial effects of ESK981 and MRTX1133. **p.** Waterfall plot of maximal tumor response to each therapy compared to their starting volume. The mice in the vehicle- and ESK981- treated groups are the same mice shown in Extended Data Fig. 12q–s, as they were treated in the same experiment. **q.** Kaplan-Meier curves of survival of KPC mice undergoing indicated therapies. (two-sided log-rank test; each comparison calculated independently). The mice in the vehicle- and ESK981- treated groups are the same mice shown in Fig. 4i,j, Extended Data Fig. 12p, as they were treated in the same experiment. **r.** Waterfall plot of maximal tumor response to each therapy compared to their starting volume. The mice in the vehicle- and ESK981- treated groups are the same mice shown in Fig. 4i,j, Extended Data Fig. 12p, as they were treated in the same experiment. **s.** Tumor volumes of KPC mice treated as indicated as measured by 3D ultrasound on the indicated treatment day. The mice in the vehicle- and ESK981- treated groups are the same mice shown in Fig. 4i,j, Extended Data Fig. 12p, as they were treated in the same experiment. Mouse cartoons used in Extended Data Fig. 12c,i,k,n,o were adapted from Adobe Stock Image (Asset #304271210). **Statistics and reproducibility: a.** Each drug dose combination effect value was a result of n = 5 biological replicates. This experiment was performed thrice with apilimod + trametinib with similar results and once with the other three combinations each. **b.** n = 4 biological replicates. P-values: apilimod+ trametinib vs DMSO:1.0e-15; apilimod+trametinib vs trametinib:1.0e-15; apilimod+trametinib vs apilimod:1.0e-15. **d.** n = 9 individual animals for vehicle, ESK981, and trametinib groups and n = 8 individual animals for ESK981 + trametinib group. **e.** n = 9 individual animals for vehicle, ESK981, and trametinib groups and n = 8 individual animals for ESK981 + trametinib group. P-values: vehicle vs ESK981+trametinib:3.8e-5. **f.** These images are representative of n = 9 for vehicle, ESK981, and trametinib groups and n = 8 for ESK981 + trametinib group. **g.** n = 9 individual animals for vehicle, ESK981, and trametinib groups and n = 8 individual animals for ESK981 + trametinib group. P-value: vehicle vs ESK981 + trametinib:6.8e-5. **l.** n = 16 tumors from 8 animals for vehicle group; n = 14 tumors from 7 animals for MRTX and ESK groups; n = 12 tumors from 6 mice for the ESK + MRTX group. **q,s.** n = 5 individual animals for vehicle; n = 7 individual animals for MRTX1133; n = 6 individual animals for ESK981; and n = 6 individual animals for ESK981 + MRTX1133 groups.

# Reporting Summary

## Statistics

For all statistical analyses, confirm that the following items are present in the figure legend, table legend, main text, or Methods section.

| n/a | Confirmed | |
|---|---|---|
| ☐ | ☒ | The exact sample size (*n*) for each experimental group/condition, given as a discrete number and unit of measurement |
| ☐ | ☒ | A statement on whether measurements were taken from distinct samples or whether the same sample was measured repeatedly |
| ☐ | ☒ | The statistical test(s) used AND whether they are one- or two-sided<br>*Only common tests should be described solely by name; describe more complex techniques in the Methods section.* |
| ☐ | ☒ | A description of all covariates tested |
| ☐ | ☒ | A description of any assumptions or corrections, such as tests of normality and adjustment for multiple comparisons |
| ☐ | ☒ | A full description of the statistical parameters including central tendency (e.g. means) or other basic estimates (e.g. regression coefficient) AND variation (e.g. standard deviation) or associated estimates of uncertainty (e.g. confidence intervals) |
| ☐ | ☒ | For null hypothesis testing, the test statistic (e.g. $F$, $t$, $r$) with confidence intervals, effect sizes, degrees of freedom and $P$ value noted<br>*Give P values as exact values whenever suitable.* |
| ☒ | ☐ | For Bayesian analysis, information on the choice of priors and Markov chain Monte Carlo settings |
| ☒ | ☐ | For hierarchical and complex designs, identification of the appropriate level for tests and full reporting of outcomes |
| ☒ | ☐ | Estimates of effect sizes (e.g. Cohen's *d*, Pearson's *r*), indicating how they were calculated |

*Our web collection on statistics for biologists contains articles on many of the points above.*

## Software and code

Policy information about availability of computer code

Data collection | No software was used for data collection.

Data analysis | Computational tools used:
Graphpad Prism 10 and included statistical tools
Fiji (ImageJ) (version 1.54i)
Imagestudio (version 5.2)
Tecan i-control (version 2)
SynergyFinder+ Web application 3.14.0
Incucyte 2023A Rev2 (2023.3.3.0 Rev2)
Seahorse Wave Controller Software (version2.6.3.5)
R (version 4.3.2)
cutadapt (version 4.1)
bowtie2 (version 2.4.5)
MAGeCK (version 0.5.9.5)
ggplot2 (version 3.4.4)
Quantstudio Design&Analysis Software (version 1.5.2)
bcl2fastq (version 2.20)
kallisto (version 0.50.1)
limma (version 3.58.1)
edgeR (version 4.0.16)
fgsea (version 1.28.0)

Masshunter Workstation Quantitative Analysis for QQQ (version 10.1 Build 10.1.733.0)
Morpheus Matrix Visualization and analysis tool web application (no version information found)
Masshunter Quantitative Analysis (version 12.0)
Mass Profiler Professional (version 15.0)
lme4 (version 1.1-35.1)
Trimmomatic 0.39
Picard MarkDuplicates 2.26.0
samtools 1.9
bedtools 2.27.1
MACS2 2.2.7
wigToBigWig 2.4
FIMO 5.5.6
HOMER 5.1
BWA MEM 0.7.17

For manuscripts utilizing custom algorithms or software that are central to the research but not yet described in published literature, software must be made available to editors and reviewers. We strongly encourage code deposition in a community repository (e.g. GitHub). See the Nature Portfolio guidelines for submitting code & software for further information.

## Data

Policy information about availability of data

All manuscripts must include a data availability statement. This statement should provide the following information, where applicable:
- Accession codes, unique identifiers, or web links for publicly available datasets
- A description of any restrictions on data availability
- For clinical datasets or third party data, please ensure that the statement adheres to our policy

All source data and raw gel images are included with the manuscript. All materials are available from the authors upon reasonable request. All raw next-generation sequencing, such as DNA sequencing for the CRISPR screen and ChIP-seq or RNA-seq, have been deposited in the Gene Expression Omnibus (GEO) repository at NCBI with the accession numbers GSE255378 and GSE277832. Processed sequencing data, such as sgRNA counts and RNA-seq CPM, are included as Supplementary Table 1 and 3, respectively.  Raw data for metabolomics and lipidomics experiments are included as Supplementary Tables (4 and 5, respectively). Other publicly available datasets were used: human genome assembly GRCh38: https://www.ncbi.nlm.nih.gov/datasets/genome/GCF_000001405.26/; murine genome assembly: GRCm38: https://www.ncbi.nlm.nih.gov/datasets/genome/GCF_000001635.20/, Hallin et al., 2019: GSE201412.

## Research involving human participants, their data, or biological material

Policy information about studies with human participants or human data. See also policy information about sex, gender (identity/presentation), and sexual orientation and race, ethnicity and racism.

| | |
|---|---|
| Reporting on sex and gender | Of the five patient samples from the University of Michigan pathological archives included in this study, four were from male patients and one was from a female patient. Gender was not reported. Sex and gender were not considered as variables in this study design. |
| Reporting on race, ethnicity, or other socially relevant groupings | All samples were from White, Non-Hispanic patients at the University of Michigan. |
| Population characteristics | There was no human population research in this study. |
| Recruitment | Patient tissues from biopsies of pancreatic tumors were acquired from the University of Michigan pathology archives. |
| Ethics oversight | Use of clinical formalin-fixed paraffin embedded specimens from the archives was approved by the University of Michigan Institutional Review Board and does not require patient consent. |

Note that full information on the approval of the study protocol must also be provided in the manuscript.

# Field-specific reporting

Please select the one below that is the best fit for your research. If you are not sure, read the appropriate sections before making your selection.

☒ Life sciences        ☐ Behavioural & social sciences        ☐ Ecological, evolutionary & environmental sciences

For a reference copy of the document with all sections, see nature.com/documents/nr-reporting-summary-flat.pdf

# Life sciences study design

All studies must disclose on these points even when the disclosure is negative.

| | |
|---|---|
| Sample size | Sample size was determined by preliminary studies and the level of the observed effect. |

| Data exclusions | Two datapoints were removed from lipidomics experiments as being statistical outliers. These are explicitly stated in the figure legends. No other data exclusions were made. |
|---|---|
| Replication | Independent experimental replicates for each experiment are explicitly stated in figure legends. In all instances, all attempts at replicating the experiments produced similar results. |
| Randomization | For animal studies, mice were randomly assigned to treatment groups. For all in vitro experiments, we used a common cell suspension to plate for both control and treatment groups. For retrospective pathological analyses or human samples, matched control and experimental sections were taken from all samples and thus did not require randomization. |
| Blinding | All histo-pathological evaluations of tissues and IHC/RNA-ISH-based scoring were carried out in a blinded manner by two independent pathologists. For animal studies, the researcher was blinded to group when measuring tumors. For all other experiments, the analyses did not require blinding as data quantification was carried out using instruments and automated workflows. |

# Reporting for specific materials, systems and methods

We require information from authors about some types of materials, experimental systems and methods used in many studies. Here, indicate whether each material, system or method listed is relevant to your study. If you are not sure if a list item applies to your research, read the appropriate section before selecting a response.

## Materials & experimental systems

| n/a | Involved in the study |
|---|---|
| ☐ | ☒ Antibodies |
| ☐ | ☒ Eukaryotic cell lines |
| ☒ | ☐ Palaeontology and archaeology |
| ☐ | ☒ Animals and other organisms |
| ☒ | ☐ Clinical data |
| ☒ | ☐ Dual use research of concern |
| ☒ | ☐ Plants |

## Methods

| n/a | Involved in the study |
|---|---|
| ☐ | ☒ ChIP-seq |
| ☒ | ☐ Flow cytometry |
| ☒ | ☐ MRI-based neuroimaging |

## Antibodies

| Antibodies used | Human PIKfyve Antibody R&D Systems AF7885<br>SQSTM1/p62 (D5L7G) Mouse mAb Cell Signaling Technology 88588S<br>Cleaved PARP (Asp214) (D64E10) XP® Rabbit mAb  Cell Signaling Technology 5625S<br>SQSTM1/p62 (D6M5X) Rabbit mAb Cell Signaling Technology 23214s<br>LC3A/B (D3U4C) XP® Rabbit mAb  Cell Signaling Technology 12741S<br>Vinculin (E1E9V) XP® Rabbit mAb (HRP Conjugate)  Cell Signaling Technology 18799S<br>GAPDH (14C10) Rabbit mAb (HRP Conjugate) Cell Signaling Technology 3683S<br>Recombinant Anti-Fatty Acid Synthase antibody [EPR7466] Abcam ab128856<br>Phospho-Acetyl-CoA Carboxylase (Ser79) (D7D11) Rabbit mAb Cell Signaling Technology 11818S<br>Acetyl-CoA Carboxylase (C83B10) Rabbit mAb Cell Signaling Technology 3676S<br>Anti-SREBP1 antibody Abcam ab28481<br>Histone H3 (96C10) Mouse mAb Cell Signaling Technology 3638S<br>Phospho-p44/42 MAPK (Erk1/2) (Thr202/Tyr204) (D13.14.4E) XP® Rabbit mAb Cell Signaling Technology 4370S<br>p44/42 MAPK (Erk1/2) (137G5) Cell Signaling Technology 4695S<br>Amersham ECL Peroxidase (HRP) Anti-Mouse IgG F(ab')2 Fragment Sheep Secondary Antibody Fisher/GE Healthcare NA931<br>Sheep IgG Horseradish Peroxidase-conjugated Antibody R&D Systems HAF016<br>anti-Rabbit IgG, peroxidase-linked species-specific whole antibody (from donkey) Secondary Antibody, Cytiva GE Healthcare UK Limited/Fisher NA934<br>Anti-Cytokeratin 19 antibody [EP1580Y] Abcam ab52625<br>CONFIRM Anti-Ki67 antibody Ventana 790-4286<br>Anti-Insulin (C27C9) Rabbit mAb Cell Signaling Technology 3014S<br>Anti-c-Myc antibody [Y69] - ChIP Grade Abcam ab32072<br>Anti-SPTLC1 antibody Abcam ab176706<br>SPTLC2 Polyclonal Antibody Invitrogen PA5-21142<br>AMPKα Antibody #2532 Cell Signaling Technology 2532S<br>Atg5 (D5F5U) Rabbit mAb #12994 Cell Signaling Technology 12994S<br>Atg7 (D12B11) Rabbit mAb #8558 Cell Signaling Technology 8558S<br>Anti-LDL Receptor antibody [EP1553Y] (ab52818) Abcam ab52818<br>XP® Anti-LAMP1 Rabbit Monoclonal Antibody [D2D11] Cell Signaling Technology 9091S<br>Goat Anti-Rabbit IgG Antibody (Alexa Fluor® 594) Jackson ImmunoResearch 111-585-045 |
|---|---|
| Validation | Human PIKfyve Antibody WB validated; human<br>SQSTM1/p62 (D5L7G) Mouse mAb KO validated; human, mouse, rat, monkey<br>Cleaved PARP (Asp214) (D64E10) XP® Rabbit mAb  drug validated; human, mouse, monkey<br>SQSTM1/p62 (D6M5X) Rabbit mAb KO validated; mouse, rat<br>LC3A/B (D3U4C) XP® Rabbit mAb  drug validated; human, mouse, rat |

Vinculin (E1E9V) XP® Rabbit mAb (HRP Conjugate)  WB validated; human, mouse, rat, monkey, dog
GAPDH (14C10) Rabbit mAb (HRP Conjugate) WB validated; human, mouse, rat, monkey, bovine, pig
Recombinant Anti-Fatty Acid Synthase antibody [EPR7466] KO validated; human, mouse, rat
Phospho-Acetyl-CoA Carboxylase (Ser79) (D7D11) Rabbit mAb drug validated; human, mouse, rat
Acetyl-CoA Carboxylase (C83B10) Rabbit mAb AB dilution validated; human, mouse, rat, hamster
Anti-SREBP1 antibody WB validated; human, mouse, rat
Histone H3 (96C10) Mouse mAb WB validated; human, mouse, rat, monkey, zebrafish, bovine, pig
Phospho-p44/42 MAPK (Erk1/2) (Thr202/Tyr204) (D13.14.4E) XP® Rabbit mAb drug validated; human, mouse, rat, hamster, monkey, mink, D. melanogaster, zebrafish, bovine, dog, pig, S. cerevisiae
p44/42 MAPK (Erk1/2) (137G5) siRNA validated; human, mouse, rat, hamster, monkey, mink, D. melanogaster, zebrafish, bovine, dog, pig, C. elegans
Anti-Cytokeratin 19 antibody [EP1580Y] WB validated; human, mouse
CONFIRM Anti-Ki67 antibody Validated for IHC
Anti-Insulin (C27C9) Rabbit mAb AB dilution validated; human, mouse, rat
Anti-c-Myc antibody [Y69] - ChIP Grade drug validated; human, mouse, rat
Anti-SPTLC1 antibody WB validated; human, mouse
SPTLC2 Polyclonal Antibody KD verified; human, mouse
AMPKα Antibody #2532 WB validated; human, mouse, rate, hamster, monkey
Atg5 (D5F5U) Rabbit mAb #12994 KO verified; human, mouse, rat
Atg7 (D12B11) Rabbit mAb #8558 siRNA validated; human, mouse, rat

# Eukaryotic cell lines

Policy information about cell lines and Sex and Gender in Research

| Cell line source(s) | PANC-1, MIA PaCa-2, Panc 04.03, SW1990, Panc 10.05, and HPAF-II  were obtained from American Type Culture Collection. 7940B was generously provided by Greggory Beatty, M.D., Ph.D. at Perlman School of Medicine at the University of Pennsylvania. The iKRAS 9805 cell line was previously described and was provided by Dr. Marina Pasca di Magliano at the University of Michigan. The UM PDAC primary cell cultures lines (UM2, UM19) were obtained from surgically resected samples from female patients and established through murine xenograft. KPC-1344 and KPC-1361 cells were derived from female KPC mice in-house according to described methods. |
| --- | --- |
| Authentication | All human cell lines were authenticated with STR genetic testing |
| Mycoplasma contamination | All cell lines were biweekly tested to be free of mycoplasma contamination. |
| Commonly misidentified lines (See ICLAC register) | None |

# Animals and other research organisms

Policy information about studies involving animals; ARRIVE guidelines recommended for reporting animal research, and Sex and Gender in Research

| Laboratory animals | For xenograft studies, 6-8-week-old CB17 severe combined immunodeficiency (SCID) mice obtained from the University of Michigan breeding colony were used. For syngeneic studies, 6-8-week-old C57BL6 mice obtained from Jackson Laboratories were used. For the autochthonous model, KPC animals were studied beginning at 10 weeks of age. For the prophylactic study, KPC animals were studied beginning at 6 weeks of age. For the Pikfyve GEMM studies, animals were studied for survival throughout their lifespan. Cohorts were taken for histological analyses at the indicated timepoints in the manuscript. |
| --- | --- |
| Wild animals | None used. |
| Reporting on sex | Both male and female mice were used throughout the study. Differences in sex were not observed. |
| Field-collected samples | None. |
| Ethics oversight | Experiments involving the Pdx1-Cre animals for the autochthonous model efficacy study were approved by the University of Glasgow Animal Welfare and Ethical Review Board and were performed under a UK Home Office license. All other animals used in this study were housed at the University of Michigan in a pathogen-free environment, and all procedures involving these animals were performed in accordance with requirements of the University of Michigan Institutional Animal Care & Use Committee (IACUC). |

Note that full information on the approval of the study protocol must also be provided in the manuscript.

# Plants

| | |
|---|---|
| Seed stocks | Not applicable |
| Novel plant genotypes | Not applicable |
| Authentication | Not applicable |

# ChIP-seq

## Data deposition

☒ Confirm that both raw and final processed data have been deposited in a public database such as GEO.

☒ Confirm that you have deposited or provided access to graph files (e.g. BED files) for the called peaks.

**Data access links**
*May remain private before publication.*

https://www.ncbi.nlm.nih.gov/geo/query/acc.cgi?acc=GSE277832 token: uruxgeokppaxnwx

**Files in database submission**

mctp_SI_37818_H5F2HDRX5_1_1.fq.gz
mctp_SI_37818_H5F2HDRX5_1_2.fq.gz
mctp_SI_37818_H5F2HDRX5_2_1.fq.gz
mctp_SI_37818_H5F2HDRX5_2_2.fq.gz
mctp_SI_37819_H5F2HDRX5_1_1.fq.gz
mctp_SI_37819_H5F2HDRX5_1_2.fq.gz
mctp_SI_37819_H5F2HDRX5_2_1.fq.gz
mctp_SI_37819_H5F2HDRX5_2_2.fq.gz
mctp_SI_37820_H5F2HDRX5_1_1.fq.gz
mctp_SI_37820_H5F2HDRX5_1_2.fq.gz
mctp_SI_37820_H5F2HDRX5_2_1.fq.gz
mctp_SI_37820_H5F2HDRX5_2_2.fq.gz
mctp_SI_37821_H5F2HDRX5_1_1.fq.gz
mctp_SI_37821_H5F2HDRX5_1_2.fq.gz
mctp_SI_37821_H5F2HDRX5_2_1.fq.gz
mctp_SI_37821_H5F2HDRX5_2_2.fq.gz
miapaca2_dmso_igg_peaks.bed
miapaca2_dmso_myc_peaks.bed
miapaca2_amg510_myc_peaks.bed
miapaca2_tram_myc_peaks.bed
miapaca2_dmso_igg.bw
miapaca2_dmso_myc.bw
miapaca2_amg510_myc.bw
miapaca2_tram_myc.bw

**Genome browser session**
(e.g. UCSC)

N/A

## Methodology

**Replicates**

One sample per condition

**Sequencing depth**

reads are 151-base, paired end
miapaca2_dmso_igg: raw reads: 26596872, uniquely mapped reads: 18164562
miapaca2_dmso_myc: raw reads: 29052973, uniquely mapped reads: 18490903
miapaca2_amg510_myc: raw reads: 30297035, uniquely mapped reads: 18729003
miapaca2_tram_myc: raw reads: 27112748, uniquely mapped reads: 17822895

**Antibodies**

Anti-c-Myc antibody [Y69] - ChIP Grade (ab32072)

**Peak calling parameters**

Trimmed, paired reads were aligned to the human reference genome (hg38, obtained from UCSC) using bwa mem 0.7.17 with options -5 -S -P -T 0. Alignments were filtered for mapping quality >= 20 using samtools 1.9. Read duplicates were removed using Picard MarkDuplicates 2.26.0. Non-primary alignments were removed using samtools view (option -F 0x900) and converted to BED format using bedtools bamtobed 2.27.1. Peaks were called from these alignments using MACS2 2.2.7.1 with default settings (peaks filtered at q < 0.05) and the -B option to generate bedGraph coverage files. Peaks were then filtered using the ENCODE Unified GRCh38 Exclusion List (https://www.encodeproject.org/files/ENCFF356LFX/). Coverages captured by MACS2 were converted to

bigWig using wigToBigWig 2.4.

Data quality

Quality was checked using the following parameters which yielded the following results:
miapaca2_dmso_igg: peaks (q < 0.05): 2966, peaks (q < 0.05 and enrichment > 5): 821
miapaca2_dmso_myc: peaks (q < 0.05): 9552, peaks (q < 0.05 and enrichment > 5): 3355
miapaca2_amg510_myc: peaks (q < 0.05): 10378, peaks (q < 0.05 and enrichment > 5): 2279
miapaca2_tram_myc: peaks (q < 0.05): 5431, peaks (q < 0.05 and enrichment > 5): 1691

Software

Basecalling: bcl2fastq 2 (Illumina)
Trimming: Trimmomatic 0.39
Alignment: BWA MEM 0.7.17
Alignment manipulation/filtering: samtools 1.9
Duplicate identification: Picard MarkDuplicates 2.26.0
BED file manipulation: Bedtools 2.27.1
Peak calling: MACS2 2.2.7.1
Coverage generation: MACS2 2.2.7.1 and wigToBigWig 2.4
Motif analysis: FIMO 5.5.6 and HOMER 5.

