## [Peer Review File · Nature]

Targeting PIKfyve-driven lipid metabolism in pancreatic cancer

Corresponding Author: Dr Arul Chinnaiyan

Version 0:

Reviewer comments:

Referee #1

(Remarks to the Author)

In this manuscript, Cheng et al. identify PIKfyve -a lipid kinase with key roles in lysosomal function- as a differentially upregulated gene in pre-neoplastic pancreatic lesions (PanINs) and overt pancreatic ductal adenocarcinoma (PDAC). Its genetic loss, protein degradation or enzymatic inhibition leads to decreased autophagy and induction of fatty acid synthesis in PDAC cells. It also suppresses early tumor formation and causes tumor regression in genetically engineered and transplant models of mouse or human PDAC. The authors identify PIKfyve inhibition as a synthetic lethality in the context of KRAS-MAPK signaling inhibition. This is proposed to occur because KRAS signaling is a driver of de novo fatty acid synthesis -a dependency when PIKfyve is inhibited-; Conversely, KRAS signaling inhibition induces autophagy, which is suppressed when PIKfyve is inhibited.

The anti-PDAC synergistic effects of concomitant inhibition of KRAS-MAPK and PIKfyve is novel. The implications are also highly translational; it is evident that this will lead to clinical trials.

However, despite use of multiple approaches to highlight the critical role of PIKfyve in PDAC but not normal pancreas and the potential translational impact of PIKfyve inhibition, the findings in this manuscript are highly descriptive and seem better fit for a translational journal readership. Mechanisms are lacking throughout.

It is unclear how/why PIKfyve is upregulated in PDAC; it is unclear how it activates de novo fatty acid synthesis through SREBP-1?; it is unclear how its inhibition suppresses autophagy in PDAC or what role autophagy plays in connection with induction of fatty acid synthesis?

Moreover, how does KRAS-MAPK signaling cross-talk with PIKfyve at both levels of regulation of autophagy and fatty acid synthesis?

And can these questions be addressed in the context of PDAC versus normal pancreatic tissue?

Minor comments:

-Autophagy detection is not thorough; Interpretation of data in Figures 3B and 4D are conflicting; according to the authors, in 3B: the ratio of LC3B II/I increases and is indicative of decreased autophagy; in 4D: the ratio of II/I is however also increased upon KRAS extinction (minus Dox) and this is interpreted as autophagy induction (lines 381-384), with p62 levels being decreased in parallel.

-Subcutaneous transplants are not considered good models for PDAC, as they do not faithfully recapitulate the stroma-dense, nutrient deprived/hypoxic environment nor do they recapitulate PDAC metabolism (Figures 2, 4 and Extended Data Fig. 2).

-The Discussion section is repetitive, re-stating and summarizing the already well-described results; would be best to use to highlight limitations, future steps of the study etc.

-It is in general good practice to avoid use of absolute language when describing findings; e.g, "never" (line 130) and "first" (lines 449 and 469) -the authors do mention in the discussion a report involving PIKfyve in lipid metabolism through its lysosomal function(line 471).

Referee #2

(Remarks to the Author)

Combining in vivo and in vitro approaches, the authors identify a key role for PIKfyve in PDAC progression. In addition, by applying a CRISPR-based screen they uncover synthetic lethality of de novo lipid synthesis created by PIKfyve depletion linked to the upregulation of key lipid synthetic enzymes (FASN and ACACA) induced by PIKfyve depletion/inhibition. The authors also uncover that KRAS-MAPK inhibition counteracts the increased expression of FASN and ACACA induced by PIKfyve depletion/inhibition and that combining PIKfyve and KRAS-MAPK inhibition synergistically suppresses PDAC growth.

Overall, the data presented in the manuscript are of high quality and fully support the conclusions. Each of the pathways explored by the authors, i.e. PIKfyve and RAS-MAPK, has a pivotal and widely recognized role in PDAC progression to the point that, as the authors mention, there are two clinical trials testing the efficacy of combined treatment of MAPK inhibitors with the autophagy flux inhibitor cloroquine, and an ongoing phase II clinical trial testing the efficacy of the PIKfyve inhibitor ESK981 in pancreatic cancer patients.

The original and important finding of the manuscript consists in the discovery that PIKfyve inhibition reshuffles the lipogenic transcriptional program of PDAC cells making their growth dependent on de novo fatty acid synthesis and that RAS-MAPK has a role in sustaining the de novo lipid biosynthesis.

However, the enthusiasm is tempered by the lack of analysis of the molecular mechanisms underpinning the lipid program reshuffling and of the possible molecular links bridging PIKfyve-dependent and MAPK-dependent pathways of lipid metabolism control. Also, the very intriguing alteration of sphingolipid metabolism is left completely unexplored as regards possible molecular mechanisms.

Along the same lines, the authors did not consider that PIKfyve controls not only autophagy but also many different trafficking steps along the endocytic pathway and that these might have an impact on signaling circuits that affect lipid metabolism programmes. In this context, it is worth mentioning that the upregulation of FASN and ACA transcripts induced by PIKfyve depletion/inhibition had already been reported in muscle (PMID: 23673157) and that, in this case, the deregulation of lipid metabolism was found to be dependent on altered glut4 trafficking to the plasma membrane and a possible consequent activation of CHREB-dependent targets. In the present manuscript, the nature of possible signaling circuits and/or transcription factors involved are left unaddressed.

Referee #3

(Remarks to the Author)

Re: Targeting PIKfyve-driven lipid homeostasis as a metabolic vulnerability in pancreatic cancer

Summary

This manuscript describes a novel role for the PI kinase PIKfyve in pancreatic cancer. It begins with the finding that PIKfyve expression is higher in PDAC lesions compared to adjacent pancreas in mouse models of PDAC and human PDAC patients. The authors cross two different genetic PDAC driver models, KC and KPC, with PIKfyve knock out mice and find that animals lacking PIKfyve develop fewer PDAC lesions and have decreased tumor burden compared to their wild-type littermates. They also find that loss of PIKfyve does not impact the development of normal pancreas. Small molecule inhibitors of PIKfyve elicit similarly striking results in these models. I commend the authors on a rigorous characterization of PIKfyve treatment in PDAC models in the first two figures of this paper.

The authors next find that PIKfyve inhibition disrupts normal lysosomal trafficking in PDAC models. A previous publication characterized the role of PIKfyve in lysosomal trafficking and the authors note that PDAC is a disease that is particularly vulnerable to this type of manipulation. An in vitro CRISPR screen of PDAC cell lines treated with a PIKfyve inhibitor revealed that genes specifically involved in de novo lipogenesis (FASN, ACACA, and ELOVL1) are essential for treatment survival. As validation, the authors perform a CRISPR knockdown of these genes and find they indeed synergize with PIKfyve inhibitor treatment to kill PDAC cells.

Water-soluble metabolomics experiments show global decreases in glycolytic metabolites (G6P, F16BP, PEP, pyruvate) after treatment with PIKfyve inhibitors. Meanwhile, the authors observe differences in some TCA cycle metabolites: succinate and aconitate are increased while alpha ketoglutarate and citrate decreased. These observations imply that PDAC cells are using other metabolites such as glutamine to fuel the TCA cycle in response to PIKfyve inhibitor treatment, but this point is beyond the scope of this paper. These data lead the authors to conclude that PDAC cells are shunting carbon atoms away from glycolysis and toward de novo lipogenesis, as suggested by the results of their CRISPR screen. Global lipidomics reveals that sphingolipids are dramatically increased upon PIKfyve inhibitor treatment, likely to compensate for increased de novo palmitic acid synthesis. Experiments in the final figure of this paper show that treatment with a PIKfyve inhibitor and a KRAS inhibitor synergistically kills PDAC cells. The authors imply this combined treatment paradigm is effective because PIKfyve inhibition and KRAS inhibition have opposite effects on lipogenesis and that creates a metabolic conflict in PDAC cells.

Outstanding Questions.

1. The biggest outstanding question for this manuscript is: why do PDAC cells upregulate de novo lipogenesis when PIKFyve activity is disrupted? The observation that KRAS-MAPK controls de novo lipogenesis is interesting and presents an outstanding treatment paradigm when combined with PIKFyve inhibitors. However, the synergistic effect of PIKFyve inhibition and KRAS inhibition on both lipogenesis and survival implies that these two pathways are independently regulating lipogenesis. Additionally, their CRISPER validation screen shows that silencing genes in the de novo lipogenesis pathway potentiates the lethality of PIKFyve inhibitors. Together, these observations imply that upregulation of de novo lipogenesis is a potential survival mechanism for cells treated with PIKFyve inhibitors. Do the authors have thoughts on why lipogenesis increases with PIKFyve inhibition and what survival advantage this gives to PDAC cells? Do the authors have any experiments planned to explore this mechanism?

2. On a related note, increased lipogenesis suggests an overall change in the energy demands of the cell. Have the authors performed experiments to look at the global metabolic state of the cell with markers like AMPK phosphorylation after PIKFyve inhibitor treatment?

3. From the metabolomics data, the authors propose that glycolytic metabolites are being shunted to de novo lipogenesis. Lipidomics analysis shows that sphingolipids are significantly increased in response to PIKFyve inhibitor treatment. The authors imply that this response, presumably mediated by the activity of the enzyme SPT1, compensates for increased de novo palmitate synthesis. Have the authors considered validating this claim with glucose tracing experiments into sphingolipids? This may be a good way to shed light on mechanistically how PDAC cells respond to PIKFyve inhibitor treatment.

4. If upregulation of sphingolipid biosynthesis is an important survival mechanism for cells treated with PIKFyve inhibitors, why wasn't SPT1 identified as essential in their CRISPR screen? If cells are shunting palmitic acid produced through de novo lipogenesis into sphingolipids for survival, elimination of SPT1 should be lethal to these cells. The observation that this enzyme did not fall out of their screen suggests there may be another fate of palmitate in their system. Have the authors explored this possibility further with additional experiments? Does SPT1 knock down potentiate the lethality of PIKFyve inhibition? Do other lipid classes, such as triglycerides, change with treatment to protect PDAC cells from higher levels of palmitate?

5. The final figure of the paper leads the authors to hypothesize that since KRAS inhibition and PIKFyve inhibition have opposite effects on lipogenesis, inhibiting both pathways synergistically kills PDAC cells. However, no lipidomics data are provided for the combined treatment conditions. Do the authors have this data? If not, they should perform this experiment.

Overall thoughts.

My opinion of this paper is very positive. It is very well written, and the authors do an excellent job of walking the reader through the logic of their experiments and interpreting their results and conclusions. The first two figures describing the effects of PIKFyve inhibition on PDAC growth are particularly noteworthy. The authors do a rigorous job characterizing the role of PIKFyve in a variety of PDAC models and leave the reader with little doubt about the importance of their findings. The final figure of the paper is also exciting: PIKFyve inhibition alongside the MAPK inhibitor MRTX1133 showed dramatic decreases in tumor growth in vivo. If these results remain consistent in the treatment of patients, their findings represent a novel and important step in the treatment of an aggressive disease with often very few treatment options.

In my opinion, the data quality for this paper is very high. The manuscript contains good descriptions and references to the experimental approaches used. Most of their conclusions seem reasonable. However, they need to conduct more metabolic experiments to support their conclusions on the role of lipid metabolism in response to PIKFyve inhibitor treatment. Overall, I think this is a great paper and should be published in Nature as a revised manuscript addressing the metabolic mechanistic questions I raised.

Version 1:

Reviewer comments:

Referee #1

(Remarks to the Author)

The authors have adequately addressed all my comments and concerns. Thank you for your diligence.

Referee #2

(Remarks to the Author)

In revising manuscript, the authors have attempted to address the major criticisms raised by this reviewer regarding the lack of mechanistic insight into the lipid metabolism reprogramming induced by PIKFyve inhibition. In response, the authors demonstrate that SREBP activation not only accompanies PIKFyve inhibition (as reported in the original manuscript) but is also necessary for the lipid reshuffling that follows. While this finding identifies an important component of the pathway, the molecular mechanism underlying SREBP activation in response to PIKFyve inhibition remains unexplored.

The authors hypothesize that lysosomal dysfunction plays a role in this activation, citing the work by Hosios et al. to support

their hypothesis. However, it is crucial to clarify that Hosios et al. do not show that lysosome inhibition activates de novo fatty acid (FA) synthesis. Instead, their study demonstrates that, only under conditions of mTORC inhibition, lysosomal activity is necessary to promote triglyceride (TG) synthesis by increasing FA release, while lysosome inhibition prevents this FA release in response to mTORC inhibition. Additionally, Hosios et al. show that mTORC1 inhibition suppresses SREBP activity (which is expected, as mTORC1 is a major inducer of SREBP-mediated lipid remodelling), but they do not provide evidence that lysosomal inhibitors activate SREBP under conditions of mTORC1 inhibition.

In the revised manuscript, the authors do present data showing that lysosomal inhibitors, such as chloroquine, activate SREBP, and use this correlation to suggest that PIKfyve inhibition may operate through a similar mechanism. However, this still leaves the molecular link between lysosomal dysfunction and SREBP activation unresolved. Further studies are needed to elucidate the specific connection between PIKfyve inhibition, lysosomal dysfunction, and SREBP activation, as this remains a critical gap in the mechanistic understanding of lipid remodelling in this context.

Further mechanistic studies are needed to establish how these two events are connected and to determine whether PIKfyve inhibition directly influences SREBP activation. Clarifying the nature of this link is particularly important, given that the authors also find SREBP activation to be necessary for sphingolipid remodelling. A deeper understanding of how SREBP activation is triggered in response to PIKfyve inhibition would provide valuable insight into the broader lipid metabolism reprogramming observed in this study.

Referee #3

(Remarks to the Author)

The authors have done an outstanding job addressing the comments made by myself and the other reviewers. I highly recommend the revised manuscript for publication in Nature.

Version 3:

Reviewer comments:

Referee #2

(Remarks to the Author)

The authors have satisfactorily addressed my comments.

Response to Referees for Manuscript 2024-03-05156

We are truly grateful for the time and effort put forth by the Editors and Reviewers at *Nature*. We have addressed all the experimental and textual concerns raised, and we believe that the helpful comments have allowed us to considerably strengthen the conclusions presented in the accompanying manuscript.

Three key highlights summarizing our updates in this revised study follow:

- We provide data that deepens our understanding of the impact of PIKfyve inhibition on lipid metabolism. This data includes demonstrating the causative role of SREBP1 activation in the lipidomic reprogramming of PDAC cells upon PIKfyve inhibition, conclusively demonstrating that PIKfyve inhibition stimulates increased utilization of glucose for synthesis of sphingolipids and showing that sphingolipid synthesis is an adaptive mechanism of PDAC in response to PIKfyve loss.
- We strengthen our mechanistic model of synthetic lethality between KRAS-MAPK and PIKfyve. For example, we show that KRAS-MAPK inhibition decreases *FASN* expression through decreased MYC transcriptional activity. We also show that KRAS-MAPK inhibition has opposing effects on the lipid profile as PIKfyve inhibition, and that dual inhibition results in a blunted adaptive response in the lipid profile of PDAC cells. This was also specifically exemplified in that KRAS-MAPK inhibition decreased sphingolipids, and that the dual inhibition of KRAS-MAPK and PIKfyve resulted in a sphingolipid level that is significantly lower than that of PIKfyve inhibition alone.
- We employed the gold-standard KPC autochthonous model and 3D ultrasound monitoring to assess the efficacy of combining KRAS-MAPK inhibitors with PIKfyve inhibitors *in vivo*. We demonstrate that the combination of a MEK inhibitor selumetinib with ESK981 resulted in 100% regression in 4 out of 6 mice and extended median survival by over 5.5 times compared to vehicle treatment. ESK981 also enhanced the antitumor efficacy of the KRAS^{G12D} inhibitor MRTX1133. To our knowledge, these effects are among the most dramatic effects reported in this model.

Referees' comments:

Referee #1 (Remarks to the Author):

In this manuscript, Cheng et al. identify PYKfyve -a lipid kinase with key roles in lysosomal function- as a differentially upregulated gene in pre-neoplastic pancreatic lesions (PanINs) and overt pancreatic ductal adenocarcinoma (PDAC). Its genetic loss, protein degradation or enzymatic inhibition leads to decreased autophagy and induction of fatty acid synthesis in PDAC cells. It also suppresses early tumor formation and causes tumor regression in genetically engineered and transplant models of mouse or human PDAC. The authors identify PYKfyve inhibition as a synthetic lethality in the context of KRAS-MAPK signaling inhibition. This is proposed to occur because KRAS signaling is a driver of de novo fatty acid synthesis -a dependency when PYKfyve is inhibited-; Conversely, KRAS signaling inhibition induces autophagy, which is suppressed when PYKfyve is inhibited.

The anti-PDAC synergistic effects of concomitant inhibition of KRAS-MAPK and PYKfyve is novel. The implications are also highly translational; it is evident that this will lead to clinical trials.

Response: We thank the reviewer for the thorough summary of our findings and also highlighting the novelty of our project. We also appreciate the reviewer's comment that the implications of our findings are highly translational in nature.

However, despite use of multiple approaches to highlight the critical role of PYKfyve in PDAC but not normal pancreas and the potential translational impact of PYKfyve inhibition, the findings in this manuscript are highly descriptive and seem better fit for a translational journal readership. Mechanisms are lacking throughout.

Response: We are grateful to the reviewer bringing up multiple interesting mechanistic questions. By addressing these inquiries, herein, we provide new mechanistic insights into the regulation of lipid metabolism in PDAC, which have significantly improved the quality of this study. All points have been addressed in a point-wise fashion below, and all significant changes to the manuscript are colored in light blue.

1. It is unclear how/why PYKfyve is upregulated in PDAC

Response: We thank the reviewer for raising this question and we agree this is an interesting phenomenon that deserves more study. While, in general, the transcriptional regulation of *PIKFYVE* remains unclear in current literature, we tested several hypotheses that shed light on this question.

First, we found that *IKZF1* (also known as Ikaros) was found to be a potential repressor of *PIKFYVE* (PMID: 33467550), and that *IKZF1* was known to be hypermethylated in pancreatic tumors (PMID: 31492175). This suggested the possibility that the hypermethylation of *IKZF1*

could be causing a higher expression of *PIKFYVE* in PDAC. To test this, we induced hypomethylation in PDAC (MIA PaCa-2 and PANC1) and normal pancreas cells (HPNE) using a 5-day treatment with 5-aza-2-cytidine and indeed saw a dramatic upregulation of *IKZF1* transcript in the PDAC cells with a more modest increase in HPNE cells (**Extended Data Fig. 1B**). However, this increase in *IKZF1* transcription was not correlated with a decrease in *PIKFYVE* (**Extended Data Fig. 1C**), suggesting that this hypothesis was not correct.

We next tested whether the increase in *PIKFYVE* expression was simply due to increased proliferation rates of cancer versus normal cells. To do this, we arrested proliferation using thymidine in three PDAC cell lines and one normal cell line for one doubling time. We hypothesized that, if the levels of *PIKFYVE* are higher in highly proliferating cells, blocking proliferation should decrease *PIKFYVE* levels. However, thymidine treatment did not consistently decrease *PIKFYVE* levels in the cells (**Extended Data Fig. 1D**), suggesting that this is also not the correct explanation.

Our next hypothesis was that PDAC cells upregulate *PIKFYVE* in response to its nutritional microenvironment. PDAC is known to have an elevated use and dependency on lysosomal processes due to its densely packed, nutrient-disrupted microenvironment. We, thus, attempted to model this in cell culture by seeding cells at different confluences and saw that cells growing at higher confluency have higher levels of *PIKFYVE* expression (**Extended Data Fig. 1E**). Additionally, when we replaced the media of the cells with low serum (1% FBS) or lipid-depleted serum, cells consistently increased *PIKFYVE* expression (**Extended Data Fig. 1F**). This suggested that the nutritional conditions of the cells can influence the levels of *PIKFYVE* expression and may explain why PDAC cells *in vivo* have higher levels of *PIKFYVE* compared to normal cells. These results are also supported by our new mechanistic understanding of the interplay between *de novo* lipid biosynthesis and lipid scavenging, which place PIKfyve activity as a central mediator of lipid homeostasis (see more in responses 2-4 below).

We again thank the reviewer for inspiring this set of experiments. While the precise mechanism of transcriptional regulation of *PIKFYVE* remains unclear, our data demonstrates that cells upregulate *PIKFYVE* in response to limited nutrients. This is in line with current literature regarding the upregulated lysosomal processes in PDAC. These new data have been added to the revised manuscript.

2. it is unclear how it activates *de novo* fatty acid synthesis through SREBP-1?

Response: We thank the reviewer for identifying this question and agree that our study would be stronger with a clear delineation of the connection between PIKfyve, SREBP1, and *de novo* fatty acid synthesis. To this end, we performed a series of experiments to strengthen these connections and have included all the data in the revised manuscript.

First, upon suggestion by Reviewer #3, we hypothesized that PIKfyve inhibition activates SREBP1 and *de novo* fatty acid synthesis through disruption of cellular energetics and AMPK. However, while direct inhibition of AMPK indeed activated SREBP1, PIKfyve inhibitors activate SREBP1

without affecting AMPK activity, as indicated by phosphorylation of ACC1 (**Extended Data Fig. 10A**). Consistent with this, knockout or knockdown of AMPK α did not impact the effect of PIKfyve inhibition on SREBP1 activation or *FASN* or *ACACA* expression (**Extended Data Fig. 10B-E**). These data suggested that PIKfyve inhibition does not activate *de novo* fatty acid synthesis through disruption of cellular energetics.

Next, we hypothesized that PIKfyve inhibition activates *de novo* fatty acid synthesis through disrupting lysosomal lipid turnover, a mechanism proposed by Hosios et al. (PMID: 36536136). Consistent with this mechanism, non-PIKfyve lysosomal inhibitors (i.e., chloroquine and bafilomycin) also increased SREBP1 activation and *FASN* and *ACACA* expression in PDAC cell lines (**Extended Data Fig. 11A-B**). Additionally, in Hosios et al., they show that lysosomal lipid turnover mechanisms do not require autophagic processes. To see if our findings align with this mechanism, we also used ATG5 and ATG7 knockout monoclonal PDAC cell lines, which we believe are effectively autophagy-deficient (**Extended Data Fig. 13A**). Utilizing these cells, we found that being autophagy-deficient did not impact the relationship of synthetic lethality between PIKfyve and fatty acid synthesis; ATG5 and ATG7 knockout cells were sensitized by ACC1 inhibition to PIKfyve inhibitors to a similar degree as control cells (**Extended Data Fig. 13B-D**). Similarly, inhibition of PIKfyve in ATG5 and ATG7 cells still increased SREBP1 activation and *FASN* and *ACACA* expression to the same degree as it did in control cells (**Extended Data Fig. 14A-B**). These data suggested that the autophagy pathway is not necessary for the relationship between PIKfyve, SREBP1, and *de novo* fatty acid synthesis and are in line with the mechanism of lysosomal lipid turnover. Additionally, inhibition of ACC1 did not sensitize cells to the ULK1 inhibitor SBI-0206965 (PMID: 26118643) to the same degree as it did to apilimod (**Extended Data Fig. 13E-G**), and inhibition of ULK1 alone did not activate SREBP1 (**Extended Data Fig. 14C**). These data suggest that autophagy inhibition is not sufficient to activate SREBP1 and *de novo* fatty acid synthesis, further suggesting that PIKfyve inhibition does not cause the observed effects of lipid metabolism primarily through disrupting autophagic flux.

Finally, while in our original manuscript we showed that PIKfyve inhibition activates SREBP1 and increases expression of *FASN* and *ACACA*, we did not provide evidence that these two were causally related. To assess whether PIKfyve inhibition increased *FASN* and *ACACA* expression through activation of SREBP1, we utilized an inhibitor of SREBP1 called fatostatin, which successfully attenuated both the activation of SREBP1 as well as the downstream increase in *FASN* and *ACACA* observed upon PIKfyve inhibition (**Extended Data Fig. 9A-B, adapted in part as Reviewer Fig. 1A-B below**). To further confirm that the effects of PIKfyve inhibition on the lipid profile of PDAC cells is also caused by SREBP1 activation, we performed targeted lipidomics on PDAC cells treated with DMSO, apilimod, ESK981, fatostatin, apilimod + fatostatin, or ESK981 + fatostatin. Consistent with this mechanism, the combination of fatostatin and PIKfyve inhibitors attenuated the impact of PIKfyve inhibitors alone on both the global lipid profile (**Extended Data Fig. 16D, copied as Reviewer Fig. 1C below**), as well as specifically sphingolipids (**Extended Data Fig. 16E, copied as Reviewer Fig. 1D below**), suggesting that SREBP1 activation was required for the effect of PIKfyve inhibition on PDAC lipid metabolism.

Taken together, our additional data strengthen the connections between PIKfyve, SREBP1, and *de novo* fatty acid synthesis. We now provide evidence for a mechanism whereby PIKfyve inhibition disrupts lipid metabolism through blocking lysosomal lipid turnover, causing an activation of *de novo* fatty acid synthesis through the activation of SREBP1.

Reviewer Figure 1: A) Immunoblot analysis of 7940B cells upon treatment with apilimod (100nM), ESK981 (1000nM), fatostatin (20 μ M), apilimod + fatostatin, or ESK981 + fatostatin as indicated for 8 hours showing changes in SREBP1 (P) and SREBP1 (M). Vinculin was used as a loading control. **B)** qPCR showing changes in *Fasn* (left) or *Acaca* (right) upon treatment with apilimod (100nM for 7940B, 300nM for PANC1 and MIA PaCa-2), ESK981 (1000nM), fatostatin

(20 μ M), apilimod + fatostatin, or ESK981 + fatostatin as indicated for 8 hours. (One way ANOVA with Šídák's multiple comparisons test). **C)** Principal component analysis (PCA) of targeted lipidomics experiment on 7940B cells treated with DMSO, apilimod (100 nM), ESK981 (1000 nM), fatostatin (20 μ M), apilimod + fatostatin, or ESK981 + fatostatin for 24 hours. Arrows are provided to highlight the change in lipid profile upon addition of fatostatin to apilimod or ESK981. Of note, one datapoint was removed from the DMSO group as a statistical outlier. **D)** Relative abundance of ceramides (left) and hexosylceramides (right) in 7940B upon treatment with apilimod (100nM), ESK981 (1000nM), fatostatin (20 μ M), apilimod + fatostatin, or ESK+ fatostatin for 24 hours. (Unpaired two-tailed t-tests). ***This new data has been added as Extended Data 9A-B and Extended Data Fig. 16D-E, respectively in the revised manuscript.***

3. it is unclear how its inhibition suppresses autophagy in PDAC or what role autophagy plays in connection with induction of fatty acid synthesis?

Response: We thank the reviewer again for the suggestion to strengthen the connection between PIKfyve, autophagy, and fatty acid synthesis. Previous work has shown that PI(3,5)P2 and PI5P, products of PIKfyve, promote autophagy (PMID: 25578879, 19793721). Consistent with this, previous work done by our group and others have shown that PIKfyve inhibition decreases the levels of PI(3,5)P2 and PI5P (PMID: 25355904) and disrupts autophagic flux (PMID: 34738088, 38011559). To strengthen this link between PIKfyve and autophagy, we have performed radioactive inositol labeling to determine the phosphorylation levels of phosphoinositides in PANC1 cells.

Indeed, PIKfyve inhibition or degradation decreased levels of PI(3,5)P2 and PI5P (**Fig. 2B-C, copied as Reviewer Fig. 2A-B, and Extended Data Fig. 4H-I**) and increased levels of the substrate PI3P (**Extended Data Fig. 3A and 4J**). This effect was observed rapidly, after just 30 minutes of adding the compounds. This effect was also specific to these phosphoinositides, as PI4P and PI(4,5)P2 were not significantly changed (**Extended Data Fig. 3B-C and 4K-L**). These data were added to the manuscript. We also included clarifying language and citations connecting PIKfyve, the phosphoinositides, and autophagy.

Reviewer Figure 2: A) Relative abundance of PI(3,5)P2 in PANC1 cells upon treatment with apilimod (1000nM) or ESK981 (1000nM) for the indicated times. (One-way ANOVA with Dunnett's). **B)** Relative abundance of PI5P in PANC1 cells upon treatment with apilimod (1000nM) or ESK981 (1000nM) for the indicated times. (One-way ANOVA with Dunnett's). ***This new data has been added as Fig. 2B-C in the revised manuscript.***

Regarding the connection between autophagy and fatty acid synthesis, as described in the previous point, we in fact found that inhibition of autophagy alone (using ULK1 as a target) was not sufficient to recapitulate the effects of PIKfyve inhibition on fatty acid synthesis. Additionally, PIKfyve inhibition obligated autophagy-deficient cells to upregulate and depend on fatty acid synthesis to a similar degree compared to control cells, again suggesting that this relationship does not primarily depend on autophagy. Instead, we believe this effect to be through affecting lysosomal lipid turnover as described in Hosios et al. (PMID: 36536136), which was suggested to not primarily depend on autophagy. We thank the reviewer again for asking this question as these were crucial details to clarify. We believe these data substantially strengthen our model and will provide new important insights to the field.

4. Moreover, how does KRAS-MAPK signaling cross-talk with PIKfyve at both levels of regulation of autophagy and fatty acid synthesis?

Response: We thank the reviewer for highlighting the two mechanistic connections between PIKfyve and KRAS-MAPK signaling that we propose: fatty acid synthesis and autophagy, and we agree that this is an important opportunity for further characterization.

KRAS-MAPK signaling is the principal vehicle for anabolic metabolism in PDAC. Reciprocally, KRAS-MAPK inhibition turns off “cell building”, and therefore potentiates autophagy to support cellular energetic homeostasis (PMID: 30833752, 30833748, 30709910). This was faithfully recapitulated in our systems (**Extended Data Fig. 20F**). The concept of blocking the “protective” autophagy induced upon KRAS-MAPK inhibition has been leveraged to develop combination therapies using autophagy inhibitors with KRAS-MAPK inhibitors, and these approaches have shown promising results in preclinical models and even in the clinical setting (PMID: 30833748, 33225411), (NCT04386057, NCT04132505). However, as a point of clarity, those published studies actually used lysosomal inhibitors, and they use autophagy inhibition as a readout for lysosomal inhibition.

In either case, in our study, given that we have shown PIKfyve perturbation to inhibit autophagic flux (**Fig. 3A, Extended Data Fig. 4B-E, G**), we also tested whether PIKfyve inhibition could block the “protective autophagy” induced by KRAS-MAPK inhibition in PDAC cells. Indeed, while KRAS-MAPK perturbation increased autophagic flux, the addition of a PIKfyve inhibition attenuated this increase in lysosomal activity in our PDAC cell models (**Fig. 4F-G, Extended Data Fig. 20G**). **Indeed, this interconnection serves as one arm for the synthetic lethal opportunity we detail and exploit in this manuscript. The second arm of the synthetic lethal opportunity involves the interconnection between KRAS-MAPK and PIKfyve through lipid metabolism, which we detail below.**

Regarding the connection between KRAS-MAPK and PIKfyve through fatty acid synthesis, we previously established that PIKfyve inhibition activates fatty acid synthesis through SREBP1 (**Fig. 3N**). We are confident in this connection given that inhibition of SREBP1 activation attenuated the effects of PIKfyve inhibition of fatty acid synthesis (**Extended Data Fig. 9A-B, 16D-E**). Similarly, KRAS signaling is also known to drive fatty acid synthesis through SREBP1, both downstream of PI3K/AKT and MAPK (PMID: 29295482), and we have included this detail in the revised manuscript. However, in our updated manuscript, we also provide evidence that KRAS-MAPK drives fatty acid synthesis through regulation of the transcription factor MYC. KRAS is known to drive PDAC cell metabolism in part through MYC transcriptional regulation (PMID: 22541435). After just eight hours of KRAS or MEK inhibition, MYC expression is dramatically decreased in PDAC cells (**Extended Data Fig. 19A-B**). We then performed chromatin immunoprecipitation followed by sequencing (ChIP-seq) and showed that, while MYC binds to the *FASN* promoter, KRAS or MEK inhibition decreases this binding after eight hours of treatment (**Extended Data Fig. 19E, copied as Reviewer Fig. 3A**).

This finding is in line with a previous report that MEK inhibition decreases the binding of MYC to the *FASN* promoter (PMID: 35771494). Of note, this early time point is also the time used for our quantitative PCR analyses showing that KRAS-MAPK inhibition decreases *FASN* transcription (**Extended Data Fig. 20B**). Importantly, in line with our proposed model, KRAS or MEK inhibition opposed the effects of PIKfyve inhibition on the global lipid profile and, specifically, the increase in synthesis of sphingolipids (**Fig. 4E, Extended Data Fig. 20E, copied as Reviewer Fig. 3B-C, respectively**), which we showed conferred resistance to PIKfyve inhibition (**Fig. 3S**). Taken together, our data suggests that PIKfyve inhibition stimulates fatty acid synthesis through SREBP1 while KRAS-MAPK inhibition represses fatty acid synthesis, at least in part, through MYC.

Reviewer Figure 3: A) ChIP-seq track of *FASN* gene locus showing peaks called by Model-based analysis of ChIP-seq (MACS, PMID: 18798982) and MYC binding motif called by Find Individual Motif Occurrences (FIMO, PMID: 21330290) in MIA PaCa-2 cells upon treatment with DMSO, AMG510, or trametinib. **B)** Principal component analysis of targeted lipidomics data showing shifts in 7940B global lipid profiles upon treatment with DMSO, apilimod (100nM, 24 hours), trametinib (100nM every 24 hours for 48 hours), or MRTX1133 (1000nM every 8 hours for 48 hours). Of note, one datapoint was removed from the DMSO group as a statistical outlier. **C)**

Relative abundance of ceramides (left) and hexosylceramides (right) in 7940B upon treatment with apilimod (100nM for 24 hours), trametinib (100nM every 24 hours for 48 hours), MRTX1133 (1000nM every 8 hours for 48 hours) apilimod + trametinib, or apilimod + MRTX. (Unpaired two-tailed t-tests). ***This new data has been added as Extended Data Fig. 19E, Fig. 4E, and Extended Data Fig. 20E, respectively, in the revised manuscript.***

5. And can these questions be addressed in the context of PDAC versus normal pancreatic tissue?

Response: We greatly appreciate this question from the reviewer and recognize its importance given the translational potential of our findings. To model normal pancreas cells in culture, we employed the hTERT-immortalized pancreatic epithelial cell HPNE. We first noticed that HPNE cells expressed lower levels of *FASN*, *ACACA*, and *SREBF1* (the gene encoding SREBP1) compared to two human PDAC cell lines PANC1 and MIA PaCa-2 (**Extended Data Fig. 12A**), which is in line with previous reports that these pathways are in part driven by KRAS signaling (PMID: 29295482). Next, we assessed how HPNE cells adapt to PIKfyve inhibition. Given that our findings suggest that PIKfyve inhibition disrupts lipid metabolism through a lysosome-dependent lipid turnover mechanism, we expected that the response to PIKfyve inhibition should be conserved between PDAC versus normal pancreatic cells. Indeed, HPNE cells also activated SREBP1 and increased expression of *FASN* and *ACACA* in an SREBP1-dependent manner, in line with this model (**Extended Data Fig. 12B-C**).

However, in contrast to PDAC cells, our genetically engineered mouse model showed that *Pikfyve* deletion in normal pancreas cells is well-tolerated, in terms of pancreas weight and function (**Fig. 1E-G, Extended Data Fig. 2A-B**). Similarly, inhibition of PIKfyve is well-tolerated in mice and in humans, as evidenced by our preclinical studies (**Extended Data Fig. 3E**) and a phase I clinical trial (PMID: 25152243). Additionally, knowing that normal pancreas cells should not harbor KRAS mutations or an overactive KRAS-MAPK signaling pathway, the synthetic lethality relationship between KRAS-MAPK inhibitors and PIKfyve inhibitors is not expected to apply to normal pancreas cells. Along these lines, the combination of PIKfyve inhibitor ESK981 and KRAS or MEK inhibitors was well-tolerated in mice (**Extended Data Fig. 22B**) and extended survival of KPC mice (**Fig. 4O, Extended Data Fig. 22I**). Collectively, these results suggest that there is a significant therapeutic window for dual application of PIKfyve inhibitors and KRAS-MAPK inhibitors.

Minor comments:

1. Autophagy detection is not thorough; Interpretation of data in Figures 3B and 4D are conflicting; according to the authors, in 3B: the ratio of LC3B II/I increases and is indicative of decreased autophagy; in 4D: the ratio of II/I is however also increased upon KRAS extinction (minus Dox) and this is interpreted as autophagy induction (lines 381-384), with p62 levels being decreased in parallel.

Response: We thank the reviewer for this comment and agree that we can improve the explanation of data regarding autophagy markers. On immunoblots, an increased ratio of LC3 II/I

can represent either an induction of autophagy or a blockade of LC3II degradation (PMID: 25484342). To determine directionality, in the LC3 II/I ratio, a lysosomal inhibitor (commonly bafilomycin A1) can be used to block lysosomal degradation of LC3 II. Using this approach, if an increased LC3 II/I ratio is retained upon lysosomal blockade, then that condition likely increased autophagic flux. Further, we used p62 as a second immunoblot marker to assist with our interpretation. In principle, p62, which is degraded by autophagy, is easier to interpret. An increase in p62 represents a blockade of autophagy while a decrease in p62 represents an induction of autophagy.

In our studies of PIKfyve inhibition, we used a combination of LC3 II/I and p62, and in all cases, we saw that both the LC3 II/I and p62 increased upon PIKfyve inhibition (**Fig. 3A, Extended Data Fig. 4B, C, G**) suggesting that PIKfyve inhibition decreased autophagic flux. In contrast, in our studies of KRAS perturbation, and when combining PIKfyve and KRAS perturbation, we saw that LC3 II/I was increased with KRAS OFF, and that the increased ratio was maintained when adding apilimod or ESK981, which served as our lysosomal inhibitors in this experiment (**Fig. 4D**). Consistent with this, p62 was decreased upon KRAS OFF but increased with the addition of apilimod or ESK981, again suggesting that KRAS OFF increased autophagic while PIKfyve inhibition decreased autophagic flux.

Employing immunoblots to monitor autophagic flux is complex and imperfect; thus, we also employed a second method to monitor autophagic flux: the GFP-LC3-RFP-LC3 Δ G autophagic flux probe (PMID: 27818143). The data generated using this system were consistent with our interpretations of the immunoblots and served as a powerful confirmation of our conclusions regarding the impact of PIKfyve and KRAS on autophagy in PDAC (**Extended Data Fig. 4D-E, Extended Data Fig. 20F-G, Fig. 4F-G**). We have improved our language in the manuscript explaining these findings, and we thank the reviewer again for pointing out this area for improvement.

2. Subcutaneous transplants are not considered good models for PDAC, as they do not faithfully recapitulate the stroma-dense, nutrient deprived/hypoxic environment nor do they recapitulate PDAC metabolism (Figures 2, 4 and Extended Data Fig. 2).

Response: We thank the reviewer for this comment and agree with this sentiment. To address this, we have performed three additional efficacy studies: two using orthotopic models and one using the KPC autochthonous model.

Our first orthotopic model was a syngeneic orthotopic model in which we implanted 7940B (KPC) cells into immune competent hosts (C57B/6) to test the efficacy of ESK981 as a single agent. This study showed that ESK981 alone was able to reduce tumor burden after three weeks of treatment (**Fig. 2J-K**), although we did not achieve any cures as we did with the combination of ESK981 with trametinib (**Fig. 4I-L**). This study also provided evidence that ESK981 alone was well-tolerated by mice, as all mice appeared healthy and maintained their starting weight throughout the trial (**Extended Data Fig. 3E**).

In our second experiment, we tested the efficacy of ESK981 as a single agent on tumors derived from human PDAC cells in the orthotopic setting using UM19 primary cells. In this study, ESK981 was also able to decrease tumor burden alone, though again it did not cure the mice (**Fig. 2 N-O**).

Our third experiment employed the gold standard, autochthonous KPC model. Here, we had 6 arms: vehicle, ESK981, selumetinib (MEK inhibitor), MRTX1133 (KRAS^{G12D} inhibitor), ESK981 + selumetinib, and ESK981 + MRTX1133. In this study, we utilized 3D ultrasound to determine when to initiate treatment and to follow tumor size. We found that the combination of ESK981 and selumetinib resulted in complete (100%) regression in 4 out of 6 tumors and partial regression in 1 out of 6 tumors, while selumetinib alone only resulted in partial regression in 3 out of 8 tumors, and ESK981 resulted in partial regression in 1 out of 6 tumors (**Fig. 4Q-R, copied as Reviewer Fig. 4A-B**). Importantly, ESK981 + selumetinib extended the median survival of these mice to 58.5 days compared to 11 days for the vehicle group (**Fig. 4P, copied as Reviewer Fig. 4C**). To our knowledge, this is one of the most dramatic effects reported for this model. Additionally, while MRTX1133 had impressive single-agent effects, ESK981 did increase the efficacy of MRTX1133, causing deeper regressions in tumor size (**Extended Data Fig. 22K-L**). We believe that these additional efficacy studies using orthotopic or endogenous models strongly support the efficacy we showed previously using the subcutaneous models, and they significantly improve our confidence in the translational applicability of our findings. We again thank the reviewer, as well as the editor, for emphasizing this point.

Reviewer Figure 4: **A)** Tumor volumes of KPC mice treated with vehicle, selumetinib (selu), ESK981 (ESK), or selumetinib + ESK981, as measured by 3D ultrasound on the indicated treatment day. **B)** Waterfall plot of maximal tumor response to each therapy. **C)** Kaplan-Meier survival curves of KPC mice undergoing indicated therapies. (Gehan-Breslow-Wilcoxon test). *This new data has been added as Figure 4P-R in the revised manuscript.*

3. The Discussion section is repetitive, re-stating and summarizing the already well-described results; would be best to use to highlight limitations, future steps of the study etc.

Response: We thank the reviewer for this critique and agree that we should use the Discussion section to cover aspects highlighted by the reviewer. We have changed the text accordingly.

4. It is in general good practice to avoid use of absolute language when describing findings; e.g, “never” (line 130) and “first” (lines 449 and 469) -the authors do mention in the discussion a report involving PYKfive in lipid metabolism through its lysosomal function(line 471).

Response: We greatly thank the reviewer for pointing this out, and we have removed the instances where we use this language.

Referee #2 (Remarks to the Author):

Combining *in vivo* and *in vitro* approaches, the authors identify a key role for PIKfyve in PDAC progression. In addition, by applying a CRISPR-based screen they uncover synthetic lethality of *de novo* lipid synthesis created by PIKfyve depletion linked to the upregulation of key lipid synthetic enzymes (FASN and ACACA) induced by PIKfyve depletion/inhibition. The authors also uncover that KRAS-MAPK inhibition counteracts the increased expression of FASN and ACACA induced by PIKfyve depletion/inhibition and that combining PIKfyve and KRAS-MAPK inhibition synergistically suppresses PDAC growth.

Overall, the data presented in the manuscript are of high quality and fully support the conclusions. Each of the pathways explored by the authors, i.e. PIKfyve and RAS-MAPK, has a pivotal and widely recognized role in PDAC progression to the point that, as the authors mention, there are two clinical trials testing the efficacy of combined treatment of MAPK inhibitors with the autophagy flux inhibitor cloroquine, and an ongoing phase II clinical trial testing the efficacy of the PIKfyve inhibitor ESK981 in pancreatic cancer patients.

Response: We thank the reviewer for the thorough summary of our findings and particularly for highlighting the quality of the data and reflecting that the data fully support the conclusions. We also thank the reviewer for sharing our enthusiasm for our findings and particularly the translational implications of our study. We also appreciate the reviewer's comments and suggestions regarding our manuscript. All points have been addressed in a point-wise fashion below, and the associated changes to the manuscript are colored in light blue.

1. The original and important finding of the manuscript consists in the discovery that PIKfyve inhibition reshuffles the lipogenic transcriptional program of PDAC cells making their growth dependent on *de novo* fatty acid synthesis and that RAS-MAPK has a role in sustaining the *de novo* lipid biosynthesis.

However, the enthusiasm is tempered by the lack of analysis of the molecular mechanisms underpinning the lipid program reshuffling and of the possible molecular links bridging PIKfyve-dependent and MAPK-dependent pathways of lipid metabolism control.

Response: We thank the reviewer for emphasizing the key findings of our study as well as the critiques raised regarding the lack of molecular mechanisms characterizing the relationships between PIKfyve, MAPK, and lipid metabolism. *To address this, we performed a series of experiments that are detailed below, and all the associated data have been added to the updated manuscript.*

We first focused on characterizing the relationship between PIKfyve and lipid metabolism. Questions regarding this relationship were communicated by Reviewer #1 above (point 2), and we reiterate partitions of that reply here. Upon suggestion by Reviewer #3, we hypothesized that

PIKfyve inhibition activates SREBP1 and *de novo* fatty acid synthesis through disruption of cellular energetics and AMPK. However, while direct inhibition of AMPK indeed activated SREBP1, PIKfyve inhibitors activate SREBP1 without affecting AMPK activity, as indicated by phosphorylation of ACC1 (**Extended Data Fig. 10A**). Consistent with this, knockout or knockdown of AMPK α did not impact the effect of PIKfyve inhibition on SREBP1 activation or *FASN* or *ACACA* expression (**Extended Data Fig. 10B-E**). These data suggested that PIKfyve inhibition does not activate *de novo* fatty acid synthesis through disruption of cellular energetics.

Next, we hypothesized that PIKfyve inhibition activates *de novo* fatty acid synthesis through disrupting lysosomal lipid turnover, a mechanism proposed by Hosios et al. (PMID: 36536136). Consistent with this mechanism, non-PIKfyve lysosomal inhibitors (i.e., chloroquine and bafilomycin) also increased SREBP1 activation and *FASN* and *ACACA* expression in PDAC cell lines (**Extended Data Fig. 11A-B**). Additionally, in Hosios et al., they show that lysosomal lipid turnover mechanisms do not require autophagic processes. To see if our findings align with this mechanism, we also used ATG5 and ATG7 knockout monoclonal PDAC cell lines, which we believe are effectively autophagy-deficient (**Extended Data Fig. 13A**). Utilizing these cells, we found that being autophagy-deficient did not impact the relationship of synthetic lethality between PIKfyve and fatty acid synthesis; ATG5 and ATG7 knockout cells were sensitized by ACC1 inhibition to PIKfyve inhibitors to a similar degree as control cells (**Extended Data Fig. 13B-D**). Similarly, inhibition of PIKfyve in ATG5 and ATG7 cells still increased SREBP1 activation and *FASN* and *ACACA* expression to the same degree as it did in control cells (**Extended Data Fig. 14A-B**). These data suggested that the autophagy pathway is not necessary for the relationship between PIKfyve, SREBP1, and *de novo* fatty acid synthesis and are in line with the mechanism of lysosomal lipid turnover. Additionally, inhibition of ACC1 did not sensitize cells to the ULK1 inhibitor SBI-0206965 (PMID: 26118643) nearly to the same degree as it did to apilimod (**Extended Data Fig. 13E-G**), and inhibition of ULK1 alone did not activate SREBP1 (**Extended Data Fig. 14C**). These data suggest that autophagy inhibition is not sufficient to activate SREBP1 and *de novo* fatty acid synthesis, further substantiating the mechanism that PIKfyve inhibition causes the observed effects on lipid metabolism through affecting lysosomal lipid homeostasis, an autophagy-independent mechanism.

Finally, while in our original manuscript we showed that PIKfyve inhibition activates SREBP1 and increases expression of *FASN* and *ACACA*, we did not provide evidence that these two were causally related. To assess whether PIKfyve inhibition increased *FASN* and *ACACA* expression through activation of SREBP1, we utilized an inhibitor of SREBP1 called fatostatin, which successfully attenuated both the activation of SREBP1 as well as the downstream increase in *FASN* and *ACACA* observed upon PIKfyve inhibition (**Extended Data Fig. 9A-B, adapted as Reviewer Fig. 5A-B**). To further confirm that the effects of PIKfyve inhibition on the lipid profile of PDAC cells is also caused by SREBP1 activation, we performed targeted lipidomics on PDAC cells treated with DMSO, apilimod, ESK981, fatostatin, apilimod + fatostatin, or ESK981 + fatostatin.

Consistent with this mechanism, the combination of fatostatin and PIKfyve inhibitors attenuated the impact of PIKfyve inhibitors alone on both the global lipid profile (**Extended Data Fig. 16D**,

copied as Reviewer Fig. 5C), as well as specifically sphingolipids (**Extended Data Fig. 16E, copied as Reviewer Fig. 5D**), suggesting that SREBP1 activation was required for the effect of PIKfyve inhibition on PDAC lipid metabolism.

Taken together, our additional data strengthens the connections between PIKfyve, SREBP1, and *de novo* fatty acid synthesis and suggests a mechanism whereby PIKfyve inhibition disrupts lipid metabolism through blocking lysosomal lipid turnover, causing an activation of *de novo* fatty acid synthesis through the activation of SREBP1.

We next sought to characterize the relationship of MAPK and lipid metabolism to bridge the connection between PIKfyve and MAPK signaling through lipids. Questions regarding this relationship were communicated by Reviewer #1 above (point 4), and so for ease of viewing, we also reiterated portions of that reply here. KRAS signaling is also known to drive fatty acid synthesis through SREBP1, both downstream of PI3K/AKT and MAPK (PMID: 29295482), and we have included this detail in the revised manuscript. However, in our updated manuscript we also provide evidence that KRAS-MAPK drives fatty acid synthesis through regulation of the transcription factor MYC. KRAS is known to drive PDAC cell metabolism in part through MYC transcriptional regulation (PMID: 22541435). After just eight hours of KRAS or MEK inhibition, MYC expression is dramatically decreased in PDAC cells (**Extended Data Fig. 19A-B**). We then performed chromatin immunoprecipitation followed by sequencing (ChIP-seq) and showed that, while MYC binds to the *FASN* promoter, KRAS or MEK inhibition decreases this binding after eight hours of treatment (**Extended Data Fig. 19E, copied as Fig. R6A**).

This finding is in line with a previous report that MEK inhibition decreases the binding of MYC to the *FASN* promoter (PMID: 35771494). Of note, this early time point is also the time used for our quantitative PCR analyses showing that KRAS-MAPK inhibition decreases *FASN* transcription (**Extended Data Fig. 20B**). Importantly, in line with our proposed model, KRAS or MEK inhibition opposed the effects of PIKfyve inhibition on the global lipid profile and specifically the increase in synthesis of sphingolipids (**Fig. 4E, Extended Data Fig. 20E, copied as Reviewer Fig. 6B-C, respectively**), which we showed conferred resistance to PIKfyve inhibition (**Fig. 3S**). Taken together, our data suggests that PIKfyve inhibition stimulates fatty acid synthesis through SREBP1 while KRAS-MAPK inhibition represses fatty acid synthesis, at least in part, through MYC.

In sum, our additional data supports a model whereby PIKfyve inhibition disrupts lysosomal lipid turnover, causing PDAC cells to activate SREBP1 which in turn induces a lipogenic transcriptional and metabolic program to favor *de novo* fatty acid synthesis, specifically sphingolipids. On the other hand, KRAS inhibition decreases MYC levels, blocking MYC's binding to the *FASN* promoter which causes decreased expression of *FASN*. This results in decreased fatty acid synthesis and opposes the effect of PIKfyve inhibition on the lipid profile of PDAC cells and specifically on the regulation of sphingolipids. We thank this reviewer for highlighting these areas for further study, and we believe this data substantially improves the quality of our manuscript.

Reviewer Figure 5: **A)** Immunoblot analysis of 7940B cells upon treatment with apilimod (100nM), ESK981 (1000nM), fatostatin (20 μ M), apilimod + fatostatin, or ESK981 + fatostatin as indicated for 8 hours showing changes in SREBP1 (P) and SREBP1 (M). Vinculin was used as a loading control. **B)** qPCR showing changes in *Fasn* (left) or *Acaca* (right) upon treatment with apilimod (100nM for 7940B, 300nM for PANC1 and MIA PaCa-2), ESK981 (1000nM), fatostatin (20 μ M), apilimod + fatostatin, or ESK981 + fatostatin as indicated for 8 hours. (One way ANOVA with Šídák's multiple comparisons test). **C)** Principal component analysis (PCA) of targeted lipidomics experiment on 7940B cells treated with DMSO, apilimod (100 nM), ESK981 (1000 nM), fatostatin (20 μ M), apilimod + fatostatin, or ESK981 + fatostatin for 24 hours. Arrows are provided to highlight the change in lipid profile upon addition of fatostatin to apilimod or ESK981. Of note, one datapoint was removed from the DMSO group as a statistical outlier. **D)** Relative abundance of ceramides (left) and hexosylceramides (right) in 7940B upon treatment with apilimod (100nM),

ESK981 (1000nM), fatostatin (20 μ M), apilimod + fatostatin, or ESK+ fatostatin for 24 hours. (Unpaired two-tailed t-tests). *This new data has been added as Extended Data 9A-B and Extended Data Fig. 16D-E, respectively, in the revised manuscript.*

Reviewer Figure 6: A) ChIP-seq track of *FASN* gene locus showing peaks called by Model-based analysis of ChIP-seq (MACS, PMID: 18798982) and MYC binding motif called by Find Individual Motif Occurrences (FIMO, PMID: 21330290) in MIA PaCa-2 cells upon treatment with DMSO,

AMG510, or trametinib. **B)** Principal component analysis of targeted lipidomics data showing shifts in 7940B global lipid profiles upon treatment with DMSO, apilimod (100nM, 24 hours), trametinib (100nM every 24 hours for 48 hours), or MRTX1133 (1000nM every 8 hours for 48 hours). Of note, one datapoint was removed from the DMSO group as a statistical outlier. **C)** Relative abundance of ceramides (left) and hexosylceramides (right) in 7940B upon treatment with apilimod (100nM for 24 hours), trametinib (100nM every 24 hours for 48 hours), MRTX1133 (1000nM every 8 hours for 48 hours) apilimod + trametinib, or apilimod + MRTX. (Unpaired two-tailed t-tests). ***These new data have been added as Extended Data Fig. 19E, Fig. 4E, and Extended Data Fig. 20E, respectively, in the revised manuscript.***

2. Also, the very intriguing alteration of sphingolipid metabolism is left completely unexplored as regards possible molecular mechanisms.

Response: We appreciate this comment from the reviewer and agree that the accumulation of sphingolipids upon PIKfyve inhibition is an interesting finding that deserves more study. To address this, we performed a series of experiments characterizing this phenomenon. ***All of the data have been added to the updated manuscript.***

We first set forth to determine if alterations in sphingolipids were due to *de novo* synthesis. Our first clue was derived from the targeted lipidomics experiment that we performed using fatostatin to block SREBP1 activation. Here, we saw that the addition of fatostatin to apilimod- or ESK981-treated conditions attenuated the increase in sphingolipids, specifically ceramides and hexosylceramides (**Extended Data Fig. 16E, copied as Reviewer Fig. 5B**). This implied that the increase in sphingolipids was due to SREBP1 activation upon PIKfyve inhibition; however, this did not definitively indicate that sphingolipids were being synthesized *de novo* upon PIKfyve inhibition. Thus, we performed stable isotope tracing using U-¹³C₆ glucose and followed sphingolipid biosynthesis by liquid chromatography-coupled mass spectrometry. This data showed a clear and substantial increase in ¹³C incorporation in ceramides upon PIKfyve inhibition versus control (**Fig. 3Q-R, copied as Reviewer Fig. 7A-B, and Extended Data Fig. 17A-F**). This strongly suggested that there was an increase in sphingolipids upon PIKfyve inhibition and that the carbon used for sphingolipid biosynthesis came at least in part from glucose, in line with our water-soluble metabolomics data.

Given that PIKfyve inhibition stimulated sphingolipid synthesis, we next asked whether this was an adaptive mechanism and whether sphingolipid synthesis provided PDAC cells with a survival benefit in the context of PIKfyve inhibition. While our original metabolic CRISPR screen using low-dose apilimod as a selective pressure did not identify sphingolipid biosynthesis-specific genes as being synthetically lethal with apilimod, we hypothesized that this was due to a low degree of selective pressure. We hypothesize that this was why genes such as *FASN* stood out so significantly. However, we performed an additional screen in parallel with the original screen using a higher dose of apilimod (1000nM vs 200nM). After analyzing this screen, we in fact found that three genes specific to sphingolipid biosynthesis, *SPTLC1*, *SPTLC2* (both subunits of serine palmitoyltransferase), and *KDSR*, were among the top hits, along with our previous hits (**Extended Data Fig. 18B-C; 18B is copied as Reviewer Fig. 7C**). This suggested that the rate

limiting steps of fatty acid synthesis present a bottleneck for PDAC cells upon PIKfyve inhibition; whereas there is more redundancy downstream, sphingolipid synthesis is a metabolic liability following PIKfyve inhibition. Finally, to validate this, we utilized CRISPR interference to knock down *SPTLC1* and *SPTLC2* in PDAC cells and found that while *SPTLC1* or *SPTLC2* knockdown did not affect cell viability (in line with recent reports PMID: 39112706), they did sensitize PDAC cells to PIKfyve inhibition (**Fig. 3S, copied as Reviewer Fig. 7D**). Taken together, our additional data builds on our previous finding that sphingolipids were increased upon PIKfyve inhibition, confirming that this was due to increased synthesis utilizing glucose, and that this is an adaptive mechanism of PDAC in response to PIKfyve loss. We again thank the reviewer for this crucial comment as it inspired this series of interesting experiments that resulted in data that substantially improved our study.

Reviewer Figure 7: A) C20 ceramide (d18:1/20:0) isotopologue distribution in 7940B cells upon incubation with $U-^{13}C_6$ glucose and treatment with DMSO, apilimod (100nM), or ESK91 (1000nM) for 24 hours. **B)** C20 ceramide (d18:1/20:0) isotopologue distribution in 7940B cells upon incubation with $U-^{13}C_6$ glucose and treatment with DMSO, apilimod (100nM), or ESK91 (1000nM)

for 24 hours, binned as indicated. (Two-way ANOVA with Dunnett's). **C)** Gene enrichment rank plot based on differential sgRNA representation in high-dose apilimod-treated versus DMSO-treated endpoint populations of the CRISPR screen experiment. Genes involved in sphingolipid synthesis are highlighted in red. Genes specifically used for sphingolipid synthesis are further highlighted with a blue box. **D)** Luminescence values from CellTiter-Glo analyses on PANC1 (left) or MIA PaCa-2 (right) cells upon knockdown of *SPTLC1*, *SPTLC2*, or control, and treatment with apilimod (3000nM) or DMSO for the indicated duration. (Two-way ANOVA with Dunnett's). ***This new data has been added as Figure 3Q-R, Extended Data Figure 18B, and Figure 3S, respectively, in the revised manuscript.***

3. Along the same lines, the authors did not consider that PIKfyve controls not only autophagy but also many different trafficking steps along the endocytic pathway and that these might have an impact on signaling circuits that affect lipid metabolism programmes.

Response: We thank the reviewer for this insightful comment. As mentioned in the response to this reviewer's first point, we demonstrated that the mechanism through which PIKfyve controls lipid metabolism extends to lysosomal programs beyond autophagy. We approached this question by determining the necessity of autophagy in this relationship and found that autophagy-deficient cells still responded to PIKfyve with increased expression and dependency on fatty acid synthesis (**Extended Data Fig. 14A-B and Extended Data Fig. 13A-D**). Our data also suggest that inhibiting autophagy alone (using an inhibitor of ULK1) does not phenocopy the impact of lipid metabolism mediated by PIKfyve inhibition (**Extended Data Fig. 14C and Extended Data Fig. 13E-G**).

Our data suggests that PIKfyve inhibition disrupts lysosomal lipid turnover, similar to the mechanism reported in Hoisos et al. (PMID: 36536136). This study revealed that mTORC1 inhibition blocked *de novo* fatty acid synthesis, which resulted in enhanced lysosomal lipid turnover to adapt to the loss of *de novo* fatty acid synthesis. One way they validated this mechanism was by blocking lipid turnover with lysosomal inhibitors, including a PIKfyve inhibitor. Thus, our findings closely align with this study and show that cells can also adapt to loss of lysosomal lipid turnover by increasing *de novo* fatty acid synthesis.

4. In this context, it is worth mentioning that the upregulation of FASN and ACA transcripts induced by PIKfyve depletion/inhibition had already been reported in muscle (PMID: 23673157) and that, in this case, the deregulation of lipid metabolism was found to be dependent on altered glut4 trafficking to the plasma membrane and a possible consequent activation of CHREB-dependent targets. In the present manuscript, the nature of possible signaling circuits and/or transcription factors involved are left unaddressed.

Response: We thank the reviewer for this comment and for bringing this provocative study to our attention. In this study, the authors conditionally knock out *Pikfyve* in murine striated muscle cells and find that mice exhibited a number of phenotypes including systemic glucose intolerance, decreased muscle insulin sensitivity, increased body weight, and adiposity. In terms of the connection of PIKfyve and *Fasn* and *Acaca*, the authors saw that *Pikfyve* loss in striated muscle

cells led to an increased expression of *Fasn* and *Acaca* in epididymal fat, which suggested that the increased adiposity could be due to increased fatty acid synthesis. This also seems consistent with the other phenotypes, such as *Pikfyve* loss causing decreased insulin sensitivity and glucose intolerance. In our study, we are particularly interested in the effects of PIKfyve loss on lipid metabolism and the expression of lipid synthesis genes in a cell autonomous system. However, this does raise the interesting point of investigating the effect of PIKfyve loss on lipid metabolism in other cells in the tumor microenvironment. It is possible that other cells may support PDAC by providing exogenous lipids, which may affect PDAC tumor sensitivity to PIKfyve inhibitors. We have added this provocative concept to the Discussion section of our manuscript and cited the relevant literature.

As referenced in the response to the reviewer's first comment, in the process of interrogating the mechanism through which PIKfyve regulates lipid metabolism, we also investigated the possibility of other signaling circuits such as AMPK and cellular energetics playing a role in PIKfyve-controlled lipid homeostasis. However, our data suggested that PIKfyve did not regulate lipid metabolism through regulating AMPK (**Extended Data Fig. 10A-E**). Additionally, to strengthen the connection between PIKfyve, SREBP1, and fatty acid synthesis, we performed rescue experiments to show that PIKfyve indeed affects cellular lipid metabolism through SREBP1 (**Extended Data Fig. 9A-B, Extended Data Fig. 16D-E**). All this data has been added to the revised manuscript. We again thank the reviewer for their insightful comments and suggestions and believe that the additional work inspired by them have substantially improved the quality of our manuscript.

Referee #3 (Remarks to the Author):

Re: Targeting PIKFyve-driven lipid homeostasis as a metabolic vulnerability in pancreatic cancer

Summary

This manuscript describes a novel role for the PI kinase PIKFyve in pancreatic cancer. It begins with the finding that PIKFyve expression is higher in PDAC lesions compared to adjacent pancreas in mouse models of PDAC and human PDAC patients. The authors cross two different genetic PDAC driver models, KC and KPC, with PIKFyve knock out mice and find that animals lacking PIKFyve develop fewer PDAC lesions and have decreased tumor burden compared to their wild-type littermates. They also find that loss of PIKFyve does not impact the development of normal pancreas. Small molecule inhibitors of PIKFyve elicit similarly striking results in these models. I commend the authors on a rigorous characterization of PIKFyve treatment in PDAC models in the first two figures of this paper.

The authors next find that PIKFyve inhibition disrupts normal lysosomal trafficking in PDAC models. A previous publication characterized the role of PIKFyve in lysosomal trafficking and the authors note that PDAC is a disease that is particularly vulnerable to this type of manipulation. An in vitro CRISPR screen of PDAC cell lines treated with a PIKFyve inhibitor revealed that genes specifically involved in de novo lipogenesis (FASN, ACACA, and ELOVL1) are essential for treatment survival. As validation, the authors perform a CRISPR knockdown of these genes and find they indeed synergize with PIKFyve inhibitor treatment to kill PDAC cells.

Water-soluble metabolomics experiments show global decreases in glycolytic metabolites (G6P, F16BP, PEP, pyruvate) after treatment with PIKFyve inhibitors. Meanwhile, the authors observe differences in some TCA cycle metabolites: succinate and aconitate are increased while alpha ketoglutarate and citrate decreased. These observations imply that PDAC cells are using other metabolites such as glutamine to fuel the TCA cycle in response to PIKFyve inhibitor treatment, but this point is beyond the scope of this paper. These data lead the authors to conclude that PDAC cells are shunting carbon atoms away from glycolysis and toward de novo lipogenesis, as suggested by the results of their CRISPR screen. Global lipidomics reveals that sphingolipids are dramatically increased upon PIKFyve inhibitor treatment, likely to compensate for increased de novo palmitic acid synthesis. Experiments in the final figure of this paper show that treatment with a PIKFyve inhibitor and a KRAS inhibitor synergistically kills PDAC cells. The authors imply this combined treatment paradigm is effective because PIKFyve inhibition and KRAS inhibition have opposite effects on lipogenesis and that creates a metabolic conflict in PDAC cells.

Response: We thank the reviewer for the detailed summary of our findings and for the positive review of our study. We greatly appreciate the commendation of this reviewer on our rigorous efforts to characterize the effects of PIKFyve inhibition on PDAC models. We also thank the

reviewer for the insightful comments regarding the possibility of PDAC utilizing different metabolites to fuel the TCA cycle in response to PIKfyve inhibitor. We agree with the reviewer that this is an interesting finding and worth further study, but we also agree with the reviewer in that it may be out of scope for this particular manuscript. We also thank the reviewer for their questions and suggestions. All points have been addressed in a point-wise fashion below, and all significant changes to the manuscript are colored in light blue.

Outstanding Questions.

1. The biggest outstanding question for this manuscript is: why do PDAC cells upregulate *de novo* lipogenesis when PIKfyve activity is disrupted? The observation that KRAS-MAPK controls *de novo* lipogenesis is interesting and presents an outstanding treatment paradigm when combined with PIKfyve inhibitors. However, the synergistic effect of PIKfyve inhibition and KRAS inhibition on both lipogenesis and survival implies that these two pathways are independently regulating lipogenesis. Additionally, their CRISPER validation screen shows that silencing genes in the *de novo* lipogenesis pathway potentiates the lethality of PIKfyve inhibitors. Together, these observations imply that upregulation of *de novo* lipogenesis is a potential survival mechanism for cells treated with PIKfyve inhibitors. Do the authors have thoughts on why lipogenesis increases with PIKfyve inhibition and what survival advantage this gives to PDAC cells? Do the authors have any experiments planned to explore this mechanism?

Response: We again thank the reviewer for highlighting these interesting observations from our study. We agree that the finding that KRAS-MAPK controls *de novo* lipogenesis, which we believe to be an adaptive mechanism that PDAC cells utilize in response to PIKfyve inhibitors, presents a very exciting therapeutic strategy. We thank the reviewer for sharing our enthusiasm about this point.

We also agree that the reason why PDAC cells upregulate lipogenesis upon PIKfyve inhibition is an important question for further interrogation. In fact, this question was raised by the other two reviewers as well. To address this question, we performed a series of experiments and added all of the data to the updated manuscript. Questions regarding this relationship were communicated by Reviewer #1 (point 2) and Reviewer #2 (point 1), and we reiterate portions of that reply here.

First, upon this reviewer's suggestion, we hypothesized that PIKfyve inhibition activates SREBP1 and *de novo* fatty acid synthesis through disruption of cellular energetics and AMPK. However, while direct inhibition of AMPK indeed activated SREBP1, PIKfyve inhibitors activate SREBP1 without affecting AMPK activity, as indicated by phosphorylation of ACC1 (**Extended Data Fig. 10A**). Consistent with this, knockout or knockdown of AMPK α did not impact the effect of PIKfyve inhibition on SREBP1 activation or *FASN* or *ACACA* expression (**Extended Data Fig. 10B-E**). These data suggested that PIKfyve inhibition does not activate *de novo* fatty acid synthesis through disruption of cellular energetics.

Next, we hypothesized that PIKfyve inhibition activates *de novo* fatty acid synthesis through disrupting lysosomal lipid turnover, a mechanism proposed by Hosios et al. (PMID: 36536136). Consistent with this mechanism, non-PIKfyve lysosomal inhibitors (i.e. chloroquine and bafilomycin) also increased SREBP1 activation and *FASN* and *ACACA* expression in PDAC cell lines (**Extended Data Fig. 11A-B**). Additionally, in Hosios et al., they show that lysosomal lipid turnover mechanisms do not require autophagic processes. To see if our findings align with this mechanism, we also used ATG5 and ATG7 knockout monoclonal PDAC cell lines, which we believe are effectively autophagy-deficient (**Extended Data Fig. 13A**). Utilizing these cells, we found that being autophagy-deficient did not impact the relationship of synthetic lethality between PIKfyve and fatty acid synthesis; ATG5 and ATG7 knockout cells were sensitized by ACC1 inhibition to PIKfyve inhibitors to a similar degree as control cells (**Extended Data Fig. 13B-D**). Similarly, inhibition of PIKfyve in ATG5 and ATG7 cells still increased SREBP1 activation and *FASN* and *ACACA* expression to the same degree as it did in control cells (**Extended Data Fig. 14A-B**). These data suggested that the autophagy pathway is not necessary for the relationship between PIKfyve, SREBP1, and *de novo* fatty acid synthesis and are in line with the mechanism of lysosomal lipid turnover. Additionally, inhibition of ACC1 did not sensitize cells to the ULK1 inhibitor SBI-0206965 (PMID: 26118643) nearly to the same degree as it did to apilimod (**Extended Data Fig. 13E-G**), and inhibition of ULK1 alone did not activate SREBP1 (**Extended Data Fig. 14C**). These data suggest that autophagy inhibition is not sufficient to activate SREBP1 and *de novo* fatty acid synthesis, further substantiating the mechanism that PIKfyve inhibition causes the observed effects on lipid metabolism through affecting lysosomal lipid homeostasis, an autophagy-independent mechanism.

In sum, our data suggests that PIKfyve disrupts lysosomal lipid turnover, a mechanism described in Hosios et al. (PMID: 36536136). This study identified this mechanism through inhibiting *de novo* fatty acid synthesis by inhibiting mTORC1 and finding that cells utilized lysosomal lipid turnover to adapt to the loss of *de novo* fatty acid synthesis. One way they validated this mechanism was by rescuing this lipid turnover by using lysosomal inhibitors, including a PIKfyve inhibitor. Thus, our findings closely align with this study and show that cells can also adapt to loss of lysosomal lipid turnover by increasing *de novo* fatty acid synthesis. We propose that PDAC uses the lysosomes not just for recycling of iron and metabolites, but also for lipids, and that blocking this lysosomal lipid recycling forces PDAC cells to upregulate fatty acid synthesis.

2. On a related note, increased lipogenesis suggests an overall change in the energy demands of the cell. Have the authors performed experiments to look at the global metabolic state of the cell with markers like AMPK phosphorylation after PIKfyve inhibitor treatment?

Response: We thank the reviewer for this suggestion, and we performed a series of experiments investigating this possibility. We incorporated this suggestion into the reply to Reviewer #3 point 1 (paragraph 3), and we reiterated portions of that reply explanation here.

As suggested, we assessed the impact of PIKfyve inhibition on AMPK activity and whether AMPK signaling was involved in the regulation of lipogenesis by PIKfyve. While direct inhibition of AMPK

indeed activated SREBP1, PIKfyve inhibitors activate SREBP1 without affecting AMPK activity, as indicated by phosphorylation of ACC1 (**Extended Data Fig. 10A**). Consistent with this, knockout or knockdown of AMPK α did not impact the effect of PIKfyve inhibition on SREBP1 activation or *FASN* or *ACACA* expression (**Extended Data Fig. 10B-E**). These data suggested that PIKfyve inhibition does not activate *de novo* fatty acid synthesis through disruption of cellular energetics.

3. From the metabolomics data, the authors propose that glycolytic metabolites are being shunted to *de novo* lipogenesis. Lipidomics analysis shows that sphingolipids are significantly increased in response to PIKfyve inhibitor treatment. The authors imply that this response, presumably mediated by the activity of the enzyme SPT1, compensates for increased *de novo* palmitate synthesis. Have the authors considered validating this claim with glucose tracing experiments into sphingolipids? This may be a good way to shed light on mechanistically how PDAC cells respond to PIKfyve inhibitor treatment.

Response: We thank the reviewer for this insightful suggestion. We have performed the U-¹³C₆ glucose stable isotope tracing experiment suggested, and the data support our hypothesis. These data showed a clear and substantial increase in ¹³C incorporation in ceramides upon PIKfyve inhibition versus control (**Fig. 3Q-R, copied as Reviewer Fig. 8A-B, and Extended Data Fig. 17A-F**). This strongly suggested that there is an increase in sphingolipids upon PIKfyve inhibition, and that the carbon used for sphingolipid biosynthesis is derived, at least in part, from glucose, in line with our aqueous metabolomics data. This was an excellent suggestion from the reviewer, and this data has been added to the revised manuscript.

Reviewer Figure 8: A) C20 ceramide (d18:1/20:0) isotopologue distribution in 7940B cells upon incubation with U-¹³C₆ glucose and treatment with DMSO, apilimod (100nM), or ESK91 (1000nM) for 24 hours. **B)** C20 ceramide (d18:1/20:0) isotopologue distribution in 7940B cells upon incubation with U-¹³C₆ glucose and treatment with DMSO, apilimod (100nM), or ESK91 (1000nM) for 24 hours, binned as indicated. (Two-way ANOVA with Dunnett's). ***This new data has been added as Figure 3Q-R in the revised manuscript.***

4. If upregulation of sphingolipid biosynthesis is an important survival mechanism for cells treated with PIKfyve inhibitors, why wasn't SPT1 identified as essential in their CRISPR screen? If cells are shunting palmitic acid produced through de novo lipogenesis into sphingolipids for survival, elimination of SPT1 should be lethal to these cells. The observation that this enzyme did not fall out of their screen suggests there may be another fate of palmitate in their system. Have the authors explored this possibility further with additional experiments? Does SPT1 knock down potentiate the lethality of PIKfyve inhibition? Do other lipid classes, such as triglycerides, change with treatment to protect PDAC cells from higher levels of palmitate?

Response: This is another excellent suggestion from the reviewer, and we greatly appreciate it. A similar question was raised by Reviewer #2 above (point 2) and is in part reiterated here. While our original metabolic CRISPR screen using low-dose apilimod as a selective pressure did not identify sphingolipid biosynthesis-specific genes as being synthetically lethal with apilimod, we hypothesized that this was due to a low degree of selective pressure. We hypothesize that this was why genes such as *FASN* stood out so significantly. However, we performed an additional screen in parallel with the original screen using a higher dose of apilimod (1000nM vs 200nM). After analyzing this screen, we in fact found that three genes specific to sphingolipid biosynthesis, *SPTLC1*, *SPTLC2* (both subunits of serine palmitoyltransferase), and *KDSR*, were among the top hits, along with our previous hits (**Extended Data Fig. 18B-C; 18B is copied as Reviewer Fig. 9A**). This suggested that the rate limiting steps of fatty acid synthesis present a bottleneck for PDAC cells upon PIKfyve inhibition; whereas there is more redundancy downstream, sphingolipid synthesis is a metabolic liability following PIKfyve inhibition. Finally, to validate this, we utilized CRISPR interference to knock down *SPTLC1* and *SPTLC2* in PDAC cells and found that while *SPTLC1* or *SPTLC2* knockdown did not affect cell viability (in line with recent reports PMID: 39112706), they did sensitize PDAC cells to PIKfyve inhibition (**Fig. 3S, copied as Reviewer Fig. 9B**). Taken together, our additional data builds on our previous finding that sphingolipids were increased upon PIKfyve inhibition, confirming that this was due to increased synthesis utilizing glucose, and that this is an adaptive mechanism of PDAC in response to PIKfyve loss. These data have been added to the revised manuscript.

Given that sphingolipid biosynthesis genes were identified in this additional screen, and also because sphingolipids were the most highly upregulated lipid classes upon PIKfyve inhibition, we focused our efforts on sphingolipid biology. Regarding the role of triglycerides, we found that triglycerides decrease upon PIKfyve inhibition, which is a separate interesting observation that deserves further study. However, we believe this may be out of scope for this particular manuscript.

Reviewer Figure 9: A) Gene enrichment rank plot based differential sgRNA representation in high-dose apilimod-treated versus DMSO-treated endpoint populations of the CRISPR screen experiment. Genes involved in sphingolipid synthesis are highlighted in red. Genes specifically used for sphingolipid synthesis are further highlighted with a blue box. **B)** Luminescence values from CellTiter-Glo analyses on PANC1 (left) or MIA PaCa-2 (right) cells upon knockdown of *SPTLC1*, *SPTLC2*, or control, and treatment with apilimod (3000nM) or DMSO for the indicated duration. (Two-way ANOVA with Dunnett's). ***This new data has been added as Extended Data Figure 18B and Figure 3S, respectively, in the revised manuscript.***

5. The final figure of the paper leads the authors to hypothesize that since KRAS inhibition and PIKfyve inhibition have opposite effects on lipogenesis, inhibiting both pathways synergistically kills PDAC cells. However, no lipidomics data are provided for the combined treatment conditions. Do the authors have this data? If not, they should perform this experiment.

Response: We thank the reviewer for another excellent suggestion. To address this, we performed targeted lipidomics on PDAC cells upon treatment with DMSO, apilimod, trametinib, MRTX1133, apilimod + trametinib, or apilimod+MRTX. In line with our hypothesis and model, we found that KRAS or MEK inhibition opposed the effects of PIKfyve inhibition on the global lipid profile, and that the combination of KRAS-MAPK inhibition with PIKfyve inhibition resulted in a lipid profile closer to that of DMSO than that of either treatment alone. Specifically, KRAS or MEK inhibition decreased synthesis of sphingolipids (**Fig. 4E, Extended Data Fig. 20E, copied as Reviewer Fig. R10A-B, respectively**), which we showed conferred resistance to PIKfyve inhibition (**Fig. 3S**). We believe that this data substantiates our claim that PIKfyve inhibition and KRAS-MAPK inhibition have opposing effects on not just transcriptional regulation of fatty acid synthesis but also the lipid profile as well. Altogether, this data bolsters our confidence in this model and the concept of combining KRAS-MAPK and PIKfyve inhibitors as a therapeutic strategy for PDAC. This data has been added to the revised manuscript.

Reviewer Figure 10: A) Principal component analysis of targeted lipidomics data showing shifts in 7940B global lipid profiles upon treatment with DMSO, apilimod (100nM, 24 hours), trametinib (100nM every 24 hours for 48 hours), MRTX1133 (1000nM every 8 hours for 48 hours), or the indicated combinations. Of note, one datapoint was removed from the DMSO group as a statistical outlier. **B)** Relative abundance of ceramides (left) and hexosylceramides (right) in 7940B cells upon treatment with apilimod (100nM for 24 hours), trametinib (100nM every 24 hours for 48 hours), MRTX1133 (1000nM every 8 hours for 48 hours), apilimod + trametinib, or apilimod + MRTX. (Unpaired two-tailed t-tests). ***This new data has been added as Fig. 4E and Extended Data Fig. 20E, respectively, in the revised manuscript.***

Overall thoughts.

My opinion of this paper is very positive. It is very well written, and the authors do an excellent job of walking the reader through the logic of their experiments and interpreting their results and conclusions. The first two figures describing the effects of PIKFyve inhibition on PDAC growth are particularly noteworthy. The authors do a rigorous job characterizing the role of PIKFyve in a variety of PDAC models and leave the reader with little doubt about the importance of their findings. The final figure of the paper is also exciting: PIKFyve inhibition alongside the MAPK inhibitor MRTX1133 showed dramatic decreases in tumor growth in vivo. If these results remain consistent in the treatment of patients, their findings represent a novel and important step in the treatment of an aggressive disease with often very few treatment options.

In my opinion, the data quality is for this paper is very high the manuscript contains good descriptions and references to the experimental approaches used. Most of their conclusions seem reasonable. However, they need to conduct more metabolic experiments to support their conclusions on the role of lipid metabolism in response to

PIKFyve inhibitor treatment. Overall, I think this is a great paper and should be published in *Nature* as a revised manuscript addressing the metabolic mechanistic questions I raised.

Response: We thank the reviewer again for the very positive outlook on our study. We greatly appreciate again the reviewer highlighting our first two figures in commenting that we “leave the reader with little doubt about the importance of our findings”. We also thank the reviewer for commenting that our data is “very high” and that it should be published in such a prestigious journal as *Nature*. We hope that the metabolic experiments we have performed have addressed the reviewer’s concerns, and we again thank the reviewer for providing such insightful suggestions.

Response to Referees for Manuscript 2024-03-05156B-Z

We are again truly grateful for the time and effort put forth by the editors and reviewers at *Nature* and for the chance to submit another revised manuscript for your consideration. We have addressed the comments raised by Reviewer 2, which we believe have led to substantial improvements in the quality of our manuscript.

Referee #1 (Remarks to the Author):

The authors have adequately addressed all my comments and concerns. Thank you for your diligence.

Response: We thank the reviewer for their insightful comments and suggestions throughout the review process, and for appreciating our diligence. We are grateful that we have addressed all their comments and concerns.

Referee #2 (Remarks to the Author):

In revising the manuscript, the authors have attempted to address the major criticisms raised by this reviewer regarding the lack of mechanistic insight into the lipid metabolism reprogramming induced by PIKfyve inhibition. In response, the authors demonstrate that SREBP activation not only accompanies PIKfyve inhibition (as reported in the original manuscript) but is also necessary for the lipid reshuffling that follows. While this finding identifies an important component of the pathway, the molecular mechanism underlying SREBP activation in response to PIKfyve inhibition remains unexplored.

Response: We thank the reviewer for highlighting our previous findings that SREBP plays a causative role in lipid reshuffling upon PIKfyve inhibition. We also agree that while this is an important finding, there were still components of the mechanism connecting PIKfyve and SREBP activation that were unaddressed. We have performed a series of experiments that have clarified additional elements of this relationship that we believe address the reviewer's concerns. Below, we have addressed these comments in a point-by-point response where we describe the additional data that we have generated to address the reviewer's specific comments. The data have all been added to the manuscript and are highlighted below.

The authors hypothesize that lysosomal dysfunction plays a role in this activation, citing the work by Hosios et al. to support their hypothesis. However, it is crucial to clarify that Hosios et al. do not show that lysosome inhibition activates *de novo* fatty acid (FA) synthesis. Instead, their study demonstrates that, only under conditions of mTORC inhibition, lysosomal activity is necessary to promote triglyceride (TG) synthesis by increasing FA release, while lysosome inhibition prevents this FA release in response to mTORC inhibition. Additionally, Hosios et al. show that mTORC1 inhibition suppresses SREBP activity (which is expected, as mTORC1 is a major inducer of SREBP-mediated lipid remodelling), but they do not provide evidence that lysosomal inhibitors activate SREBP under conditions of mTORC1 inhibition.

Response: We thank the reviewer for this comment, and we regret our subpar exposition of the work by Hosios et al. As the reviewer highlighted, Hosios et al. does not show that lysosomal inhibition activates *de novo* lipid synthesis but instead shows that, under mTORC (and thus *de novo* synthesis) inhibition, cells employ lysosomes for membrane lipid turnover, resulting in fatty acid release and triglyceride synthesis. In contrast, as described in detail below, we show that when PIKfyve is inhibited and lysosome function is disrupted, lipids are sequestered in lysosomes, decreasing cellular lipid availability, forcing cells to increase SREBP activation and *de novo* lipid synthesis. Described another way, while Hosios et al. showed that mTORC/SREBP inhibition leads to utilization of lysosomal activity, we show that lysosome inhibition leads to utilization of SREBP activity. These mechanisms seem to have some overlap, and, thus, we found the Hosios et al. study particularly helpful in shaping our understanding. However, they are clearly different, and we regret implying that they may be the same. As a note, Hosios et al. does employ a PIKfyve inhibitor in their study, which blocked fatty acid release from lysosomes upon mTORC1 inhibition. This further supports the idea that PIKfyve inhibition may block lysosomal lipid release and, thus, disrupt cellular lipid metabolism in certain cellular contexts. Again, we thank the reviewer for bringing up this important clarification, and we have added these elements to the Discussion section of our updated manuscript.

In the revised manuscript, the authors do present data showing that lysosomal inhibitors, such as chloroquine, activate SREBP, and use this correlation to suggest that PIKfyve inhibition may operate through a similar mechanism. However, this still leaves the molecular link between lysosomal dysfunction and SREBP activation unresolved.

Further studies are needed to elucidate the specific connection between PIKfyve inhibition, lysosomal dysfunction, and SREBP activation, as this remains a critical gap in the mechanistic understanding of lipid remodelling in this context.

Further mechanistic studies are needed to establish how these two events are connected and to determine whether PIKfyve inhibition directly influences SREBP activation. Clarifying the nature of this link is particularly important, given that the authors also find SREBP activation to be necessary for sphingolipid remodelling. A deeper understanding of how SREBP activation is triggered in response to PIKfyve inhibition would provide valuable insight into the broader lipid metabolism reprogramming observed in this study.

Response: We thank the reviewer again for this insightful discussion and for commenting on areas where our work made progress and where more work could be done. We agree with these comments and, herein, provide data to support a clarified model connecting PIKfyve inhibition, lysosomal dysfunction, disruption of lipid metabolism, and SREBP activation, as well as downstream lipid remodeling.

First, while our previous manuscript version focused primarily on the effect of PIKfyve inhibition on creating a synthetic lethal relationship with fatty acids and sphingolipids, our CRISPR screen and RNA-seq experiments also indicated that this relationship held true with cholesterol.

Specifically, in our CRISPR screen with high dose PIKfyve inhibition (IC80; low dose, IC20), genes involved in cholesterol synthesis, such as *FDPS*, *FDFT1*, *SQLE*, and *LSS*, presented as synthetically essential upon PIKfyve inhibition (**Reviewer Figure 1A-E, from Fig. 3C-E and Extended Data Fig. 7N-O in the revised manuscript**). Likewise, many genes involved in cholesterol acquisition and synthesis were strongly upregulated upon PIKfyve inhibition, including *HMGCR*, *LDLR*, *FDFT1*, *SQLE*, and others (**Reviewer Figure 2A-D, from Fig. 3H-I and Extended Data Fig. 8B and I in the revised manuscript**). Given that cholesterol and sphingolipids play crucial roles as structural lipids in cell membranes, we hypothesized that metabolism of cell membranes were central to this mechanism.

Redacted

Reviewer Figure 2: A) Pathway enrichment analysis of RNA-seq performed on 7940B KPC cells treated with either apilimod (AP; 25 nM) or ESK981 (ESK; 250 nM) for 8 hours. Dot sizes are inversely proportional to false discovery rate (FDR). The color scheme is reflective of the normalized enrichment score (NES). **B)** GSEA plots of cholesterol homeostasis using the fold change rank-ordered gene signature from the 7940B cells treated with apilimod (top) or ESK981 (bottom) for 8 hours. **C)** Volcano plot of RNA-seq analysis on 7940B cells treated with apilimod (25 nM) for 8 hours highlighting SREBP target genes. Vertical dashed lines indicate \log_2 fold change = ± 0.5 . Horizontal dashed line indicates $\text{FDR} = 10^{-6}$. **D)** qPCR showing changes in RNA levels of *Sqle*, *Fdft1*, *Hmgcr*, or *Ldlr* in 7940B cells upon treatment with apilimod (100nM), or ESK981 (1000nM) for 8 hours. (One-way ANOVA with Dunnett's).

As the reviewer highlighted, our previous data suggested that lysosomal dysfunction was likely part of the connection between PIKfyve inhibition and SREBP activation (**Extended Data Fig. 16A-B**). Knowing now that cholesterol homeostasis is also central to this relationship, we employed the free cholesterol stain filipin to assess the impact of PIKfyve inhibition on lysosomes and cholesterol trafficking. As expected, we found that PIKfyve inhibition induced lysosomal vacuolization (Rivero-Ríos and Wesiman, *Curr Opin Cell Biol*, 2022; Qiao et al., *Nature Cancer*, 2021), and, notably, prominent accumulation of cholesterol on the lysosomal membranes (**Reviewer Figure 3A, from Extended Data Fig. 17A in the revised manuscript**). We further investigated whether such disruption of lysosomal cholesterol trafficking could be the reason why PIKfyve inhibition causes the observed SREBP activation. To this end, we utilized U18666A, an inhibitor of NPC1, which exports cholesterol out of the lysosome (Lu et al., *Elife*, 2015) as a comparative tool. U18666A treatment led to accumulation of cholesterol at lysosomal (LAMP1 positive) puncta (**Reviewer Figure 4A, from Extended Data Fig. 17A in the revised manuscript**), consistent with previous reports (Poh et al., *Antiviral Res*, 2012). Importantly, we also found that blockade of lysosomal export of cholesterol phenocopied PIKfyve inhibition in terms of SREBP activation and upregulation of *FASN*, suggesting that lysosomal trapping of lipids was sufficient to activate SREBP (**Reviewer Figure 4B-C, from Extended Data Fig. 16C-D in the revised manuscript**). Further, we confirmed that this effect was observed as early as 4 hours of treatment (**Reviewer Figure 5A-B, from Extended Data Fig. 19A-B in the revised**

manuscript), which was the earliest timepoint discernable SREBP activation was detected, further linking these two observations (**Extended Data Fig. 18D-E**). Of note, lysosomal membranes are known to harbor sphingolipids and other lipids, which raises the possibility that other lipids are also accumulating at lysosomal membranes. Knowing that accumulation of cholesterol inside lysosomes was sufficient to activate SREBP and that PIKfyve inhibition leads to accumulation of cholesterol on lysosomal membranes, we concluded that this was a key mechanism through which PIKfyve inhibition activates SREBP.

Reviewer Figure 3: A) Immunofluorescence images of PANC1 cells treated with DMSO, apilimod (AP; 1000nM) or ESK981 (ESK; 1000nM) for 24 hours stained with filipin or LAMP1. Scalebars= 5µm. The images for the DMSO are the same as those used in Reviewer Figure 4A, as these data were generated from the same experiment and are displayed as a unified figure in the manuscript.

Reviewer Figure 4: A) Immunofluorescence images of PANC1 cells treated with DMSO or U18666A (5μg/ml) for 24 hours and stained with filipin or LAMP1. Scalebars= 5μm. The images for the DMSO are the same as those used in Reviewer Figure 3, as these data were generated from the same experiment and are displayed as a unified figure in the manuscript. **B)** Immunoblots of PANC1 and MIA PaCa-2 cells treated with apilimod (AP; 300nM), ESK981 (ESK; 1000nM), chloroquine (CQ; 30μM), bafilomycin (BAF; 30nM), or U18666A (5μg/ml) for 8 hours showing changes in premature SREBP1 (P) and mature SREBP1 (M). Histone H3 or vinculin are used as loading controls. **C)** qPCR of MIA PaCa-2, PANC1, and 7940B cells assessing changes in RNA of *FASN* upon treatment with apilimod (100nM for 7940B, 300nM for PANC1, MIA PaCa-2), or U18666A (5μg/ml) for 8 hours. (One-way ANOVA with Dunnett's).

Reviewer Figure 5: A) Immunofluorescence images of PANC1 cells upon treatment with DMSO or apilimod (300nM) for 4 hours stained with filipin and LAMP1. Scalebars=5 μ m. **B)** Immunofluorescence images of 7940B cells upon treatment with DMSO, apilimod (100nM), ESK981 (1000nM), or U18666A (5 μ g/ml) for 4 hours stained with filipin. Scalebars=20 μ m.

Next, we asked the question of why accumulation of lipids in lysosomes led to SREBP activation. One of the most well-known ways SREBP is activated is through sterol deprivation. When sterols are abundant, they prevent SREBP cleavage-activating protein (SCAP) from binding SREBP. Under sterol deprivation, SCAP binds SREBP and escorts it to the Golgi where it undergoes proteolytic cleavage and activation. This raises the possibility that lysosomal accumulation of

lipids effectively sequesters them, leading to cellular lipid starvation. Consistent with this model, we were able to rescue the activation of SREBP1 and the upregulation of downstream genes by supplementing cells with free cholesterol (10 μ g/ml cholesterol and 1 μ g/ml 25-hydroxycholesterol) (Reviewer Figure 6A-B, from Fig. 3N-O and Extended Data Fig. 20A-B in the revised manuscript). With these combined data, we concluded that PIKfyve inhibition results in lysosomal vacuolization, which sequesters lipids on lysosomal membranes, forcing PDAC cells to activate SREBPs and downstream lipid reshuffling (Reviewer Figure 7, adapted from Fig. 5 in the revised manuscript). We thank the reviewer again for highlighting this concern, and we believe that the additional data we provide to address the reviewer's comments greatly improve the clarity of our mechanistic model and overall manuscript quality.

Reviewer Figure 6: A) Immunoblot of MIA PaCa-2 and PANC1 cells upon treatment with DMSO, apilimod (AP; 300nM), or ESK981 (ESK; 1000nM) with or without cholesterol supplementation (10 μ g/ml cholesterol and 1 μ g/ml 25-hydroxycholesterol) for 8 hours detecting changes of premature SREBP1 (P) and mature SREBP1 (M) protein levels. Vinculin or Histone H3 were used as a loading control. **B)** qPCR showing changes in RNA levels of *HMGCR* in MIA PaCa-2 and PANC1 cells upon treatment with DMSO, apilimod (300nM), or ESK981 (1000nM) with or without cholesterol supplementation (10 μ g/ml cholesterol and 1 μ g/ml 25-hydroxycholesterol) for 8 hours. (One way ANOVA with Šídák's multiple comparisons test).

Redacted

Referee #3 (Remarks to the Author):

The authors have done an outstanding job addressing the comments made by myself and the other reviewers. I highly recommend the revised manuscript for publication in *Nature*.

Response: We thank the reviewer for their insightful comments and suggestions throughout the review process, and we are grateful that we have addressed all their comments and concerns. We are especially thankful for the strongly positive review and recommendation for publication of this work in *Nature*.

Response to Referees for Manuscript 2024-03-05156C

Referees' comments:

Referee #2 (Remarks to the Author):

The authors have satisfactorily addressed my comments.

Response: We thank the reviewer for their helpful comments and questions throughout the review process. We are glad we have addressed their comments through this round of revisions.